# ReTabAD: A Benchmark for Restoring Semantic Context in Tabular Anomaly Detection

**Sanghyu Yoon[1]**[*]**, Dongmin Kim[1]**[*]**, Suhee Yoon[1], Ye Seul Sim[1], Seungdong Yoa[1]**
**Hye-Seung Cho[1], Soonyoung Lee[1], Hankook Lee[2]**[‡]**, Woohyung Lim[1]**[†]
[1]LG AI Research, Seoul, South Korea    [2]Sungkyunkwan University, Suwon, South Korea
{sanghyu.yoon, dmkim, suhee.yoon, ysl.sim, seungdong.yoa,
hs.cho, soonyoung.lee, w.lim}@lgresearch.ai, hankook.lee@skku.edu
[*]Equal contribution    [†]Corresponding author

## Abstract

In tabular anomaly detection (AD), textual semantic context often carries critical signals, as the definition of an anomaly is closely tied to domain-specific context. However, existing benchmarks provide only raw data points without semantic context, overlooking rich textual metadata such as feature descriptions and domain knowledge that experts rely on in practice. This limitation restricts research flexibility and prevents models from fully leveraging domain knowledge for detection. **ReTabAD** addresses this gap by **Re**storing textual semantics to enable context-aware **Tab**ular **A**nomaly **D**etection research. We provide (1) 20 carefully curated tabular datasets enriched with structured textual metadata, together with implementations of state-of-the-art AD algorithms—including classical, deep learning, and LLM-based approaches—and (2) a zero-shot LLM framework that leverages semantic context without task-specific training, establishing a strong baseline for future research. Furthermore, this work provides insights into the role and utility of textual metadata in AD through experiments and analysis. Results show that semantic context improves detection performance and enhances interpretability by supporting domain-aware reasoning. These findings establish ReTabAD as a benchmark for systematic exploration of context-aware AD. The resource is publicly released at https://yoonsanghyu.github.io/ReTabAD/.

## 1 Introduction

*Anomaly detection (AD)*, the task of identifying patterns that diverge from the distribution of normal data, represents a fundamental challenge across diverse domains such as finance (Al-Hashedi & Magalingam, 2021), cybersecurity (Landauer et al., 2023), manufacturing (Kharitonov et al., 2022), and healthcare (Fernando et al., 2021). In particular, anomaly detection in tabular data, *i.e.*, *tabular AD*, is of critical importance, as structured datasets—including sensor logs, transaction records, and clinical measurements—form the backbone of real-world operational systems. Despite its practical importance, tabular AD remains a challenging problem due to the heterogeneous nature of tabular datasets, which often combine numerical, categorical, and even textual attributes with rich semantic meaning.

DAMI Repository (Campos et al., 2016) and ADBench (Han et al., 2022) have made significant contributions to tabular AD by establishing standardized benchmarks that enabled fair and rigorous comparisons across AD algorithms. Despite this contribution, these benchmarks were developed under earlier detection paradigms that prioritized only numerical features with conventional machine learning pipelines—for instance, converting categorical values into arbitrary integer codes, or discarding non-numerical fields (as shown in Figure 1). More critically, they excluded textual metadata that practitioners routinely rely on in real-world applications, *e.g.*, feature descriptions, domain context, categorical semantics, measurement units, and operational constraints. This omission is particularly problematic because the definition of an anomaly is inherently "context-dependent" (Chandola et al., 2009; Pang et al., 2021)—without understanding feature semantics and domain constraints, models may misclassify benign deviations as anomalies or fail to detect subtle but critical abnormalities.

Table 1: **Comparison of ReTabAD with Existing Resources.** Prior work can be categorized into benchmarks that mainly provide datasets for evaluation, and libraries that offer algorithm implementations and pipelines. ReTabAD uniquely combines both: raw tabular values, structured metadata, and ready-to-use pipelines for classical, deep learning, and LLM-based AD models.

| Resource Type | Coverage (number) | | Data Type | | Algorithm Type | | | Information Used | |
|---|---|---|---|---|---|---|---|---|---|
| | Datasets | Algorithms | Tabular data | Metadata | ML | DL | LLM | Statistic | Semantic |
| **Benchmark** | | | | | | | | | |
| DAMI Repository (Campos et al., 2016) | 23 | 12 | ✓ | ✗ | ✓ | ✗ | ✗ | ✓ | ✗ |
| ADBench (Han et al., 2022) | 57 | 14 | ✓ | ✗ | ✓ | ✓ | ✗ | ✓ | ✗ |
| **Library** | | | | | | | | | |
| PyOD (Zhao et al., 2019) | - | 10+ | ✓ | ✗ | ✓ | ✓ | ✗ | ✓ | ✗ |
| DeepOD (Xu et al., 2023a) | - | 9 | ✓ | ✗ | ✗ | ✓ | ✗ | ✓ | ✗ |
| **ReTabAD (ours)** | 20 | **20** | ✓ | ✓ | ✓ | ✓ | ✓ | ✓ | ✓ |

Recent advances in Large language models (LLMs) demonstrate improved capabilities for handling textual representations of numerical data, particularly in contextual reasoning (Kojima et al., 2022; Fons et al., 2024). These advances suggest new opportunities for leveraging semantic metadata in tabular AD that has been missing from traditional approaches. In particular, such textual metadata can serve as valuable domain priors—enabling models to form more plausible decision boundaries, better interpret anomalous behavior, and generalize more effectively in small-data settings. Early efforts such as AnoLLM (Tsai et al., 2025) have begun to explore this direction, revealing the potential of LLM-based approaches in tabular settings. However, their application remains constrained to utilizing only column names, due to the lack of richer textual annotations in existing benchmarks. This disconnect underscores the need for a new benchmark that restores semantic context, allowing researchers to meaningfully evaluate and advance context-aware tabular AD.

Motivated by these considerations, we introduce ReTabAD, the first *context-aware tabular AD* benchmark that incorporates rich textual metadata into AD, enabling models to ground their predictions in semantic context rather than purely numerical patterns. As highlighted in Table 1, our benchmark prioritizes quality by curating 20 tabular datasets with rich textual metadata and carefully validated anomaly definitions. This design stands in contrast to ADBench (Han et al., 2022), which achieves broader coverage by simply transforming CV/NLP datasets into tabular format—an approach that increases dataset quantity but lacks the semantic context necessary for context-aware AD. Alongside these curated datasets, ReTabAD enables comprehensive evaluation across 20 algorithms, ranging from classical and deep learning methods to the latest LLM-based approach. Furthermore, we also propose a zero-shot LLM baseline that leverages the provided metadata without any task-specific training to establish a fair starting point for context-aware AD. Empirically, incorporating textual metadata improves zero-shot LLM performance by an average of +7.6% AUROC across multiple models (see Table 4). As a result, our zero-shot LLM approach achieves performance comparable to state-of-the-art *training-based* methods, despite never being trained on task-specific data. Together, these contributions position ReTabAD as a unified and forward-looking benchmark that enables systematic study of tabular AD in both traditional and context-aware paradigms.

To sum up, our contributions are summarized as follows:

- **The first context-aware tabular AD benchmark**: We present ReTabAD, which provides 20 carefully curated tabular datasets enriched with textual metadata and faithful anomaly definitions, and enables standardized evaluation across 20 unsupervised algorithms, including classical, deep learning, and LLM-based approaches.

- **Zero-shot tabular AD framework leveraging textual metadata**: We introduce and evaluate a zero-shot LLM framework that establishes not only a strong and competitive baseline but also a new evaluation paradigm for context-aware tabular AD. By leveraging semantic context without task-specific training, ReTabAD provides a standardized reference point for future research.

- **Insights into context-aware tabular AD**: We systematically analyze how incorporating textual metadata affects not only model performance but also reasoning quality. Specifically, we evaluate alignment between predicted key features and supervised attributions, and qualitatively examine model-generated reasoning texts, providing insights into how context enhances interpretability and anomaly attribution.

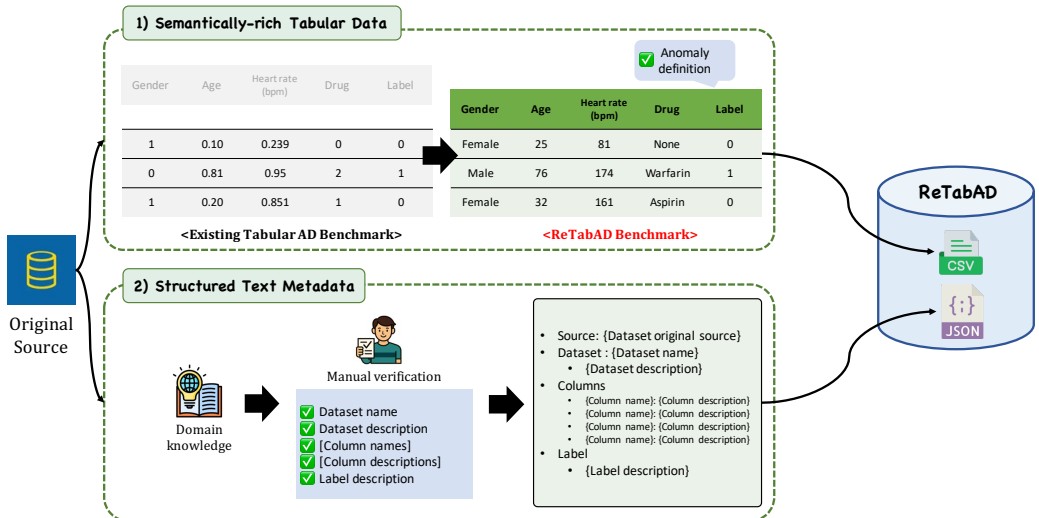

Figure 1: **Overall Data Collection and Annotation Process of ReTabAD.** We preserve semantically rich tabular values and augment them with structured textual metadata, enabling rigorous evaluation of context-aware AD.

Overall, our work emphasizes the importance of semantic information (*e.g.*, textual metadata) in AD for tabular data. We hope that this perspective, together with our new benchmark, will guide future research toward developing context-aware approaches.

## 2    LIMITATIONS ON PREVIOUS TABULAR AD BENCHMARKS

Benchmarking efforts in tabular AD have primarily aimed to provide reproducible and systematic comparisons of algorithms across diverse datasets. The comparative studies (Goldstein & Uchida, 2016; Campos et al., 2016; Ruff et al., 2021) established the first generation of evaluation resources, offering systematic comparisons of a broad range of algorithms on real-world datasets. In particular, the DAMI Repository (Campos et al., 2016) introduced semantically meaningful tabular datasets with well-defined anomaly classes, demonstrating the importance of defining anomalies in a way that reflects meaningful deviations. With the rise of deep learning, ADBench (Han et al., 2022) further advanced the field by integrating both classical and deep anomaly detection algorithms, collecting datasets from multiple domains, and analyzing them from diverse perspectives. Together with open-source libraries such as PyOD (Zhao et al., 2019) and DeepOD (Xu et al., 2023a), these resources have substantially advanced the field by enabling standardized protocols and large-scale experimental studies.

Despite these contributions, existing benchmarks lack textual metadata that makes semantic context explicit. Although anomaly classes are often well defined, the datasets are typically provided only in numerical form, and the conversion processes—such as one-hot encoding, text embedding, or dimensionality reduction—inevitably discard important semantic information. Some benchmarks also incorporate datasets from non-tabular modalities like images and speech, where feature-level semantics cannot be clearly specified. As a result, most prior research has been confined to improving performance within these numeric-only settings, focusing on better modeling of given datasets rather than leveraging the rich textual information that practitioners use in real-world anomaly detection.

## 3    ReTabAD: TABULAR AD BENCHMARK RESTORING SEMANTIC CONTEXT

To overcome these limitations as described in Section 2, we introduce ReTabAD, a next-generation benchmark that focuses exclusively on genuinely tabular datasets while systematically integrating rich semantic information. This design enables the systematic evaluation of context-aware AD.

Table 2: **Text Metadata Structure.** Example from the Cardiotocography (Campos & Bernardes, 2000) dataset in ReTabAD.

**Dataset: Cardiotocography**
**Source:** https://archive.ics.uci.edu/dataset/193/cardiotocography
**Description:** Measurements from fetal heart rate (FHR) and uterine contractions (UC) from 2126 cardiotocograms.

| Column | Type | Logical Type | Description |
|---|---|---|---|
| LB | int | numerical | FHR baseline (bpm) |
| AC | float | numerical | Accelerations per second |
| UC | float | numerical | Uterine contractions per second |
| ASTV | int | numerical | % time with abnormal short term variability |
| ⋮ | ⋮ | ⋮ | ⋮ |
| Tendency | string | categorical | Histogram tendency (Left-skewed, Symmetric) |

**Label:** Binary label (0: Normal, 1: Suspect/Pathologic)

## 3.1 DATASETS

**Data Collection and Annotation.** ReTabAD covers 20 tabular datasets drawn from widely used repositories and real-world application domains. Building upon the comprehensive coverage of ADBench (Han et al., 2022), we curate a subset of datasets and enrich them with textual metadata according to the following criteria: (1) datasets must have a clearly defined domain and ground-truth anomaly labels, (2) dataset size should be manageable for diverse models while reflecting real-world class imbalance, (3) sufficient documentation should exist to recover reliable semantic descriptions, and (4) datasets that are too easy, where performance is already saturated, should be discarded. In addition, we incorporate new datasets by inspecting their class label definitions and class-wise sample ratios, and then annotating anomalies accordingly to construct well-posed AD tasks. A complete list of the datasets and their description is summarized in Table 7 in the Appendix A and an overview of the data collection and annotation process is illustrated in Figure 1.

**Semantically-rich Tabular Data.** To incorporate rich semantic information into tabular data, we consider the following two principles:

- *Raw Numerical Preservation*: ReTabAD preserves numerical features in their original scales rather than applying unspecified normalization or standardization. Maintaining raw values retains domain-meaningful interpretations that support intuitive reasoning about anomalies. For instance, a resting heart rate of 200 bpm in an adult is directly recognizable as a critical medical anomaly, whereas its normalized counterpart only conveys statistical rarity without clinical significance.

- *Categorical Text Restoration*: Existing benchmarks often provide categorical features in inconsistent formats (*e.g.*, integer-encoded, text-embeddings) or apply inappropriate normalization that discards semantic meaning. Contrary to this practice, ReTabAD restores categorical attributes to their original textual values instead of applying such transformations. This restoration enables models, particularly language-based approaches, to reason over human-interpretable categories rather than arbitrary numerical surrogates.

**Structured Text Metadata.** We provide each dataset supplemented with a JSON-formatted metadata file, containing structured semantic descriptions omitted in previous benchmarks. To ensure reliability, we provide direct links to the original data sources, enabling researchers to verify our interpretations. The metadata is organized into three categories:

- *Dataset-Level Descriptions*: High-level descriptors, including dataset name, purpose, origin, and collection methodology. Each entry includes links to original publications or repositories to enhance credibility.

- *Column-Level Descriptions*: For each column, ReTabAD provides its name, human-readable description, logical type (*e.g.*, numerical, categorical, ordinal, binary), measurement units, if available. All descriptions are cross-referenced with the original dataset documentation.

- *Label-Level Descriptions*: For each dataset, ReTabAD clearly specifies which classes are considered normal and which are treated as anomalies. These anomaly definitions are derived directly from the original dataset documentation, with full source attribution.

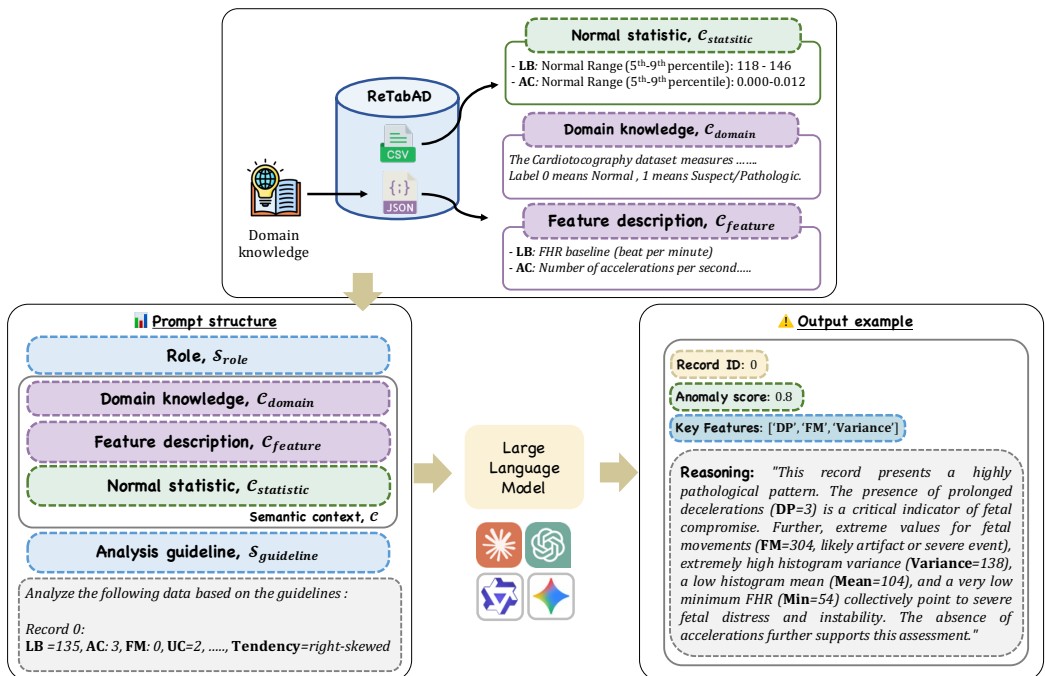

Figure 2: **Zero-shot LLM Baseline Overview.** We design prompts to evaluate the role of semantic metadata by incorporating domain knowledge ($\mathcal{C}_{domain}$), feature descriptions ($\mathcal{C}_{feature}$), and normal statistics ($\mathcal{C}_{statistic}$). The LLM generates outputs including anomaly scores, key features, and anomaly reasoning.

## 3.2 ALGORITHMS

To provide a comprehensive comparison, we evaluate 20 representative *unsupervised AD models*, with full algorithmic details provided in Appendix B. These models are grouped into three categories: (1) **Classical methods**: We include a diverse range of foundational algorithms that remain widely used. These encompass distance-based (*e.g.*, KNN (Ramaswamy et al., 2000)), density-based (*e.g.*, LOF (Breunig et al., 2000)), boundary-based (*e.g.*, OCSVM (Schölkopf et al., 1999)), and ensemble techniques (*e.g.*, Isolation Forest (Liu et al., 2008)). (2) **Deep learning methods**: We consider representative deep models designed to capture complex, non-linear representations of normality beyond classical heuristics. These include methods based on deep one-class classification (*e.g.*, DeepSVDD (Ruff et al., 2018)), reconstruction (*e.g.*, RCA (Liu et al., 2021), MCM (Yin et al., 2024)), and contrastive learning (*e.g.*, GOAD (Bergman & Hoshen, 2020), NeuTraL (Qiu et al., 2021)). (3) **LLM-based methods**: We include AnoLLM (Tsai et al., 2025), which leverages semantic information but is constrained to using only column names as context. To further leverage rich and structured metadata, we propose a new zero-shot LLM framework that directly utilizes context data as input for AD. This investigation illustrates the potential of applying LLMs to AD tasks when comprehensive semantic cues are available. Implementation details are provided in Section 3.3.

## 3.3 ZERO-SHOT LLM FRAMEWORK FOR CONTEXT-AWARE AD

**Motivation.** Despite the importance of textual metadata in real-world tabular datasets, there has been no systematic attempt to leverage such information. Conventional training-based AD models are designed to process data as numerical or categorical vectors, making it structurally difficult to directly integrate textual descriptions that contain the semantic information of features. Although there have been attempts to utilize textual metadata, most have remained limited—for instance, relying only on column names without capturing the broader context. As a consequence, such approaches fail to leverage critical signals for AD conveyed by metadata, including inter-feature relationships and domain-specific contextual cues. To address this limitation, we propose a zero-shot LLM-based AD framework that explores the potential of semantic context through context-aware reasoning over tabular data. By enabling the LLM to interpret feature semantics and relationships in natural language,

our framework shifts the focus from purely statistical patterns to high-level contextual understanding and points toward a new paradigm for context-aware AD.

**Our Design.** To utilize LLMs for tabular AD, we propose a zero-shot LLM framework as a strong baseline for context-aware AD, as illustrated in Figure 2. We employ a carefully designed prompt that integrates domain knowledge and structured reasoning guidelines. The prompt for each data instance $x_i$ consists of three key components: a **System Prompt** ($\mathcal{S}$), a **Data Input Formatter** ($\mathcal{T}(x_i)$), and a **Structured Output Query** ($\mathcal{Q}$).

1. **System Prompt** ($\mathcal{S}$): The system prompt provides the necessary context for the task. It is constructed by concatenating the following information(detailed templates are in the Appendix D.3):
   - **Role and Task Definition** ($\mathcal{S}_{\text{role}}$): Assigns the LLM an expert persona for each dataset's domain (*e.g.*, "You are a domain expert in finance").
   - **Semantic Context** ($\mathcal{C}$): Supplies essential metadata $\mathcal{M}$ about the dataset, including:
     - **Domain Knowledge** ($\mathcal{C}_{\text{domain}}$): A brief background of dataset and label information.
     - **Feature Description** ($\mathcal{C}_{\text{feature}}$): Descriptions, units, and meanings of each feature.
     - **Normal Statistic** ($\mathcal{C}_{\text{statistic}}$): Normal ranges or distributional context derived from the training normal data.
   - **Analysis Guidelines** ($\mathcal{S}_{\text{guideline}}$): Provides general principles for identifying anomalies based on the given context.
2. **Data Input Formatter** ($\mathcal{T}(x_i)$): Each data instance $x_i$ is serialized into a human-readable text format to serve as input. For example, a record is formatted as:

   "Record $i$: [feature_name_1=value_1], [feature_name_2=value_2], ...".

   This use of feature names, rather than generic indices, provides essential semantic context. For efficiency, multiple instances are processed in a single batch.
3. **Structured Output Query** ($\mathcal{Q}$): To ensure consistent and automatically parsable responses, the final part of the prompt queries the LLM to return its analysis in a structured JSON format. The required format is:

   { "anomaly_score": s, "key_features": F, "reasoning": e }

   where s is the anomaly score in $[0, 1]$, F is a list of key features, and e is the explanation in text.

## 3.4 EVALUATION

**Problem Setup.** Conventional tabular anomaly detectors evaluate each data instance $X = \{x_1, \cdots, x_n\}$, where $x \in \mathbb{R}^K$, using an anomaly scoring function $f(x)$ that determines whether the instance is normal or anomalous. ReTabAD extends this traditional setup by incorporating semantic metadata $\mathcal{M}$, which encompasses domain knowledge, feature descriptions, and contextual information about normal behavior patterns. Concretely, this work considers a metadata-aware anomaly scoring function $f(x, \mathcal{M})$, capturing how well a sample aligns with the metadata-informed notion of normality. This work analyzes how such a semantic context $\mathcal{M}$ influences anomaly detection performance.

**Hyperparameter Optimization.** We follow the default model architectures and tune key hyperparameters, including preprocessing settings. Hyperparameters for training-based methods are faithfully searched, and detailed configurations are provided in the Appendix C for reproducibility.

**Evaluator LLMs.** We evaluate a set of leading LLMs in a zero-shot AD setting to assess their general semantic reasoning capabilities. Specifically, we include state-of-the-art reasoning-oriented models—GPT-4.1 (Fachada et al., 2025), Claude-3.7-sonnet (Anthropic, 2024a), Qwen3-235B (Yang et al., 2025), and Gemini-2.5-pro (Comanici et al., 2025)—as well as a lightweight yet competitive model, GPT-4o-mini (OpenAI, 2024). For consistency and clarity, we report detailed ablation and qualitative analyses with Gemini-2.5-pro.

**Evaluation Protocol.** ReTabAD follows the *one-class classification setting*, where the training set consists of 50% of the normal samples, while the test set consists of the remaining normal samples and all anomalous samples, as in previous studies (Bergman & Hoshen, 2020; Livernoche et al., 2024). For performance evaluation, we primarily report the Area Under the Receiver Operating Characteristic Curve (AUROC), a standard metric in AD (Ruff et al., 2018). To mitigate randomness, each experiment is run five times with different seeds, and Full results with the Area Under the

Table 3: **AUROC Performance on the ReTabAD.** This table compares training-based methods and our zero-shot LLM (Gemini-2.5-pro). *Full Desc* corresponds to the original prompt setting (Type D), and *No Desc* reports results without metadata descriptions (Type A).

| Dataset | Training-based Methods | | | | | | | | | | Zero-shot LLM | |
|---|---|---|---|---|---|---|---|---|---|---|---|---|
| | IForest | OCSVM | LOF | DeepSVDD | NeuTraL | SLAD | DIF | MCM | DRL | AnoLLM | *No Desc* | *Full Desc* |
| automobile | 0.648 | 0.645 | 0.688 | 0.845 | 0.810 | 0.780 | 0.647 | 0.801 | 0.762 | 0.625 | 0.806 | 0.767 |
| backdoor | 0.888 | 0.876 | 0.702 | 0.800 | 0.948 | 0.935 | 0.973 | 0.961 | 0.902 | 0.885 | 0.804 | 0.827 |
| campaign | 0.732 | 0.700 | 0.669 | 0.721 | 0.760 | 0.717 | 0.677 | 0.761 | 0.737 | 0.738 | 0.596 | 0.813 |
| cardiotocography | 0.830 | 0.868 | 0.826 | 0.813 | 0.791 | 0.764 | 0.740 | 0.843 | 0.847 | 0.735 | 0.643 | 0.829 |
| census | 0.620 | 0.647 | 0.596 | 0.712 | 0.686 | 0.687 | 0.726 | 0.729 | 0.728 | 0.737 | 0.662 | 0.895 |
| churn | 0.506 | 0.651 | 0.465 | 0.573 | 0.651 | 0.571 | 0.494 | 0.580 | 0.553 | 0.557 | 0.499 | 0.809 |
| cirrhosis | 0.849 | 0.823 | 0.843 | 0.807 | 0.831 | 0.810 | 0.820 | 0.830 | 0.814 | 0.850 | 0.694 | 0.850 |
| covertype | 0.968 | 0.998 | 0.980 | 0.971 | 0.998 | 0.999 | 0.998 | 0.998 | 0.995 | 0.990 | 0.861 | 0.961 |
| credit | 0.639 | 0.717 | 0.590 | 0.638 | 0.690 | 0.691 | 0.689 | 0.671 | 0.680 | 0.495 | 0.501 | 0.681 |
| equip | 0.980 | 0.987 | 0.987 | 0.984 | 0.985 | 0.981 | 0.987 | 0.986 | 0.984 | 0.986 | 0.975 | 0.978 |
| gallstone | 0.650 | 0.671 | 0.704 | 0.731 | 0.780 | 0.758 | 0.649 | 0.757 | 0.745 | 0.574 | 0.615 | 0.667 |
| glass | 0.944 | 0.959 | 0.974 | 0.963 | 0.968 | 0.952 | 0.936 | 0.966 | 0.949 | 0.907 | 0.832 | 0.919 |
| glioma | 0.722 | 0.725 | 0.711 | 0.766 | 0.784 | 0.886 | 0.686 | 0.790 | 0.764 | 0.708 | 0.586 | 0.895 |
| quasar | 0.915 | 0.930 | 0.966 | 0.804 | 0.954 | 0.952 | 0.793 | 0.950 | 0.897 | 0.957 | 0.816 | 0.961 |
| seismic | 0.710 | 0.710 | 0.615 | 0.729 | 0.716 | 0.727 | 0.736 | 0.716 | 0.714 | 0.746 | 0.646 | 0.741 |
| stroke | 0.679 | 0.716 | 0.625 | 0.722 | 0.703 | 0.678 | 0.705 | 0.753 | 0.727 | 0.687 | 0.562 | 0.776 |
| vertebral | 0.500 | 0.638 | 0.540 | 0.581 | 0.697 | 0.679 | 0.340 | 0.615 | 0.477 | 0.597 | 0.461 | 0.755 |
| wbc | 0.961 | 0.969 | 0.960 | 0.969 | 0.785 | 0.938 | 0.679 | 0.957 | 0.954 | 0.871 | 0.861 | 0.968 |
| wine | 0.987 | 0.957 | 0.974 | 0.974 | 0.988 | 0.980 | 0.760 | 0.979 | 0.958 | 0.933 | 0.663 | 0.991 |
| yeast | 0.838 | 0.875 | 0.803 | 0.820 | 0.841 | 0.846 | 0.815 | 0.856 | 0.841 | 0.809 | 0.731 | 0.861 |
| Average Rank | 7.50 | 5.35 | 7.70 | 6.40 | 4.30 | 5.50 | 7.85 | **3.95** | 6.30 | 7.35 | 10.80 | **4.10** |
| Average AUROC | 0.778 | 0.803 | 0.761 | 0.796 | 0.818 | 0.817 | 0.742 | 0.825 | 0.801 | 0.769 | 0.691 | **0.847** |

Precision-Recall Curve (AUPRC) (Saito & Rehmsmeier, 2015) and standard deviations are in the Appendix E, F.

## 4    EXPERIMENTS ON RETABAD BENCHMARK

This section explores the potential of context-aware approaches in tabular AD. Specifically, we design our experiments to investigate the following research questions:

- Does semantic context $\mathcal{M}$ improve AD performance? (§4.1)
- Which types of semantic information contribute most to detection performance? (§4.2)
- Can LLMs leverage domain-specific knowledge to identify specialized anomaly patterns? (§4.3)

### 4.1    CONTEXT-AWARE AD PERFORMANCE

Table 3 presents a detailed comparison across 20 datasets. We report representative AD methods in the main table; the complete per-dataset results for all evaluated models are provided in Appendix F. Despite requiring no training, our zero-shot LLM framework with textual metadata achieves an average rank of 4.08, approaching the state-of-the-art training-based method MCM (Yin et al., 2024), demonstrating that semantic context can serve as a strong signal for AD. Notably, the benefit of semantic context is especially pronounced in datasets such as glioma, where 20 out of 21 features are categorical and primarily expressed as binary (yes/no) attributes. In such settings, raw tabular values (*e.g.*, ATRX=1, IDH1=0) carry little meaning without domain-specific descriptions. By leveraging column-level metadata (*e.g.*, "ATRX mutation present"), the LLM can better interpret the clinical implications of these attributes, leading to substantial performance gain. This highlights how semantic context is crucial when features are categorical or domain-specific, providing not only improved detection accuracy but also guidance for developing future AD models that are semantically grounded.

We quantify the overall effect of incorporating textual metadata on zero-shot LLMs. As summarized in Table 4, enriching the tabular input with semantic context consistently yields substantial gains across all evaluator models, where *No Desc.* refers to the setting that provides only normal statistics without metadata and *Full Desc.* includes all available textual information. On average, we observe a +7.6 percentage point improvement in AUROC over the *No Desc.* baseline, indicating that models are

Table 4: **Effect of Textual Metadata on Zero-shot LLMs.** We report average AUROC across 20 datasets of ReTabAD.

| Evaluator LLM | *No Desc.* | *Full Desc.* |
|---|---|---|
| GPT-4o-mini | 0.692 | **0.726** |
| GPT-4.1 | 0.696 | **0.735** |
| Claude-3.7-sonnet | 0.725 | **0.777** |
| Qwen3-235B | 0.665 | **0.747** |
| Gemini-2.5-pro | 0.691 | **0.847** |

Table 5: **Win Rate Comparison across Different Context Types on the ReTabAD.** Win rates are computed based on AUROC performance, where each type represents different combinations of contextual information: Domain knowledge, Feature description, and Normal statistics.

| Type | Context Information | | | Win Rate | | | | |
|---|---|---|---|---|---|---|---|---|
| | $\mathcal{C}_{statistic}$ | $\mathcal{C}_{feature}$ | $\mathcal{C}_{domain}$ | GPT-4o-mini | GPT-4.1 | Claude-3.7-Sonnet | Qwen3-235B | Gemini-2.5-pro |
| A (*No Desc.*) | ✓ | ✗ | ✗ | 0.15 | 0.20 | 0.00 | 0.05 | 0.05 |
| B | ✓ | ✓ | ✗ | 0.05 | 0.15 | 0.20 | 0.20 | 0.00 |
| C | ✗ | ✓ | ✓ | 0.25 | 0.30 | 0.10 | 0.10 | 0.15 |
| D (*Full Desc.*) | ✓ | ✓ | ✓ | **0.55** | **0.35** | **0.70** | **0.65** | **0.80** |

better able to ground their predictions when provided with feature descriptions and dataset semantics. Interestingly, the performance gap is even more pronounced for reasoning-oriented LLMs such as Qwen3-235B and Gemini-2.5-pro, which show the largest relative gains among all models. This suggests that these models are particularly capable of leveraging structured textual cues to form more contextually grounded anomaly judgments, whereas lighter models still benefit but to a lesser extent.

## 4.2 ABLATION STUDY ON CONTEXTUAL INFORMATION

To evaluate the relative contribution of different information sources, we design an ablation study with four prompt variants (Type A–D) in our ReTabAD benchmark as follows:

- **Type A** (No Desc.): Normal statistic only (anonymized column names *e.g.*, AA, AB, ...)
- **Type B**: Normal statistic + Feature description
- **Type C**: Feature description + Domain knowledge
- **Type D** (Full Desc.): Normal statistic + Feature description + Domain knowledge

The ablation results (Table 5) clearly demonstrate that combining all three sources of semantic information yields the highest win rate across all LLMs. Column descriptions generally improve performance when added to normal statistics (Type A $\rightarrow$ B) (see Table 11 in the Appendix E), while domain descriptions alone (Type C) produce mixed results—highlighting the necessity of numerical context for stable reasoning. Type D exhibits strong synergistic effects, indicating that LLMs benefit from both statistical grounding and domain-level priors to accurately detect anomalies. Overall, these findings show that even without domain knowledge, the inclusion of feature descriptions already leads to clear performance gains, and that semantic information (both domain and feature) becomes substantially more powerful when coupled with statistical context.

## 4.3 FEATURE ATTRIBUTION AND REASONING ANALYSIS

**Key Feature Alignment with Supervised Attributions.** We design a quantitative evaluation to measure whether LLMs identify features that truly drive anomaly classification, focusing exclusively on *anomalous instances*. Normal instances are not evaluated individually, as they are primarily used to establish the decision boundary and represent expected behavior. Hence, our metric specifically measures whether the model's reasoning highlights features that explain *why* a sample is anomalous. Specifically, we train an XGBoost (Chen & Guestrin, 2016) on ground-truth labels and compute SHAP (Lundberg & Lee, 2017) values for each instance. For a given anomalous instance $x_i$, we define the reference set $R_i^{(K)}$ as the top-$K$ features ranked by absolute SHAP values, and compare it with the LLM's predicted top-$K$ features $\hat{F}_i^{(K)}$ using F1@K, which serves as a *proxy metric* for domain-informed reasoning. Table 6 illustrates representative datasets where textual metadata provides clear benefits, while full results across all 20 datasets are reported in the Appendix E. These examples show that semantic context can help LLMs

Table 6: **Impact of Textual Metadata on Reasoning Alignment.** We report F1-Score@K ($K = 1, 3$) on anomalous instances for Type A (No Desc.) and Type D (Full Desc.) using Gemini-2.5-Pro.

| Dataset | F1@1 | | F1@3 | |
|---|---|---|---|---|
| | Type A | Type D | Type A | Type D |
| campaign | 0.086 | **0.344** | 0.159 | **0.466** |
| churn | 0.133 | **0.288** | 0.160 | **0.400** |
| glioma | 0.009 | **0.551** | 0.044 | **0.589** |

recover the features that drive anomalous behavior. This improvement is especially pronounced in `glioma` and `campaign`, where key biomarkers (`IDH1`, `TP53`) and key determinants of campaign success (`poutcome` (outcome of the previous campaign), `duration` (length of the last contact)) are consistently recovered, yielding large F1 gains. These results demonstrate that textual metadata alone enables LLMs to account for much of the explanatory signal behind anomalies, allowing them to identify the key features that drive abnormal behavior without any task-specific training.

**Qualitative Analysis of Reasoning Text.** To quantitatively explore reasoning quality, we measure detection AUROC improvement when reasoning texts of detected anomalies are provided as contextual examples in the prompt, prioritizing samples with higher anomaly scores. As shown in Figure 3, we observe that performance initially fluctuates with very few examples, improves steadily up to five examples, and then saturates, confirming that our evaluation captures meaningful variations in reasoning quality. This suggests that the quality of reasoning can meaningfully affect detection outcomes. Importantly, the effect is more pronounced when comparing against Type A explanations: higher-quality reasoning (Type D) offers richer, domain-relevant insights and leads to greater performance gains.

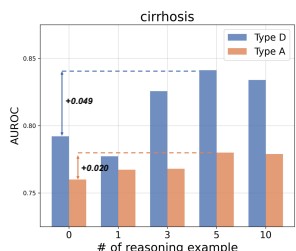

Figure 3: **Quantitative Evaluation of LLM Reasoning via Performance Gain.** Results are reported on the `cirrhosis` dataset using Gemini-2.5-Pro.

To illustrate further, Figure 4 presents an example from the `cirrhosis` dataset. Type A explanation merely states a numeric deviation, offering little insight into why the sample is anomalous. In contrast, Type D explanation highlights *"The elevated Prothrombin time (11.50) indicates compromised liver synthetic function"*. Importantly, the notion of liver synthetic function is not provided in the metadata itself; rather, it emerges as an intermediate diagnostic step derived from the model's prior knowledge. This shows that, when enriched with textual metadata, the LLM not only grounds its reasoning in the observed feature but also extends beyond the given descriptions to provide domain meaningful interpretation. This contrast demonstrates how textual metadata can elevate shallow observations into domain-aware explanations that incorporate both provided context and external knowledge.

---

**Cirrhosis.json**

**Metadata:**

```
"dataset_name" : "cirrhosis",
"description": "The dataset includes clinical features for predicting survival
states of patients with liver cirrhosis.",
"columns": [
{
"name": "Prothrombin",
"dtype": "float",
"logical_type": "numerical",
"description": "Prothrombin time in seconds"
},
]
...
"label_description": "In this dataset, the normal/abnormal classification is
based on patient survival status. "
```

**Type A** (w/ $c_{statistic}$)

**GT label:** 1 (Anomaly)
**Anomaly score:** 0.6
**Reasoning:** *Feature AP* is above the normal range.

**Type D** (w/ $c_{statistic}$, $c_{feature}$, $c_{domain}$)

**GT label:** 1 (Anomaly)
**Anomaly score:** 0.68
**Reasoning:** The elevated *Prothrombin time* (11.50) is the key anomaly, indicating *compromised liver synthetic function*. While other markers are currently stable, impaired coagulation is a direct measure of liver failure and a significant predictor of *mortality*, warranting an abnormal classification.

Figure 4: **Reasoning Text Examples on the `cirrhosis` dataset.** Left: JSON example. Right: Comparison of Type A (numeric deviation only) and Type D (domain grounded), generated using Gemini-2.5-Pro.

## 5 CONCLUSION

We present ReTabAD, a benchmark that restores textual semantics for context-aware tabular AD. By pairing raw tabular data with textual metadata, ReTabAD enables models not only to detect anomalies but to reason about why they occur. Our results show that semantic context improves performance and interpretability, allowing zero-shot LLMs to approach state-of-the-art training-based detectors. Beyond benchmarking, ReTabAD can serve as a foundation for diverse applications

including semantic-guided data augmentation and hybrid approaches that combine statistical and semantic cues. We hope this resource accelerates research toward context-aware, interpretable anomaly detection that is robust and deployable in real-world systems.

## ETHICS STATEMENT

This work utilizes several publicly available healthcare datasets (e.g., Cardiotocography, Cirrhosis, Glioma) that contain sensitive clinical information. We emphasize that all experiments are conducted strictly for research purposes and do not involve direct patient interaction. While our benchmark and methods are designed to advance anomaly detection research, false alarms in healthcare applications may lead to unnecessary interventions, patient anxiety, or resource misallocation. Therefore, the models and results reported here should not be interpreted as ready for clinical deployment. Any practical adoption in healthcare settings must be accompanied by rigorous validation, ethical review, and close collaboration with medical professionals.

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

# A    ReTabAD Dataset Details

ReTabAD benchmark consists of 20 datasets, each accompanied by a JSON text metadata file. This comprehensive collection encompasses a wide range of domains and data characteristics, as summarized in Table 7. Our benchmark covers a broad set of application domains, including healthcare (7 datasets: cardiotocography, cirrhosis, gallstone, glioma, stroke, vertebral, wbc), finance (2 datasets: campaign, credit), cybersecurity/network security (backdoor), and diverse scientific and real-world domains such as biology (yeast), astronomy (quasar), geophysics (seismic), chemistry (wine), forensics (glass), environment (covertype), demographics (census), telecommunications (churn), and manufacturing (automobile, equip). This broad domain representation is particularly valuable for evaluating context-aware tabular anomaly detection, as it enables assessment of whether algorithms can leverage domain-specific knowledge to improve detection performance. The diversity allows researchers to investigate how contextual understanding of different application areas—such as medical diagnostic patterns in healthcare datasets versus fraud detection patterns in financial datasets—can enhance anomaly detection capabilities beyond traditional domain-agnostic approaches.

## A.1    Data Collection

The datasets were carefully curated by selecting genuine tabular datasets from ADBench (Han et al., 2022) and tracing back to their original sources, with the majority sourced from the UCI Machine Learning Repository[1]. By accessing the original data sources, we identified and corrected erroneous preprocessing steps present in ADBench and verified the appropriate definitions of anomalies for each dataset. This rigorous validation process resulted in datasets that differ from their ADBench counterparts, ensuring higher quality and more reliable anomaly labels. We additionally filtered out datasets lacking sufficient metadata, with performance saturation across methods (defined as AUROC/AUPRC $\geq 0.99$ for most models), or having small sample sizes ($< 100$ samples) to maintain benchmark discriminative power. We further enriched our collection by incorporating additional high-quality datasets from Kaggle[2] to ensure broader domain coverage. We provide both raw data and preprocessing code to ensure reproducibility and transparency. Complete source information, dataset access links, and preprocessing scripts are available in our benchmark repository, and all datasets are publicly accessible.

Table 7: Dataset Statistics with Domain Information

| Dataset Name | Domain | Datapoints | Columns | Normal Count | Anomaly Count | Anomaly Ratio (%) |
|---|---|---|---|---|---|---|
| automobile | Manufacturing | 159 | 25 | 117 | 42 | 26.42 |
| backdoor | Network | 50,000 | 42 | 48,779 | 1,221 | 2.44 |
| campaign | Finance | 7,842 | 16 | 6,056 | 1,786 | 22.77 |
| cardiotocography | Healthcare | 2,126 | 21 | 1,655 | 471 | 22.15 |
| census | Demographics | 50,000 | 41 | 47,121 | 2,879 | 5.76 |
| churn | Telecommunications | 7,032 | 19 | 5,163 | 1,869 | 26.6 |
| cirrhosis | Healthcare | 247 | 17 | 165 | 82 | 33.20 |
| covertype | Environment | 50,000 | 12 | 49,520 | 480 | 0.96 |
| credit | Finance | 30,000 | 23 | 23,364 | 6,636 | 22.12 |
| equip | Manufacturing | 7,672 | 6 | 6,905 | 767 | 10.00 |
| gallstone | Healthcare | 241 | 38 | 161 | 80 | 33.20 |
| glass | Forensics | 214 | 9 | 163 | 51 | 23.83 |
| glioma | Healthcare | 730 | 23 | 487 | 243 | 33.29 |
| quasar | Astronomy | 50,000 | 8 | 40,520 | 9,480 | 18.96 |
| seismic | Geophysics | 2,584 | 18 | 2,414 | 170 | 6.58 |
| stroke | Healthcare | 4,909 | 10 | 4,700 | 209 | 4.26 |
| vertebral | Healthcare | 310 | 6 | 210 | 100 | 32.26 |
| wbc | Healthcare | 535 | 30 | 357 | 178 | 33.27 |
| wine | Chemistry | 178 | 13 | 130 | 48 | 26.97 |
| yeast | Biology | 1,484 | 8 | 1,389 | 95 | 6.40 |

## A.2    Tabular Data Preprocessing

To ensure fair and consistent evaluation across models, we apply a standardized preprocessing pipeline to all datasets. We provide both raw data and comprehensive preprocessing code for each dataset to

---

[1] https://archive.ics.uci.edu/

[2] https://www.kaggle.com/

guarantee reproducibility and transparency in our benchmark. This systematic approach addresses common inconsistencies found in existing benchmarks and ensures that all datasets maintain their inherent characteristics while providing a unified foundation for algorithm comparison. The main preprocessing steps are as follows:

- **Dataset Size Standardization:**
    - Datasets exceeding 50,000 samples are randomly downsampled to 50,000 instances using `np.random.choice` with seed 42 for reproducibility and also provide csv file.

- **Anomaly Definition Verification:**
    - For datasets previously available in processed benchmarks (*e.g.*, ADBench), anomaly definitions are verified against the original dataset documentation to ensure correct semantic interpretation. Ambiguous or incorrectly defined anomaly classes are corrected based on domain knowledge and the specifications provided by the dataset creators.
    - For newly incorporated datasets, we exclusively use public *classification* datasets from authoritative repositories such as the UCI Machine Learning Repository or Kaggle. Anomaly labels are deterministically derived by remapping minority or domain-defined abnormal classes to the anomaly category according to the original ground-truth class taxonomy, without introducing any subjective annotation.
    - Complete anomaly label definitions for all datasets included in our benchmark are provided in Table 8 to ensure full transparency and reproducibility.

- **Anomaly Ratio Balancing:**
    - Anomaly class proportions are capped at a maximum of 1/3 of the total dataset.

- **Missing Value Handling:**
    - Any instance containing missing values in input features or target labels is completely removed.
    - No imputation is applied to avoid introducing artificial patterns or bias.

- **Feature Value Restoration:**
    - Categorical features are strictly identified and restored based on original metadata from the UCI Machine Learning Repository and other data sources.
    - Original categorical values are recovered from encoded representations to preserve semantic meaning.
    - Numerical features are restored to their original scales by referencing source documentation, as normalization criteria applied in existing benchmarks were often unclear or undocumented.
    - Each column is assigned column names based on original dataset documentation.

Table 8: Dataset Information Summary

| Dataset Name | Description | Label Definition |
|---|---|---|
| automobile | Car specifications and insurance risk ratings from the 1985 Auto Imports Database. | High-risk cars with symboling +2 or +3 are considered anomalies, while normal-risk cars with other symboling values are normal. |
| backdoor | Network traffic data for detecting backdoor attacks. | Backdoor attacks are considered anomalies, while normal network traffic is normal. |
| campaign | Bank marketing data with client demographics and outcomes. | Clients who subscribed to term deposits are considered anomalies, while those who did not subscribe are normal. |
| cardiotocography | Fetal heart rate and uterine contraction measurements from CTGs. | Suspect or Pathologic fetal states are considered anomalies, while Normal states are normal. |
| census | U.S. Census data with demographic and socioeconomic attributes. | High-income individuals (>\$50K) are considered anomalies, while low-income individuals (<=\$50K) are normal. |
| churn | Telco customer data with demographics, service usage, and account details. | Customers who churned are considered anomalies, while customers who stayed are normal. |
| cirrhosis | Mayo Clinic study data on primary biliary cirrhosis patients. | Patients who died are considered anomalies, while censored patients or those who survived are normal. |
| covertype | Forest cover type data with ecological and cartographic variables. | Cottonwood/Willow forests (low elevation deciduous riparian) are considered anomalies, while Lodgepole Pine forests (mid to high elevation coniferous) are normal. |
| credit | Credit card dataset with demographics and payment history. | Customers who defaulted on their credit card payment are considered anomalies, while customers who did not default are normal. |
| equip | Industrial equipment sensor data for fault detection. | Faulty equipment states are considered anomalies, while normal operation states are normal. |
| gallstone | Clinical data of patients with gallstone disease including lab results. | Patients with gallstones present are considered anomalies, while patients without gallstones are normal. |
| glass | Forensic glass dataset with oxide content measurements for type identification. | Architectural window glass (classes 1–3) is treated as normal, while container, tableware, and headlamp glass (classes 5–7) are treated as anomalies. |
| glioma | Clinical and genetic data of glioma patients with tumor grades. | Glioblastoma Multiforme (GBM) cases are considered anomalies, while Lower-Grade Glioma (LGG) cases are normal. |
| quasar | SDSS dataset with spectral features of stars, galaxies, and quasars. | Quasars (QSO) are considered anomalies as rare astronomical objects, while stars and galaxies are normal common objects. |
| seismic | Mining dataset of seismic bumps for hazardous event prediction. | Hazardous states with high energy seismic bumps are considered anomalies, while non-hazardous states are normal. |
| stroke | Patient data with demographics, health risks, and stroke outcomes. | Patients who had a stroke are considered anomalies, while patients who did not have a stroke are normal. |
| vertebral | Biomechanical features from spinal X-rays of vertebral conditions. | Original 'Normal' class samples are considered anomalies, while pathological classes (Hernia and Spondylolisthesis) are normal. |
| wbc | Breast cancer dataset from digitized cell nuclei features. | Malignant breast cancer cases are considered anomalies, while benign cases are normal. |
| wine | Chemical analysis of Italian wines from three cultivars. | Wine cultivar 3 is considered an anomaly as it differs from the other two typical cultivars. |
| yeast | Protein features for predicting subcellular localization in yeast. | Proteins localized to endoplasmic reticulum membrane (ME1/ME2) are considered anomalies due to rare localization. |

A.3 METADATA FORMAT

We newly establish structured metadata to provide comprehensive dataset information for evaluating context-aware tabular AD. The metadata follows a standardized format designed for extensibility to accommodate future datasets and evolving research needs. This structured approach enables seamless integration of new datasets while maintaining consistency across the benchmark. Table 2 illustrates the metadata structure using the Cardiotocography dataset as an example from the ReTabAD benchmark.

- **Dataset Description:** The metadata begins with essential dataset identifiers including the dataset name, original source URL, and a concise description of the data collection context. This information provides the foundational context necessary for domain information.
- **Feature Description:** Each feature is systematically documented with four key attributes: (1) *Feature name* - the original column identifier, (2) *Type* - the raw data type (int, float, string), (3) *Logical type* - the semantic classification (numerical, categorical, ordinal, binary), and (4) *Description* - a detailed explanation of the feature's meaning and measurement units. This multi-layered typing system enables our prompt generator to apply appropriate analysis strategies for different feature characteristics.
- **Label Description:** The metadata explicitly defines the anomaly detection target, specifying the label column name, data type, and semantic interpretation. For binary classification tasks, the mapping between numeric values (0/1) and semantic labels (Normal/Anomaly) is clearly established to ensure consistent interpretation across different prompt types.

This structured metadata format enables our system to automatically generate contextually appropriate prompts while maintaining consistency across diverse datasets. The standardized schema supports both domain-specific analysis (using feature descriptions and dataset context) and domain-agnostic analysis (using only statistical properties and logical types), facilitating comprehensive evaluation of contextual information impact on anomaly detection performance. Additionally, our metadata-driven approach can automatically recommend appropriate data preprocessing strategies based on logical types—for instance, suggesting standardization for numerical features, one-hot encoding for categorical features, and ordinal encoding for ordinal features—even when provided with only raw tabular data, as demonstrated in Table 9.

Table 9: Recommended preprocessing methods for each `logical_type`.

| logical_type | Details |
|---|---|
| numerical | **Meaning:** Continuous or discrete numeric values
**Recommended Preprocessing:** StandardScaler (z-score), MinMaxScaler, log transform (if skewed), clipping or winsorization
**Notes:** Improves stability in statistical and deep models |
| categorical | **Meaning:** Unordered categories
**Recommended Preprocessing:** One-hot encoding (low cardinality), learned embedding (high cardinality), label encoding (for trees)
**Notes:** Prefer embedding for many categories |
| ordinal | **Meaning:** Ordered categories
**Recommended Preprocessing:** Ordinal encoding (e.g., Low=0, Medium=1, High=2)
**Notes:** Numeric mapping should preserve order |
| binary | **Meaning:** Two distinct values
**Recommended Preprocessing:** 0/1 encoding
**Notes:** Single column usually sufficient |

## B ReTabAD ALGORITHM DETAILS

Our benchmark covers a comprehensive set of unsupervised anomaly detection algorithms. For methods that are not available in existing libraries, we directly implemented them based on the original sources to ensure their faithful integration into our benchmark. In addition, we include all implementations available in widely used libraries (PyOD[3] and DeepOD[4]). To ensure consistency, we standardize every algorithm to follow a unified `fit`/`predict` interface in the scikit-learn style,

---

[3]https://pyod.readthedocs.io/
[4]https://deepod.readthedocs.io/en/latest/

enabling seamless comparison across methods. Furthermore, we expose the core hyperparameters of each algorithm, making them tunable within our framework so that hyperparameter optimization (HPO) can be conducted fairly and reproducibly across all categories of models.

## B.1 CLASSICAL METHODS

**IForest.** Isolation Forest (IForest) (Liu et al., 2008) detects anomalies by recursively partitioning the feature space using random decision trees. Since anomalies are typically easier to isolate than normal instances, they yield shorter average path lengths in the trees. This property enables efficient detection with linear time complexity, making IForest suitable for high-dimensional datasets.

**KNN.** The $k$-Nearest Neighbors (KNN) method (Ramaswamy et al., 2000) identifies anomalies based on the distance between a data point and its neighbors. Anomalous points tend to reside far from dense clusters of normal samples, leading to larger distance-based scores. Variants such as average or median distance to the $k$ neighbors are commonly used to improve robustness.

**OCSVM.** One-Class Support Vector Machine (OCSVM) (Schölkopf et al., 2001) learns a decision boundary that encloses the majority of the data while maximizing the margin in a high-dimensional feature space. Samples falling outside this boundary are flagged as anomalies. Despite its sensitivity to kernel and parameter choices, OCSVM remains a widely adopted baseline for one-class anomaly detection.

**LOF.** The Local Outlier Factor (LOF) (Breunig et al., 2000) algorithm measures the local deviation of a point's density relative to its neighbors. Instances with significantly lower local density are considered anomalous. LOF is effective in detecting context-dependent anomalies but can be sensitive to neighborhood size and parameter selection.

**PCA.** Principal Component Analysis (PCA) for anomaly detection (Shyu et al., 2003) projects data into a lower-dimensional subspace that captures the majority of variance. Reconstruction error or the residual in the discarded components is then used as the anomaly score. This approach is effective when anomalies do not conform to the dominant low-dimensional structure of normal data.

## B.2 DEEP LEARNING METHODS

**DeepSVDD.** Deep Support Vector Data Description (DeepSVDD) (Ruff et al., 2018) extends the classic one-class classification paradigm to deep neural networks. It learns a feature representation that maps normal samples into a compact hypersphere in the latent space, while anomalies lie farther away. This method is a widely used baseline for deep one-class anomaly detection.

**REPEN.** REPEN (Pang et al., 2018) proposes a random distance-based framework for anomaly detection in ultrahigh-dimensional data. By learning representations that preserve relative distances between randomly sampled triplets, the method improves the effectiveness of distance-based outlier detectors applied on the learned embedding space.

**RDP.** Random Distance Prediction (RDP) (Wang et al., 2020) trains a Siamese neural network to predict distances in a randomly projected space, learning representations that preserve data proximity. These learned embeddings significantly improve anomaly detection and clustering performance without using any labels.

**RCA.** The RCA method (Liu et al., 2021) introduces a collaborative autoencoder framework where multiple autoencoders jointly learn reconstructions. Anomalous samples tend to disrupt collaboration, yielding higher reconstruction errors. This collaborative structure enhances robustness against complex anomaly patterns.

**GOAD.** GOAD (Bergman & Hoshen, 2020) frames anomaly detection as a self-supervised classification task. By applying diverse geometric transformations to data and predicting transformation labels, the model learns discriminative representations of normality. Anomalies fail to conform to the learned classification boundaries, enabling detection.

**NeuTraL.** NeuTraL (Qiu et al., 2021) introduces neural transformation learning, where input samples are transformed via trainable mappings. The model learns to distinguish normality based on consistency across multiple transformations, making it applicable to non-image domains such as tabular data.

**ICL.** Internal Contrastive Learning (ICL) (Shenkar & Wolf, 2022) proposes a contrastive framework specifically designed for tabular anomaly detection. By constructing positive and negative pairs from within each sample's feature space, ICL learns discriminative feature representations that enhance anomaly separability.

**DIF.** Deep Isolation Forest (DIF) (Xu et al., 2023a) integrates the principles of Isolation Forest with deep representation learning. The method first learns embeddings that capture salient data characteristics, followed by isolation-based partitioning to detect anomalies more effectively in high-dimensional settings.

**SLAD.** Scale Learning for Anomaly Detection (SLAD) (Xu et al., 2023b) introduces auxiliary supervisory signals derived from scale consistency. By enforcing scale-invariant representations, SLAD enhances the generalization capability of deep anomaly detection models, especially for complex tabular datasets.

**MCM.** Masked Cell Modeling (MCM) (Yin et al., 2024) learns feature correlations by reconstructing randomly masked cells with a trainable mask generator. Anomalies are detected via high reconstruction errors, and a diversity loss ensures complementary masks.

**NPT-AD.** NPT-AD (Thimonier et al., 2024) applies Non-Parametric Transformers (NPT) to model both feature–feature and sample–sample dependencies. By reconstructing masked features of normal samples, anomalies are detected via higher reconstruction errors, achieving strong performance on tabular benchmarks.

**DRL.** Decomposed Representation Learning (DRL) (Ye et al., 2025a) remaps tabular data into a constrained latent space by decomposing each normal sample into a weighted combination of randomly generated orthogonal basis vectors. A novel discrepancy constraint further amplifies the separation between normal and anomalous patterns, resulting in strong detection performance across diverse tabular benchmarks.

**Disent-AD.** Disent-AD (Ye et al., 2025b) disentangles tabular attributes into two distinct but correlated subsets using a two-head self-attention module. The model reconstructs the original sample from each subset to capture intrinsic correlations of normal data, enabling robust one-class anomaly detection.

### B.3 LLM BASED METHODS

**AnoLLM.** AnoLLM (Tsai et al., 2025) employs large language models for unsupervised tabular anomaly detection by serializing mixed-type rows into text and fine-tuning an LLM to model the data distribution. Anomaly scores are derived from negative log-likelihoods, offering strong performance on mixed-feature benchmarks and competitive results on predominantly numerical datasets.

**Zero-shot LLM.** We additionally introduce a zero-shot baseline that leverages large language models for tabular anomaly detection. Each data row is serialized into a textual prompt together with semantic metadata, and the LLM assigns anomaly scores based on its contextual reasoning without any fine-tuning. This setting highlights the potential of LLMs as strong baselines for context-aware tabular anomaly detection.

## C   Experimental Details of Baseline Methods

### C.1   Hyperparameter settings

Table 10 summarizes the hyperparameter search ranges for each model. By default, each configuration searches over scaling_type $\in \{standard, minmax\}$ and cat_encoding $\in \{int, onehot\}$.

| Model | Default Parameters | Hyperparameter Search Range |
|---|---|---|
| OCSVM | kernel: rbf
contamination: 0.1 | nu: {0.1, 0.2, 0.3, 0.4, 0.5}
contamination: {0.1, 0.2, 0.3} |
| LOF | novelty: True
metric: euclidean
p: 2 | n_neighbors: {10, 20, 30, 50}
contamination: {0.01, 0.05, 0.1}
leaf_size: {10, 30, 50} |
| KNN | leaf_size: 30
metric: minkowski
p: 2 | n_neighbors: {10, 20, 30, 50}
contamination: {0.01, 0.05, 0.1} |
| IForest | max_features: 1.0
bootstrap: False | n_estimators: {50, 100, 200, 300, 500}
contamination: {0.05, 0.1, 0.15, 0.2, 0.25, 0.3} |
| PCA | svd_solver: auto
weighted: True
n_selected_components: null | n_components: {null, 2, 3, 5, 10, 15, 20, 25, 30}
contamination: {0.1, 0.2, 0.3}
whiten: {True, False} |
| DeepSVDD | bias: False | lr: {0.005, 0.001, 0.0005, 0.0001}
rep_dim: {32, 64, 128}
hidden_dims: {100,50; 128,64; 256,128; 512,256,128} |
| REPEN | act: ReLU
init_score_ensemble_size: 50
init_score_subsample_size: 8 | lr: {0.001, 0.0001}
rep_dim: {128, 256}
hidden_dims: {100,50; 256,128} |
| RDP | rep_dim: 128
hidden_dims: 100,50 | lr: {0.001, 0.0001}
hidden_dims: {100,50; 256,128}
rep_dim: {32, 128} |
| RCA | act: LeakyReLU
alpha: 0.5
inference_ensemble: 10 | lr: {0.005, 0.001, 0.0005, 0.0001}
rep_dim: {32, 128, 256}
hidden_dims: {100,50; 128,64; 256,128; 512,256,128} |
| GOAD | act: LeakyReLU
n_layers: 5
n_trans: 256
hidden_dim: 8 | lr: {0.001, 0.0001}
trans_dim: {32, 128}
n_trans: {256, 512} |
| NeuTraL | act: LeakyReLU
trans_type: residual
trans_hidden_dims: 50
n_trans: 11 | lr: {0.001, 0.0001}
rep_dim: {32, 128}
hidden_dims: {100,50; 256,128} |
| SLAD | act: LeakyReLU
n_slad_ensemble: 20
magnify_factor: 200
subspace_pool_size: 50
dist_size: 10 | lr: {0.001, 0.0001}
hidden_dims: {100, 200}
n_unified_feature: {128, 256} |
| ICL | act: LeakyReLU
bias: False
temperature:0.01 | lr: {0.001, 0.0001}
rep_dim: {32, 128}
hidden_dims: {100,50; 256,128} |
| DeepIsolationForest | bias: False
lr: 0.001
act: ReLU
max_samples: 256 | rep_dim: {32, 128, 256}
hidden_dims: {100,50; 256,128; 512,256}
n_ensemble: {50, 200}
n_estimators: {6, 10} |
| MCM | enc_layers, dec_layers: 3
hidden_dim: 256
z_dim: 128
mask_layers: 3 | lr: {0.005, 0.001, 0.0005, 0.0001}
mask_number: {10, 15}
lambda: {1, 5, 10} |
| NPT-AD | num_heads: 4
att_score_norm: softmax | lr: {0.01, 0.001, 0.0005, 0.0001}
masking_probability: {0.15, 0.2, 0.25} |
| DRL | enc_layers, dec_layers: 3
hidden_dim: 128
input_info_ratio: 0.1
contrastive_learning_ratio: 0.06
n_basis_vec: 5 | lr: {0.005, 0.001, 0.0005, 0.0001}
input_info_ratio: {0.05, 0.1, 0.15}
contrastive_learning_ratio: {0.03, 0.06, 0.09} |
| DisentAD | qkv_bias: True
n_attr: 18
num_heads: 2 | lr: {0.0001, 0.00001}
hidden_dim: {128, 256} |
| AnoLLM | lr:0.00005
training_steps: 2000
n_permutations: 21 | model: {SmolLM-135M, SmolLM-360M} |

Table 10: Default parameters and hyperparameter search ranges for baselines.

# D DETAILED DESCRIPTION OF ZERO-SHOT LLM FRAMEWORK

## D.1 EVALUATOR LLMS

**GPT-4o-mini (OpenAI).** A lightweight variant of OpenAI's GPT-4o, designed for cost-efficient inference while retaining strong reasoning capabilities. It is optimized for practical deployment scenarios where efficiency is prioritized over maximum performance. (`target_model = gpt-4o-mini-2024-07-18`)

**GPT-4.1 (OpenAI).** The successor to GPT-4, providing enhanced reasoning ability, improved tool-use integration, and stronger multimodal alignment. It represents OpenAI's flagship model as of late 2024. (`target_model = gpt-4.1`)

**Claude-3.7-sonnet (Anthropic).** A mid-sized model in Anthropic's Claude family, optimized for balanced reasoning and efficiency. The "sonnet" variant is designed to offer strong performance on structured tasks while maintaining lower inference cost compared to larger Claude models. (`target_model = claude-3-7-sonnet-20250219`)

**Gemini-2.5-pro (Google DeepMind).** Google's flagship multimodal model, capable of processing text, images, and structured inputs. The "pro" version provides stronger reasoning than lighter variants, and is designed for research and production-scale applications. (`target_model = gemini-2.5-pro`)

**Qwen3-235B (Alibaba).** A large-scale autoregressive language model trained on multilingual and multimodal data. With 235 billion parameters, Qwen3 represents Alibaba's third-generation LLM, aiming to provide competitive performance on both English and Chinese benchmarks. (`target_model = qwen3-235b-a22b`)

## D.2 IMPLEMENTATION DETAILS OF ZERO-SHOT LLM FRAMEWORK

**Data Parameters.** By default, tabular inputs are serialized in raw form without normalization or discretization. However, both normalization (`serialize_normalize_method`) and discretization into a user-defined number of buckets (`serialize_n_buckets`) are available as options. For normal statistics, we provide 5th/95th percentiles derived from the training data (only normal), which serve as representative ranges for contextual grounding.

**Model Parameters.** We use large language models (LLMs) in a zero-shot setting, where the backbone LLM can be selected flexibly (e.g., GPT-4o). The inference batch size is set to 15, with a maximum of 5 retries per query to ensure instruction following and prevent output format errors.

## D.3 PROMPT TEMPLATE

To systematically assess the role of semantic context in anomaly detection, we design a dedicated `PromptGenerator` that produces four distinct prompt types (A–D). Each type selectively incorporates different levels of contextual information—ranging from raw numerical inputs to full metadata descriptions—allowing controlled ablations of the information available to the LLM. All numerical attributes are consistently formatted by rounding to three decimal places, ensuring clarity and reproducibility in prompt construction.

- **Type A** (No Desc.): Normal statistic only (anonymized column names *e.g.*, AA, AB, ...)
- **Type B**: Normal statistic + Feature Description
- **Type C**: Feature description + Domain knowledge
- **Type D** (Full Desc.): Normal statistic + Feature description + Domain knowledge

**Type A Prompt**

```
Analyze the data for anomaly detection.

## Statistical Context
### Numerical features– Normal Ranges:
AA: Normal Range (5th–95th percentile): A – B
AB: Normal Range (5th–95th percentile): A – B

### Categorical features – Normal Values:
AK: D, E, F

Analyze the following data :
Record 0: AA={feature_1 value}, … , AK={feature_k value}
…
Record B: AA={feature_1 value}, … , AK={feature_k value}

Return valid JSON array:
[{"record_id":"0","anomaly_score":0.0,"reasoning":"Analysis result",
"key_features":["feature1", "feature2"]},
…
[{"record_id":"1","anomaly_score":0.0,"reasoning":"Analysis result",
"key_features":["feature1", "feature2"]}]
```

Figure 5: **Type A Prompt Example**

**Type B Prompt**

```
Analyze the data for anomaly detection.

## Features:
### Numerical features:
{feature_1 name}: {feature_1 description}
{feature_2 name}: {feature_1 description}

…

### Categorical features:
{feature_K name} {feature_K description}

## Statistical Context
### Numerical features– Normal Ranges:
{feature_1 name}: Normal Range (5th–95th percentile): A – B
{feature_2 name}: Normal Range (5th–95th percentile): A – B

### Categorical features – Normal Values:
{feature_K name}:D, E, F

## Analysis Guidelines:
– Use feature descriptions to understand data context.
– Look for values that seem unusual given the feature meanings.
– **CRITICAL**: Consider relationships between categorical and numerical features.
– Check if categorical–numerical combinations make logical sense based on feature descriptions.
– Even familiar categorical values can be anomalous if paired with inconsistent numerical values.

Analyze the following data based on the guidelines provided above:
Record 0: {feature_1 name}={feature_1 value}, … , {feature_k name}={feature_k value}
…
Record B: {feature_1 name}={feature_1 value}, … , {feature_k name}={feature_k value}

Return valid JSON array:
[{"record_id":"0","anomaly_score":0.0,"reasoning":"Analysis result",
"key_features":["feature1", "feature2"]},
…
[{"record_id":"1","anomaly_score":0.0,"reasoning":"Analysis result",
"key_features":["feature1", "feature2"]}]
```

Figure 6: **Type B Prompt Example**

---
**Type C Prompt**

```
You are a senior {dataset_name} expert analyzing data for anomaly detection.

## Dataset: {dataset_name} {dataset_description}
## Target Label: {target_description}

## Features:
### Numerical features:
{feature_1 name}: {feature_1 description}
{feature_2 name}: {feature_1 description}

…

### Categorical features:
{feature_K name} {feature_K description}

## Analysis Guidelines:
- Use dataset context and feature descriptions to guide analysis.
- Consider domain-specific patterns and relationships.
- **CRITICAL**: Evaluate categorical-numerical feature combinations for logical consistency.
- Look for anomalies where categorical values are normal but numerical patterns don't align with domain
expectations.
- Use domain knowledge to identify implausible feature combinations.

Analyze the following data based on the guidelines provided above:
Record 0: {feature_1 name}={feature_1 value}, … , {feature_k name}={feature_k value}

…
Record B: {feature_1 name}={feature_1 value}, … , {feature_k name}={feature_k value}

Return valid JSON array:
[{"record_id":"0","anomaly_score":0.0,"reasoning":"Analysis result",
"key_features":["feature1", "feature2"]},

…
[{"record_id":"1","anomaly_score":0.0,"reasoning":"Analysis result",
"key_features":["feature1", "feature2"]}]
```
---

Figure 7: **Type C Prompt Example**

---
**Type D Prompt**

```
You are a senior {dataset_name} expert analyzing data for anomaly detection.

## Dataset: {dataset_name} {dataset_description}
## Target Label: {target_description}

## Features:
### Numerical features:
{feature_1 name}: {feature_1 description}
{feature_2 name}: {feature_1 description}

…

### Categorical features:
{feature_K name} {feature_K description}

## Statistical Context
### Numerical features- Normal Ranges:
{feature_1 name}: Normal Range (5th–95th percentile): A – B
{feature_2 name}: Normal Range (5th–95th percentile): A – B

### Categorical features – Normal Values:
{feature_K name}:D, E, F

## Analysis Guidelines:
### Numerical Analysis:
- Compare each numerical feature to the normal distribution baselines above.
- Identify values beyond 95th percentile OR below 5th percentile (likely abnormal).
- Consider cumulative effect of multiple borderline values.
- **Individual normal values do not guarantee normal overall patterns – evaluate holistic
combinations.**

### Categorical Pattern Analysis:
- **Don't just check if categorical values are 'in-distribution**'.
- Look beyond individual categorical validity – assess contextual appropriateness given other features.
- Identify contradictory categorical combinations or misalignment with numerical severity indicators.

### Domain Analysis:
- Use dataset context and feature descriptions to guide analysis.
- Look for meaningful patterns across multiple features.
- Consider how features relate based on domain knowledge and semantic context.
- Evaluate logical consistency between categorical and numerical features.
- Assess whether feature combinations make domain-specific logical sense.
- Leverage target label definition

Analyze the following data based on the guidelines provided above:
Record 0: {feature_1 name}={feature_1 value}, … , {feature_k name}={feature_k value}

…
Record B: {feature_1 name}={feature_1 value}, … , {feature_k name}={feature_k value}

Return valid JSON array:
[{"record_id":"0","anomaly_score":0.0,"reasoning":"Analysis result",
"key_features":["feature1", "feature2"]},

…
[{"record_id":"1","anomaly_score":0.0,"reasoning":"Analysis result",
"key_features":["feature1", "feature2"]}]
```
---

Figure 8: **Type D Prompt Example**

# E    COMPARISON OF THE LLMS

Table 11: Comparison of the LLMs' AUROC performance across four context types (Type A–D). Each column represents a different LLM evaluated under four prompt conditions: Type A, Type B, Type C, and Type D.

| Dataset | GPT-4o-mini | | | | GPT-4.1 | | | | Claude-3.7-sonnet | | | | Qwen3-235B | | | | Gemini-2.5-pro | | | |
|---|---|---|---|---|---|---|---|---|---|---|---|---|---|---|---|---|---|---|---|---|
| | Type A | Type B | Type C | Type D | Type A | Type B | Type C | Type D | Type A | Type B | Type C | Type D | Type A | Type B | Type C | Type D | Type A | Type B | Type C | Type D |
| automobile | 0.584 | 0.568 | 0.537 | 0.554 | 0.684 | 0.610 | 0.540 | 0.606 | 0.781 | 0.778 | 0.647 | 0.807 | 0.590 | 0.556 | 0.642 | 0.660 | 0.806 | 0.751 | 0.546 | 0.767 |
| | (±0.036) | (±0.041) | (±0.047) | (±0.043) | (±0.020) | (±0.034) | (±0.046) | (±0.033) | (±0.015) | (±0.016) | (±0.029) | (±0.012) | (±0.035) | (±0.042) | (±0.026) | (±0.023) | (±0.012) | (±0.017) | (±0.044) | (±0.016) |
| backdoor | 0.759 | 0.806 | 0.819 | 0.762 | 0.743 | 0.763 | 0.793 | 0.709 | 0.812 | 0.855 | 0.835 | 0.776 | 0.780 | 0.797 | 0.824 | 0.792 | 0.804 | 0.762 | 0.553 | 0.827 |
| | (±0.017) | (±0.012) | (±0.010) | (±0.016) | (±0.018) | (±0.016) | (±0.013) | (±0.019) | (±0.011) | (±0.007) | (±0.009) | (±0.015) | (±0.014) | (±0.013) | (±0.013) | (±0.012) | (±0.012) | (±0.016) | (±0.041) | (±0.010) |
| campaign | 0.526 | 0.505 | 0.458 | 0.493 | 0.562 | 0.581 | 0.502 | 0.595 | 0.554 | 0.555 | 0.522 | 0.618 | 0.508 | 0.530 | 0.479 | 0.557 | 0.596 | 0.558 | 0.528 | 0.813 |
| cardiotocography | 0.771 | 0.778 | 0.724 | 0.781 | 0.741 | 0.764 | 0.686 | 0.776 | 0.686 | 0.798 | 0.827 | 0.834 | 0.587 | 0.689 | 0.650 | 0.753 | 0.643 | 0.685 | 0.784 | 0.829 |
| | (±0.016) | (±0.015) | (±0.019) | (±0.015) | (±0.018) | (±0.016) | (±0.021) | (±0.015) | (±0.021) | (±0.013) | (±0.010) | (±0.009) | (±0.032) | (±0.020) | (±0.024) | (±0.017) | (±0.025) | (±0.021) | (±0.015) | (±0.010) |
| census | 0.655 | 0.669 | 0.672 | 0.654 | 0.656 | 0.706 | 0.710 | 0.709 | 0.657 | 0.674 | 0.665 | 0.662 | 0.635 | 0.713 | 0.564 | 0.677 | 0.662 | 0.721 | 0.686 | 0.895 |
| | (±0.023) | (±0.022) | (±0.022) | (±0.023) | (±0.023) | (±0.019) | (±0.018) | (±0.018) | (±0.023) | (±0.021) | (±0.022) | (±0.022) | (±0.025) | (±0.018) | (±0.034) | (±0.021) | (±0.022) | (±0.018) | (±0.020) | (±0.006) |
| churn | 0.532 | 0.470 | 0.478 | 0.572 | 0.577 | 0.479 | 0.475 | 0.527 | 0.486 | 0.463 | 0.743 | 0.805 | 0.524 | 0.533 | 0.612 | 0.687 | 0.499 | 0.481 | 0.483 | 0.809 |
| | (±0.040) | (±0.046) | (±0.045) | (±0.036) | (±0.033) | (±0.046) | (±0.046) | (±0.040) | (±0.044) | (±0.047) | (±0.018) | (±0.012) | (±0.041) | (±0.040) | (±0.033) | (±0.021) | (±0.043) | (±0.045) | (±0.045) | (±0.012) |
| cirrhosis | 0.760 | 0.730 | 0.761 | 0.806 | 0.665 | 0.745 | 0.594 | 0.811 | 0.796 | 0.762 | 0.803 | 0.823 | 0.742 | 0.738 | 0.588 | 0.854 | 0.694 | 0.775 | 0.650 | 0.850 |
| | (±0.016) | (±0.019) | (±0.016) | (±0.012) | (±0.022) | (±0.017) | (±0.032) | (±0.011) | (±0.013) | (±0.016) | (±0.012) | (±0.010) | (±0.018) | (±0.018) | (±0.032) | (±0.007) | (±0.020) | (±0.015) | (±0.024) | (±0.007) |
| covertype | 0.884 | 0.883 | 0.864 | 0.892 | 0.932 | 0.886 | 0.694 | 0.942 | 0.884 | 0.898 | 0.875 | 0.916 | 0.785 | 0.867 | 0.700 | 0.932 | 0.861 | 0.931 | 0.860 | 0.961 |
| | (±0.006) | (±0.006) | (±0.007) | (±0.005) | (±0.003) | (±0.006) | (±0.020) | (±0.003) | (±0.006) | (±0.005) | (±0.007) | (±0.004) | (±0.012) | (±0.007) | (±0.019) | (±0.003) | (±0.007) | (±0.003) | (±0.007) | (±0.002) |
| credit | 0.540 | 0.508 | 0.522 | 0.511 | 0.504 | 0.475 | 0.496 | 0.613 | 0.524 | 0.518 | 0.500 | 0.573 | 0.522 | 0.550 | 0.560 | 0.552 | 0.501 | 0.507 | 0.516 | 0.681 |
| | (±0.046) | (±0.049) | (±0.048) | (±0.049) | (±0.050) | (±0.053) | (±0.050) | (±0.033) | (±0.048) | (±0.048) | (±0.050) | (±0.036) | (±0.048) | (±0.045) | (±0.044) | (±0.045) | (±0.050) | (±0.049) | (±0.048) | (±0.021) |
| equip | 0.972 | 0.969 | 0.923 | 0.977 | 0.941 | 0.927 | 0.833 | 0.943 | 0.970 | 0.977 | 0.902 | 0.972 | 0.929 | 0.957 | 0.888 | 0.967 | 0.975 | 0.973 | 0.890 | 0.978 |
| | (±0.001) | (±0.002) | (±0.004) | (±0.001) | (±0.003) | (±0.004) | (±0.009) | (±0.003) | (±0.002) | (±0.001) | (±0.005) | (±0.001) | (±0.004) | (±0.002) | (±0.006) | (±0.002) | (±0.001) | (±0.001) | (±0.006) | (±0.001) |
| gallstone | 0.540 | 0.547 | 0.573 | 0.589 | 0.563 | 0.624 | 0.531 | 0.599 | 0.574 | 0.590 | 0.574 | 0.601 | 0.556 | 0.672 | 0.592 | 0.623 | 0.615 | 0.625 | 0.468 | 0.667 |
| | (±0.046) | (±0.044) | (±0.036) | (±0.034) | (±0.035) | (±0.025) | (±0.047) | (±0.035) | (±0.036) | (±0.034) | (±0.036) | (±0.033) | (±0.037) | (±0.022) | (±0.031) | (±0.025) | (±0.039) | (±0.025) | (±0.049) | (±0.022) |
| glass | 0.897 | 0.895 | 0.916 | 0.913 | 0.868 | 0.882 | 0.953 | 0.944 | 0.871 | 0.891 | 0.906 | 0.916 | 0.872 | 0.903 | 0.841 | 0.867 | 0.832 | 0.909 | 0.934 | 0.919 |
| | (±0.005) | (±0.005) | (±0.004) | (±0.004) | (±0.007) | (±0.006) | (±0.002) | (±0.003) | (±0.007) | (±0.005) | (±0.005) | (±0.004) | (±0.007) | (±0.005) | (±0.008) | (±0.007) | (±0.009) | (±0.004) | (±0.003) | (±0.004) |
| glioma | 0.545 | 0.638 | 0.700 | 0.729 | 0.628 | 0.624 | 0.776 | 0.755 | 0.717 | 0.692 | 0.778 | 0.676 | 0.568 | 0.903 | 0.841 | 0.867 | 0.586 | 0.651 | 0.598 | 0.895 |
| | (±0.044) | (±0.025) | (±0.019) | (±0.017) | (±0.025) | (±0.025) | (±0.015) | (±0.017) | (±0.018) | (±0.020) | (±0.015) | (±0.021) | (±0.036) | (±0.005) | (±0.008) | (±0.007) | (±0.033) | (±0.024) | (±0.033) | (±0.005) |
| quasar | 0.576 | 0.820 | 0.585 | 0.770 | 0.810 | 0.859 | 0.226 | 0.846 | 0.908 | 0.882 | 0.649 | 0.940 | 0.736 | 0.890 | 0.435 | 0.919 | 0.816 | 0.875 | 0.269 | 0.961 |
| | (±0.035) | (±0.010) | (±0.033) | (±0.015) | (±0.011) | (±0.007) | (±0.051) | (±0.008) | (±0.004) | (±0.006) | (±0.024) | (±0.003) | (±0.018) | (±0.005) | (±0.038) | (±0.004) | (±0.011) | (±0.010) | (±0.053) | (±0.002) |
| seismic | 0.602 | 0.603 | 0.640 | 0.654 | 0.630 | 0.745 | 0.557 | 0.653 | 0.642 | 0.651 | 0.621 | 0.653 | 0.626 | 0.625 | 0.571 | 0.689 | 0.646 | 0.625 | 0.642 | 0.741 |
| | (±0.033) | (±0.033) | (±0.025) | (±0.023) | (±0.026) | (±0.017) | (±0.037) | (±0.023) | (±0.025) | (±0.024) | (±0.026) | (±0.023) | (±0.026) | (±0.026) | (±0.036) | (±0.020) | (±0.024) | (±0.026) | (±0.025) | (±0.018) |
| stroke | 0.571 | 0.583 | 0.571 | 0.586 | 0.562 | 0.576 | 0.586 | 0.628 | 0.574 | 0.590 | 0.570 | 0.589 | 0.574 | 0.594 | 0.520 | 0.637 | 0.562 | 0.584 | 0.569 | 0.776 |
| | (±0.036) | (±0.034) | (±0.036) | (±0.033) | (±0.037) | (±0.035) | (±0.033) | (±0.025) | (±0.035) | (±0.034) | (±0.036) | (±0.034) | (±0.035) | (±0.032) | (±0.041) | (±0.033) | (±0.037) | (±0.033) | (±0.036) | (±0.015) |
| vertebral | 0.531 | 0.479 | 0.730 | 0.612 | 0.427 | 0.416 | 0.264 | 0.407 | 0.485 | 0.518 | 0.759 | 0.679 | 0.468 | 0.390 | 0.521 | 0.261 | 0.461 | 0.403 | 0.872 | 0.755 |
| | (±0.040) | (±0.046) | (±0.017) | (±0.033) | (±0.050) | (±0.052) | (±0.069) | (±0.055) | (±0.044) | (±0.041) | (±0.016) | (±0.021) | (±0.047) | (±0.054) | (±0.077) | (±0.068) | (±0.047) | (±0.055) | (±0.007) | (±0.017) |
| wbc | 0.949 | 0.948 | 0.942 | 0.957 | 0.868 | 0.912 | 0.953 | 0.944 | 0.950 | 0.950 | 0.951 | 0.953 | 0.849 | 0.885 | 0.770 | 0.919 | 0.861 | 0.959 | 0.943 | 0.968 |
| | (±0.002) | (±0.003) | (±0.003) | (±0.002) | (±0.007) | (±0.004) | (±0.002) | (±0.005) | (±0.003) | (±0.003) | (±0.002) | (±0.002) | (±0.008) | (±0.006) | (±0.015) | (±0.004) | (±0.003) | (±0.003) | (±0.003) | (±0.002) |
| wine | 0.893 | 0.923 | 0.946 | 0.946 | 0.743 | 0.865 | 0.995 | 0.896 | 0.879 | 0.916 | 0.949 | 0.960 | 0.686 | 0.788 | 0.656 | 0.920 | 0.663 | 0.943 | 0.999 | 0.991 |
| | (±0.005) | (±0.004) | (±0.003) | (±0.003) | (±0.018) | (±0.007) | (±0.000) | (±0.005) | (±0.006) | (±0.004) | (±0.003) | (±0.002) | (±0.020) | (±0.012) | (±0.023) | (±0.004) | (±0.022) | (±0.003) | (±0.000) | (±0.000) |
| yeast | 0.746 | 0.737 | 0.731 | 0.760 | 0.819 | 0.806 | 0.470 | 0.798 | 0.752 | 0.747 | 0.746 | 0.778 | 0.762 | 0.745 | 0.675 | 0.807 | 0.731 | 0.718 | 0.672 | 0.861 |
| | (±0.017) | (±0.018) | (±0.018) | (±0.016) | (±0.010) | (±0.012) | (±0.046) | (±0.013) | (±0.017) | (±0.017) | (±0.017) | (±0.015) | (±0.016) | (±0.017) | (±0.022) | (±0.018) | (±0.018) | (±0.019) | (±0.022) | (±0.007) |
| **Average** | 0.692 | 0.703 | 0.705 | 0.726 | 0.696 | 0.712 | 0.632 | 0.735 | 0.725 | 0.735 | 0.741 | 0.777 | 0.665 | 0.716 | 0.631 | 0.747 | 0.691 | 0.721 | 0.673 | 0.847 |

Table 12: Comparison of the LLMs' AUPRC performance across four context types (Type A–D). Each column represents a different LLM evaluated under four prompt conditions: Type A, Type B, Type C, and Type D.

| Dataset | GPT-4o-mini | | | | GPT-4.1 | | | | Claude-3.7-sonnet | | | | Qwen3-235B | | | | Gemini-2.5-pro | | | |
|---|---|---|---|---|---|---|---|---|---|---|---|---|---|---|---|---|---|---|---|---|
| | Type A | Type B | Type C | Type D | Type A | Type B | Type C | Type D | Type A | Type B | Type C | Type D | Type A | Type B | Type C | Type D | Type A | Type B | Type C | Type D |
| automobile | 0.517 | 0.450 | 0.428 | 0.457 | 0.568 | 0.477 | 0.449 | 0.606 | 0.683 | 0.702 | 0.544 | 0.733 | 0.478 | 0.468 | 0.574 | 0.606 | 0.701 | 0.672 | 0.440 | 0.683 |
| | (±0.042) | (±0.036) | (±0.051) | (±0.028) | (±0.021) | (±0.039) | (±0.044) | (±0.018) | (±0.019) | (±0.025) | (±0.048) | (±0.014) | (±0.037) | (±0.041) | (±0.022) | (±0.031) | (±0.016) | (±0.023) | (±0.052) | (±0.015) |
| backdoor | 0.098 | 0.125 | 0.149 | 0.096 | 0.114 | 0.103 | 0.186 | 0.083 | 0.154 | 0.178 | 0.175 | 0.144 | 0.138 | 0.241 | 0.248 | 0.186 | 0.123 | 0.116 | 0.051 | 0.161 |
| | (±0.031) | (±0.038) | (±0.024) | (±0.042) | (±0.033) | (±0.041) | (±0.019) | (±0.046) | (±0.027) | (±0.021) | (±0.034) | (±0.039) | (±0.035) | (±0.018) | (±0.016) | (±0.028) | (±0.040) | (±0.044) | (±0.049) | (±0.023) |
| campaign | 0.387 | 0.378 | 0.349 | 0.369 | 0.407 | 0.412 | 0.375 | 0.434 | 0.398 | 0.415 | 0.389 | 0.448 | 0.370 | 0.392 | 0.363 | 0.413 | 0.440 | 0.413 | 0.390 | 0.683 |
| | (±0.026) | (±0.033) | (±0.045) | (±0.019) | (±0.024) | (±0.017) | (±0.041) | (±0.017) | (±0.035) | (±0.022) | (±0.038) | (±0.035) | (±0.029) | (±0.020) | (±0.046) | (±0.016) | (±0.018) | (±0.027) | (±0.043) | (±0.018) |
| cardiotocography | 0.411 | 0.573 | 0.534 | 0.573 | 0.585 | 0.621 | 0.551 | 0.653 | 0.518 | 0.671 | 0.699 | 0.708 | 0.434 | 0.577 | 0.526 | 0.610 | 0.502 | 0.322 | 0.412 | 0.711 |
| | (±0.037) | (±0.014) | (±0.026) | (±0.026) | (±0.019) | (±0.025) | (±0.043) | (±0.012) | (±0.040) | (±0.047) | (±0.021) | (±0.009) | (±0.043) | (±0.047) | (±0.016) | (±0.009) | (±0.051) | (±0.044) | (±0.041) | (±0.007) |
| census | 0.114 | 0.083 | 0.078 | 0.080 | 0.180 | 0.176 | 0.177 | 0.179 | 0.211 | 0.210 | 0.244 | 0.187 | 0.233 | 0.269 | 0.120 | 0.204 | 0.228 | 0.194 | 0.169 | 0.188 |
| | (±0.044) | (±0.047) | (±0.049) | (±0.046) | (±0.018) | (±0.020) | (±0.016) | (±0.022) | (±0.013) | (±0.015) | (±0.009) | (±0.024) | (±0.011) | (±0.006) | (±0.041) | (±0.017) | (±0.012) | (±0.019) | (±0.025) | (±0.021) |
| churn | 0.436 | 0.410 | 0.408 | 0.463 | 0.479 | 0.479 | 0.412 | 0.411 | 0.420 | 0.411 | 0.406 | 0.629 | 0.699 | 0.428 | 0.438 | 0.533 | 0.541 | 0.415 | 0.412 | 0.703 |
| | (±0.023) | (±0.035) | (±0.036) | (±0.017) | (±0.020) | (±0.020) | (±0.038) | (±0.039) | (±0.031) | (±0.037) | (±0.037) | (±0.011) | (±0.008) | (±0.033) | (±0.024) | (±0.014) | (±0.035) | (±0.041) | (±0.038) | (±0.007) |
| cirrhosis | 0.629 | 0.697 | 0.667 | 0.707 | 0.649 | 0.730 | 0.610 | 0.813 | 0.792 | 0.783 | 0.741 | 0.845 | 0.737 | 0.753 | 0.639 | 0.848 | 0.750 | 0.749 | 0.662 | 0.846 |
| | (±0.032) | (±0.016) | (±0.021) | (±0.015) | (±0.029) | (±0.012) | (±0.036) | (±0.009) | (±0.010) | (±0.011) | (±0.018) | (±0.006) | (±0.014) | (±0.013) | (±0.033) | (±0.009) | (±0.013) | (±0.013) | (±0.024) | (±0.006) |
| covertype | 0.038 | 0.076 | 0.048 | 0.067 | 0.135 | 0.077 | 0.213 | 0.145 | 0.200 | 0.193 | 0.108 | 0.533 | 0.097 | 0.253 | 0.043 | 0.194 | 0.090 | 0.186 | 0.122 | 0.279 |
| | (±0.021) | (±0.026) | (±0.016) | (±0.015) | (±0.025) | (±0.008) | (±0.017) | (±0.017) | (±0.013) | (±0.013) | (±0.030) | (±0.004) | (±0.007) | (±0.041) | (±0.010) | (±0.011) | (±0.029) | (±0.010) | (±0.025) | (±0.006) |
| credit | 0.391 | 0.410 | 0.425 | 0.421 | 0.367 | 0.345 | 0.364 | 0.427 | 0.344 | 0.393 | 0.421 | 0.507 | 0.387 | 0.391 | 0.408 | 0.401 | 0.370 | 0.364 | 0.377 | 0.526 |
| | (±0.025) | (±0.019) | (±0.016) | (±0.018) | (±0.033) | (±0.039) | (±0.036) | (±0.014) | (±0.041) | (±0.024) | (±0.017) | (±0.009) | (±0.026) | (±0.025) | (±0.021) | (±0.022) | (±0.034) | (±0.036) | (±0.030) | (±0.007) |
| equip | 0.649 | 0.640 | 0.630 | 0.552 | 0.687 | 0.625 | 0.659 | 0.712 | 0.924 | 0.907 | 0.754 | 0.920 | 0.718 | 0.842 | 0.703 | 0.867 | 0.918 | 0.806 | 0.705 | 0.919 |
| | (±0.030) | (±0.032) | (±0.034) | (±0.042) | (±0.017) | (±0.029) | (±0.022) | (±0.013) | (±0.005) | (±0.006) | (±0.020) | (±0.004) | (±0.015) | (±0.008) | (±0.018) | (±0.007) | (±0.004) | (±0.010) | (±0.019) | (±0.004) |
| gallstone | 0.498 | 0.588 | 0.522 | 0.537 | 0.544 | 0.594 | 0.520 | 0.551 | 0.592 | 0.497 | 0.558 | 0.584 | 0.542 | 0.615 | 0.592 | 0.576 | 0.591 | 0.591 | 0.492 | 0.615 |
| | (±0.033) | (±0.018) | (±0.027) | (±0.024) | (±0.022) | (±0.016) | (±0.028) | (±0.021) | (±0.015) | (±0.014) | (±0.020) | (±0.017) | (±0.023) | (±0.010) | (±0.012) | (±0.019) | (±0.013) | (±0.014) | (±0.032) | (±0.010) |
| glass | 0.615 | 0.655 | 0.707 | 0.608 | 0.846 | 0.837 | 0.928 | 0.836 | 0.836 | 0.812 | 0.887 | 0.850 | 0.800 | 0.839 | 0.762 | 0.820 | 0.754 | 0.891 | 0.895 | 0.886 |
| | (±0.031) | (±0.027) | (±0.019) | (±0.033) | (±0.008) | (±0.009) | (±0.003) | (±0.009) | (±0.009) | (±0.011) | (±0.009) | (±0.007) | (±0.009) | (±0.016) | (±0.010) | (±0.017) | (±0.005) | (±0.004) | (±0.004) | (±0.005) |
| glioma | 0.576 | 0.569 | 0.480 | 0.594 | 0.605 | 0.605 | 0.758 | 0.773 | 0.624 | 0.556 | 0.509 | 0.846 | 0.548 | 0.613 | 0.557 | 0.725 | 0.594 | 0.640 | 0.577 | 0.865 |
| | (±0.020) | (±0.021) | (±0.036) | (±0.018) | (±0.017) | (±0.017) | (±0.012) | (±0.011) | (±0.015) | (±0.022) | (±0.030) | (±0.007) | (±0.025) | (±0.018) | (±0.022) | (±0.013) | (±0.018) | (±0.014) | (±0.021) | (±0.006) |
| quasar | 0.359 | 0.635 | 0.376 | 0.534 | 0.586 | 0.670 | 0.261 | 0.663 | 0.843 | 0.791 | 0.438 | 0.899 | 0.537 | 0.809 | 0.279 | 0.848 | 0.586 | 0.715 | 0.296 | 0.918 |
| | (±0.039) | (±0.013) | (±0.037) | (±0.020) | (±0.017) | (±0.011) | (±0.044) | (±0.012) | (±0.007) | (±0.009) | (±0.033) | (±0.004) | (±0.021) | (±0.008) | (±0.041) | (±0.006) | (±0.017) | (±0.013) | (±0.042) | (±0.003) |
| seismic | 0.161 | 0.200 | 0.163 | 0.238 | 0.427 | 0.228 | 0.105 | 0.197 | 0.173 | 0.174 | 0.177 | 0.208 | 0.200 | 0.188 | 0.149 | 0.227 | 0.210 | 0.182 | 0.176 | 0.214 |
| | (±0.032) | (±0.026) | (±0.021) | (±0.019) | (±0.010) | (±0.018) | (±0.035) | (±0.027) | (±0.030) | (±0.030) | (±0.029) | (±0.024) | (±0.026) | (±0.028) | (±0.043) | (±0.020) | (±0.023) | (±0.029) | (±0.031) | (±0.022) |
| stroke | 0.106 | 0.098 | 0.083 | 0.096 | 0.586 | 0.670 | 0.261 | 0.663 | 0.098 | 0.090 | 0.109 | 0.143 | 0.103 | 0.108 | 0.083 | 0.124 | 0.103 | 0.101 | 0.096 | 0.206 |
| | (±0.029) | (±0.032) | (±0.035) | (±0.031) | (±0.017) | (±0.011) | (±0.044) | (±0.012) | (±0.032) | (±0.034) | (±0.028) | (±0.028) | (±0.030) | (±0.028) | (±0.035) | (±0.025) | (±0.030) | (±0.031) | (±0.033) | (±0.016) |
| vertebral | 0.493 | 0.447 | 0.430 | 0.433 | 0.442 | 0.429 | 0.364 | 0.421 | 0.430 | 0.401 | 0.453 | 0.419 | 0.462 | 0.429 | 0.379 | 0.378 | 0.460 | 0.420 | 0.802 | 0.694 |
| | (±0.027) | (±0.032) | (±0.035) | (±0.033) | (±0.033) | (±0.035) | (±0.041) | (±0.037) | (±0.035) | (±0.031) | (±0.037) | (±0.036) | (±0.035) | (±0.040) | (±0.040) | (±0.035) | (±0.037) | (±0.037) | (±0.011) | (±0.018) |
| wbc | 0.779 | 0.814 | 0.551 | 0.834 | 0.860 | 0.906 | 0.958 | 0.944 | 0.920 | 0.932 | 0.953 | 0.965 | 0.858 | 0.862 | 0.746 | 0.894 | 0.837 | 0.940 | 0.925 | 0.963 |
| | (±0.012) | (±0.009) | (±0.028) | (±0.008) | (±0.006) | (±0.004) | (±0.002) | (±0.003) | (±0.004) | (±0.003) | (±0.002) | (±0.001) | (±0.007) | (±0.006) | (±0.015) | (±0.005) | (±0.008) | (±0.003) | (±0.003) | (±0.001) |
| wine | 0.450 | 0.564 | 0.399 | 0.594 | 0.678 | 0.794 | 0.993 | 0.828 | 0.709 | 0.880 | 0.994 | 0.995 | 0.612 | 0.719 | 0.576 | 0.888 | 0.611 | 0.887 | 0.999 | 0.985 |
| | (±0.036) | (±0.021) | (±0.043) | (±0.028) | (±0.016) | (±0.010) | (±0.001) | (±0.009) | (±0.014) | (±0.006) | (±0.001) | (±0.001) | (±0.025) | (±0.014) | (±0.022) | (±0.007) | (±0.025) | (±0.005) | (±0.000) | (±0.001) |
| yeast | 0.294 | 0.269 | 0.170 | 0.301 | 0.403 | 0.356 | 0.114 | 0.334 | 0.404 | 0.343 | 0.248 | 0.414 | 0.296 | 0.269 | 0.262 | 0.390 | 0.286 | 0.276 | 0.190 | 0.449 |
| | (±0.035) | (±0.038) | (±0.047) | (±0.033) | (±0.023) | (±0.027) | (±0.041) | (±0.031) | (±0.023) | (±0.030) | (±0.043) | (±0.022) | (±0.035) | (±0.038) | (±0.039) | (±0.026) | (±0.037) | (±0.036) | (±0.046) | (±0.019) |
| **Average** | 0.400 | 0.434 | 0.380 | 0.424 | 0.507 | 0.507 | 0.463 | 0.534 | 0.514 | 0.522 | 0.490 | 0.599 | 0.462 | 0.503 | 0.422 | 0.535 | 0.485 | 0.494 | 0.460 | 0.625 |

# F FULL BASELINE RESULTS

Table 13: AUROC Performance across all baselines.

| Dataset | OCSVM | LOF | KNN | IForest | PCA | DeepSVDD | REPEN | RDP | RCA | GOAD | NeuTraL | SLAD | ICL | DIF | MCM | NPT-AD | DRL | DisentAD | AnoLLM |
|---|---|---|---|---|---|---|---|---|---|---|---|---|---|---|---|---|---|---|---|
| automobile | 0.645 | 0.688 | 0.656 | 0.648 | 0.497 | 0.845 | 0.611 | 0.774 | 0.679 | 0.676 | 0.810 | 0.780 | 0.813 | 0.647 | 0.801 | 0.642 | 0.762 | 0.644 | 0.624 |
|  | (±0.025) | (±0.029) | (±0.015) | (±0.034) | (±0.028) | (±0.030) | (±0.059) | (±0.017) | (±0.025) | (±0.065) | (±0.012) | (±0.030) | (±0.018) | (±0.023) | (±0.013) | (±0.043) | (±0.066) | (±0.029) | (±0.021) |
| backdoor | 0.876 | 0.702 | 0.934 | 0.888 | 0.822 | 0.800 | 0.856 | 0.897 | 0.915 | 0.889 | 0.948 | 0.935 | 0.978 | 0.973 | 0.961 | 0.971 | 0.902 | 0.886 | 0.885 |
|  | (±0.004) | (±0.008) | (±0.004) | (±0.004) | (±0.003) | (±0.116) | (±0.013) | (±0.001) | (±0.003) | (±0.007) | (±0.001) | (±0.001) | (±0.013) | (±0.002) | (±0.003) | (±0.001) | (±0.017) | (±0.009) | (±0.010) |
| campaign | 0.700 | 0.669 | 0.754 | 0.732 | 0.750 | 0.721 | 0.661 | 0.693 | 0.722 | 0.740 | 0.757 | 0.717 | 0.672 | 0.677 | 0.761 | 0.739 | 0.737 | 0.668 | 0.738 |
|  | (±0.004) | (±0.005) | (±0.007) | (±0.006) | (±0.003) | (±0.014) | (±0.013) | (±0.010) | (±0.010) | (±0.002) | (±0.008) | (±0.002) | (±0.007) | (±0.004) | (±0.005) | (±0.012) | (±0.018) | (±0.018) | (±0.012) |
| cardiotocography | 0.868 | 0.826 | 0.780 | 0.830 | 0.840 | 0.813 | 0.766 | 0.766 | 0.858 | 0.798 | 0.780 | 0.764 | 0.734 | 0.740 | 0.843 | 0.771 | 0.847 | 0.849 | 0.735 |
|  | (±0.006) | (±0.010) | (±0.011) | (±0.006) | (±0.005) | (±0.015) | (±0.051) | (±0.008) | (±0.005) | (±0.005) | (±0.017) | (±0.018) | (±0.019) | (±0.007) | (±0.019) | (±0.001) | (±0.019) | (±0.006) | (±0.014) |
| census | 0.647 | 0.596 | 0.738 | 0.620 | 0.627 | 0.712 | 0.811 | 0.717 | 0.745 | 0.708 | 0.686 | 0.687 | 0.754 | 0.726 | 0.729 | 0.700 | 0.728 | 0.673 | 0.737 |
|  | (±0.001) | (±0.001) | (±0.001) | (±0.000) | (±0.002) | (±0.048) | (±0.007) | (±0.002) | (±0.002) | (±0.002) | (±0.002) | (±0.002) | (±0.005) | (±0.005) | (±0.007) | (±0.005) | (±0.036) | (±0.054) | (±0.009) |
| churn | 0.651 | 0.465 | 0.532 | 0.506 | 0.579 | 0.573 | 0.546 | 0.523 | 0.616 | 0.683 | 0.651 | 0.571 | 0.568 | 0.494 | 0.580 | 0.647 | 0.553 | 0.578 | 0.557 |
|  | (±0.007) | (±0.007) | (±0.003) | (±0.010) | (±0.023) | (±0.071) | (±0.058) | (±0.004) | (±0.030) | (±0.004) | (±0.014) | (±0.010) | (±0.005) | (±0.024) | (±0.039) | (±0.012) | (±0.038) |  | (±0.018) |
| cirrhosis | 0.823 | 0.843 | 0.843 | 0.849 | 0.822 | 0.807 | 0.873 | 0.795 | 0.834 | 0.823 | 0.817 | 0.786 | 0.804 | 0.820 | 0.830 | 0.803 | 0.814 | 0.829 | 0.850 |
|  | (±0.012) | (±0.013) | (±0.012) | (±0.013) | (±0.017) | (±0.018) | (±0.009) | (±0.023) | (±0.012) | (±0.015) | (±0.019) | (±0.022) | (±0.024) | (±0.011) | (±0.020) | (±0.018) | (±0.012) | (±0.017) | (±0.015) |
| covertype | 0.998 | 0.980 | 0.991 | 0.968 | 0.976 | 0.971 | 0.986 | 0.992 | 0.998 | 0.998 | 0.998 | 0.999 | 0.998 | 0.998 | 0.998 | 0.967 | 0.995 | 0.897 | 0.990 |
|  | (±0.000) | (±0.003) | (±0.000) | (±0.003) | (±0.001) | (±0.007) | (±0.006) | (±0.001) | (±0.000) | (±0.000) | (±0.000) | (±0.001) | (±0.000) | (±0.000) | (±0.007) | (±0.002) |  | (±0.010) | (±0.003) |
| credit | 0.717 | 0.590 | 0.692 | 0.639 | 0.684 | 0.638 | 0.661 | 0.593 | 0.702 | 0.682 | 0.690 | 0.691 | 0.587 | 0.689 | 0.671 | 0.561 | 0.680 | 0.649 | 0.495 |
|  | (±0.003) | (±0.007) | (±0.003) | (±0.003) | (±0.003) | (±0.039) | (±0.012) | (±0.009) | (±0.002) | (±0.016) | (±0.002) | (±0.019) | (±0.002) | (±0.004) | (±0.007) | (±0.020) | (±0.016) | (±0.019) |  |
| equip | 0.987 | 0.987 | 0.987 | 0.980 | 0.986 | 0.984 | 0.986 | 0.973 | 0.986 | 0.940 | 0.985 | 0.981 | 0.966 | 0.987 | 0.986 | 0.987 | 0.984 | 0.979 | 0.986 |
|  | (±0.000) | (±0.000) | (±0.000) | (±0.000) | (±0.000) | (±0.000) | (±0.000) | (±0.001) | (±0.000) | (±0.004) | (±0.001) | (±0.001) | (±0.002) | (±0.000) | (±0.001) | (±0.001) | (±0.001) | (±0.003) | (±0.002) |
| gallstone | 0.671 | 0.704 | 0.702 | 0.650 | 0.637 | 0.731 | 0.619 | 0.725 | 0.720 | 0.733 | 0.767 | 0.758 | 0.731 | 0.649 | 0.757 | 0.630 | 0.745 | 0.643 | 0.574 |
|  | (±0.047) | (±0.032) | (±0.040) | (±0.021) | (±0.026) | (±0.020) | (±0.026) | (±0.029) | (±0.033) | (±0.027) | (±0.029) | (±0.028) | (±0.023) | (±0.026) | (±0.034) | (±0.021) | (±0.029) | (±0.042) | (±0.028) |
| glass | 0.959 | 0.974 | 0.953 | 0.944 | 0.938 | 0.963 | 0.936 | 0.965 | 0.959 | 0.940 | 0.968 | 0.943 | 0.965 | 0.936 | 0.966 | 0.931 | 0.949 | 0.944 | 0.907 |
|  | (±0.010) | (±0.010) | (±0.014) | (±0.016) | (±0.011) | (±0.012) | (±0.018) | (±0.011) | (±0.014) | (±0.017) | (±0.010) | (±0.013) | (±0.008) | (±0.012) | (±0.012) | (±0.009) | (±0.013) | (±0.014) | (±0.016) |
| glioma | 0.725 | 0.711 | 0.787 | 0.722 | 0.715 | 0.766 | 0.727 | 0.739 | 0.787 | 0.772 | 0.784 | 0.886 | 0.831 | 0.686 | 0.790 | 0.724 | 0.764 | 0.777 | 0.708 |
|  | (±0.012) | (±0.016) | (±0.006) | (±0.013) | (±0.012) | (±0.012) | (±0.039) | (±0.015) | (±0.008) | (±0.009) | (±0.013) | (±0.005) | (±0.026) | (±0.012) | (±0.019) | (±0.019) | (±0.040) | (±0.041) | (±0.022) |
| quasar | 0.930 | 0.966 | 0.969 | 0.915 | 0.894 | 0.804 | 0.931 | 0.953 | 0.950 | 0.901 | 0.954 | 0.952 | 0.856 | 0.793 | 0.950 | 0.826 | 0.897 | 0.922 | 0.961 |
|  | (±0.008) | (±0.009) | (±0.004) | (±0.007) | (±0.013) | (±0.135) | (±0.014) | (±0.004) | (±0.012) | (±0.014) | (±0.006) | (±0.009) | (±0.025) | (±0.042) | (±0.007) | (±0.023) | (±0.024) | (±0.009) | (±0.014) |
| seismic | 0.710 | 0.615 | 0.741 | 0.710 | 0.671 | 0.729 | 0.739 | 0.719 | 0.729 | 0.720 | 0.704 | 0.713 | 0.727 | 0.706 | 0.736 | 0.716 | 0.705 | 0.714 | 0.741 |
|  | (±0.002) | (±0.007) | (±0.005) | (±0.006) | (±0.003) | (±0.015) | (±0.011) | (±0.006) | (±0.005) | (±0.004) | (±0.005) | (±0.006) | (±0.008) | (±0.005) | (±0.019) | (±0.003) | (±0.016) | (±0.015) | (±0.011) |
| stroke | 0.716 | 0.625 | 0.732 | 0.679 | 0.678 | 0.722 | 0.709 | 0.674 | 0.764 | 0.702 | 0.704 | 0.678 | 0.635 | 0.705 | 0.753 | 0.586 | 0.727 | 0.801 | 0.687 |
|  | (±0.004) | (±0.022) | (±0.005) | (±0.003) | (±0.004) | (±0.005) | (±0.010) | (±0.009) | (±0.011) | (±0.005) | (±0.004) | (±0.007) | (±0.017) | (±0.007) | (±0.005) | (±0.002) | (±0.025) | (±0.003) | (±0.010) |
| vertebral | 0.638 | 0.540 | 0.469 | 0.500 | 0.524 | 0.581 | 0.340 | 0.551 | 0.566 | 0.687 | 0.687 | 0.600 | 0.512 | 0.340 | 0.615 | 0.589 | 0.477 | 0.664 | 0.597 |
|  | (±0.043) | (±0.021) | (±0.038) | (±0.013) | (±0.061) | (±0.062) | (±0.101) | (±0.018) | (±0.020) | (±0.037) | (±0.040) | (±0.042) | (±0.045) | (±0.008) | (±0.128) | (±0.056) | (±0.042) | (±0.043) | (±0.014) |
| wbc | 0.969 | 0.960 | 0.961 | 0.961 | 0.960 | 0.969 | 0.876 | 0.924 | 0.964 | 0.945 | 0.748 | 0.916 | 0.920 | 0.679 | 0.957 | 0.862 | 0.954 | 0.981 | 0.871 |
|  | (±0.006) | (±0.006) | (±0.006) | (±0.004) | (±0.006) | (±0.005) | (±0.012) | (±0.010) | (±0.007) | (±0.008) | (±0.030) | (±0.008) | (±0.012) | (±0.016) | (±0.006) | (±0.003) | (±0.010) | (±0.004) | (±0.031) |
| wine | 0.957 | 0.974 | 0.976 | 0.987 | 0.980 | 0.974 | 0.472 | 0.939 | 0.973 | 0.939 | 0.989 | 0.982 | 0.969 | 0.760 | 0.979 | 0.874 | 0.958 | 0.976 | 0.933 |
|  | (±0.012) | (±0.014) | (±0.012) | (±0.006) | (±0.007) | (±0.011) | (±0.213) | (±0.032) | (±0.011) | (±0.026) | (±0.003) | (±0.012) | (±0.018) | (±0.033) | (±0.012) | (±0.007) | (±0.012) | (±0.013) | (±0.020) |
| yeast | 0.875 | 0.803 | 0.831 | 0.838 | 0.861 | 0.820 | 0.826 | 0.786 | 0.866 | 0.836 | 0.841 | 0.846 | 0.643 | 0.815 | 0.856 | 0.806 | 0.841 | 0.912 | 0.809 |
|  | (±0.008) | (±0.010) | (±0.007) | (±0.014) | (±0.009) | (±0.015) | (±0.049) | (±0.004) | (±0.008) | (±0.007) | (±0.012) | (±0.019) | (±0.030) | (±0.004) | (±0.014) | (±0.018) | (±0.033) | (±0.030) | (±0.018) |

Table 14: AUPRC Performance across all baselines.

| Dataset | OCSVM | LOF | KNN | IForest | PCA | DeepSVDD | REPEN | RDP | RCA | GOAD | NeuTraL | SLAD | ICL | DIF | MCM | NPT-AD | DRL | DisentAD | AnoLLM |
|---|---|---|---|---|---|---|---|---|---|---|---|---|---|---|---|---|---|---|---|
| automobile | 0.578 | 0.618 | 0.534 | 0.543 | 0.419 | 0.762 | 0.496 | 0.637 | 0.566 | 0.620 | 0.718 | 0.665 | 0.733 | 0.544 | 0.675 | 0.539 | 0.661 | 0.619 | 0.553 |
|  | (±0.039) | (±0.031) | (±0.017) | (±0.039) | (±0.040) | (±0.065) | (±0.043) | (±0.049) | (±0.035) | (±0.094) | (±0.021) | (±0.045) | (±0.029) | (±0.038) | (±0.038) | (±0.057) | (±0.069) | (±0.032) | (±0.025) |
| backdoor | 0.043 | 0.052 | 0.092 | 0.032 | 0.019 | 0.131 | 0.168 | 0.276 | 0.054 | 0.245 | 0.908 | 0.867 | 0.922 | 0.494 | 0.188 | 0.482 | 0.132 | 0.290 | 0.319 |
|  | (±0.002) | (±0.006) | (±0.005) | (±0.001) | (±0.000) | (±0.039) | (±0.011) | (±0.007) | (±0.003) | (±0.027) | (±0.000) | (±0.003) | (±0.014) | (±0.044) | (±0.012) | (±0.031) | (±0.025) | (±0.033) | (±0.012) |
| campaign | 0.539 | 0.524 | 0.572 | 0.559 | 0.580 | 0.561 | 0.517 | 0.517 | 0.560 | 0.563 | 0.590 | 0.563 | 0.514 | 0.507 | 0.588 | 0.571 | 0.561 | 0.502 | 0.573 |
|  | (±0.007) | (±0.004) | (±0.009) | (±0.010) | (±0.005) | (±0.013) | (±0.007) | (±0.011) | (±0.012) | (±0.006) | (±0.011) | (±0.006) | (±0.007) | (±0.007) | (±0.007) | (±0.016) | (±0.018) | (±0.008) | (±0.011) |
| cardiotocography | 0.788 | 0.735 | 0.707 | 0.730 | 0.750 | 0.767 | 0.672 | 0.697 | 0.774 | 0.699 | 0.724 | 0.712 | 0.695 | 0.678 | 0.757 | 0.651 | 0.767 | 0.784 | 0.630 |
|  | (±0.012) | (±0.013) | (±0.016) | (±0.008) | (±0.008) | (±0.025) | (±0.040) | (±0.011) | (±0.016) | (±0.021) | (±0.016) | (±0.017) | (±0.025) | (±0.009) | (±0.021) | (±0.033) | (±0.025) | (±0.011) | (±0.018) |
| census | 0.354 | 0.168 | 0.335 | 0.136 | 0.153 | 0.346 | 0.411 | 0.304 | 0.395 | 0.304 | 0.308 | 0.281 | 0.371 | 0.315 | 0.275 | 0.348 | 0.324 | 0.222 |  |
|  | (±0.005) | (±0.003) | (±0.003) | (±0.002) | (±0.001) | (±0.060) | (±0.015) | (±0.003) | (±0.004) | (±0.006) | (±0.003) | (±0.005) | (±0.005) | (±0.012) | (±0.010) | (±0.022) | (±0.040) | (±0.014) |  |
| churn | 0.520 | 0.395 | 0.420 | 0.407 | 0.486 | 0.477 | 0.464 | 0.430 | 0.505 | 0.563 | 0.527 | 0.471 | 0.458 | 0.406 | 0.463 | 0.531 | 0.463 | 0.484 | 0.429 |
|  | (±0.010) | (±0.007) | (±0.003) | (±0.005) | (±0.014) | (±0.071) | (±0.048) | (±0.002) | (±0.034) | (±0.005) | (±0.003) | (±0.010) | (±0.009) | (±0.005) | (±0.025) | (±0.036) | (±0.005) | (±0.046) | (±0.016) |
| cirrhosis | 0.836 | 0.850 | 0.850 | 0.866 | 0.834 | 0.819 | 0.876 | 0.792 | 0.845 | 0.814 | 0.820 | 0.813 | 0.816 | 0.835 | 0.841 | 0.801 | 0.822 | 0.836 | 0.824 |
|  | (±0.014) | (±0.014) | (±0.013) | (±0.012) | (±0.011) | (±0.011) | (±0.011) | (±0.022) | (±0.020) | (±0.021) | (±0.027) | (±0.013) | (±0.016) | (±0.016) | (±0.036) | (±0.015) | (±0.017) |  |  |
| covertype | 0.871 | 0.802 | 0.874 | 0.320 | 0.304 | 0.825 | 0.468 | 0.743 | 0.915 | 0.890 | 0.940 | 0.914 | 0.924 | 0.923 | 0.880 | 0.267 | 0.781 | 0.335 | 0.654 |
|  | (±0.011) | (±0.011) | (±0.004) | (±0.023) | (±0.010) | (±0.031) | (±0.114) | (±0.016) | (±0.012) | (±0.026) | (±0.003) | (±0.009) | (±0.011) | (±0.015) | (±0.027) | (±0.014) | (±0.062) | (±0.021) | (±0.020) |
| credit | 0.610 | 0.412 | 0.556 | 0.483 | 0.547 | 0.540 | 0.559 | 0.410 | 0.600 | 0.575 | 0.583 | 0.467 | 0.566 | 0.545 | 0.379 | 0.525 | 0.495 | 0.352 |  |
|  | (±0.005) | (±0.006) | (±0.014) | (±0.003) | (±0.005) | (±0.033) | (±0.012) | (±0.008) | (±0.004) | (±0.008) | (±0.027) | (±0.003) | (±0.017) | (±0.003) | (±0.006) | (±0.006) | (±0.037) | (±0.021) | (±0.018) |
| equip | 0.972 | 0.971 | 0.972 | 0.950 | 0.970 | 0.965 | 0.969 | 0.941 | 0.971 | 0.913 | 0.965 | 0.954 | 0.935 | 0.970 | 0.970 | 0.972 | 0.966 | 0.955 | 0.972 |
|  | (±0.000) | (±0.001) | (±0.000) | (±0.001) | (±0.001) | (±0.001) | (±0.001) | (±0.002) | (±0.001) | (±0.005) | (±0.001) | (±0.003) | (±0.004) | (±0.000) | (±0.001) | (±0.002) | (±0.004) | (±0.009) | (±0.002) |
| gallstone | 0.667 | 0.702 | 0.691 | 0.637 | 0.630 | 0.724 | 0.627 | 0.727 | 0.710 | 0.718 | 0.753 | 0.746 | 0.741 | 0.649 | 0.744 | 0.618 | 0.733 | 0.659 | 0.580 |
|  | (±0.052) | (±0.036) | (±0.046) | (±0.008) | (±0.036) | (±0.031) | (±0.027) | (±0.023) | (±0.037) | (±0.033) | (±0.028) | (±0.032) | (±0.022) | (±0.036) | (±0.035) | (±0.021) | (±0.022) | (±0.048) | (±0.027) |
| glass | 0.920 | 0.942 | 0.894 | 0.869 | 0.878 | 0.918 | 0.850 | 0.919 | 0.907 | 0.896 | 0.928 | 0.891 | 0.936 | 0.881 | 0.917 | 0.889 | 0.896 | 0.886 | 0.808 |
|  | (±0.027) | (±0.021) | (±0.038) | (±0.049) | (±0.035) | (±0.041) | (±0.043) | (±0.035) | (±0.043) | (±0.041) | (±0.035) | (±0.038) | (±0.026) | (±0.029) | (±0.042) | (±0.015) | (±0.039) | (±0.036) | (±0.024) |
| glioma | 0.761 | 0.673 | 0.785 | 0.706 | 0.703 | 0.753 | 0.712 | 0.718 | 0.801 | 0.781 | 0.741 | 0.863 | 0.775 | 0.684 | 0.802 | 0.719 | 0.702 | 0.784 | 0.688 |
|  | (±0.013) | (±0.025) | (±0.007) | (±0.014) | (±0.016) | (±0.023) | (±0.038) | (±0.016) | (±0.009) | (±0.010) | (±0.024) | (±0.006) | (±0.023) | (±0.011) | (±0.008) | (±0.013) | (±0.034) | (±0.024) | (±0.022) |
| quasar | 0.918 | 0.949 | 0.952 | 0.758 | 0.863 | 0.707 | 0.891 | 0.932 | 0.937 | 0.860 | 0.932 | 0.926 | 0.822 | 0.776 | 0.927 | 0.748 | 0.850 | 0.908 | 0.936 |
|  | (±0.006) | (±0.009) | (±0.002) | (±0.022) | (±0.025) | (±0.191) | (±0.016) | (±0.002) | (±0.010) | (±0.036) | (±0.003) | (±0.005) | (±0.020) | (±0.006) | (±0.034) | (±0.040) | (±0.008) | (±0.021) | (±0.013) |
| seismic | 0.248 | 0.168 | 0.257 | 0.247 | 0.204 | 0.256 | 0.259 | 0.231 | 0.253 | 0.245 | 0.232 | 0.239 | 0.239 | 0.253 | 0.249 | 0.215 | 0.246 | 0.231 | 0.283 |
|  | (±0.004) | (±0.005) | (±0.006) | (±0.010) | (±0.004) | (±0.005) | (±0.011) | (±0.009) | (±0.006) | (±0.005) | (±0.008) | (±0.007) | (±0.011) | (±0.006) | (±0.016) | (±0.011) | (±0.009) | (±0.009) | (±0.009) |
| stroke | 0.228 | 0.116 | 0.227 | 0.214 | 0.179 | 0.178 | 0.232 | 0.153 | 0.247 | 0.198 | 0.162 | 0.185 | 0.150 | 0.217 | 0.242 | 0.111 | 0.206 | 0.275 | 0.163 |
|  | (±0.011) | (±0.004) | (±0.008) | (±0.009) | (±0.006) | (±0.007) | (±0.007) | (±0.007) | (±0.012) | (±0.007) | (±0.010) | (±0.010) | (±0.011) | (±0.012) | (±0.013) | (±0.002) | (±0.004) | (±0.010) | (±0.010) |
| vertebral | 0.545 | 0.513 | 0.446 | 0.463 | 0.485 | 0.584 | 0.394 | 0.518 | 0.504 | 0.597 | 0.616 | 0.569 | 0.525 | 0.385 | 0.582 | 0.541 | 0.461 | 0.577 | 0.535 |
|  | (±0.032) | (±0.021) | (±0.021) | (±0.013) | (±0.010) | (±0.049) | (±0.052) | (±0.013) | (±0.012) | (±0.030) | (±0.033) | (±0.033) | (±0.043) | (±0.002) | (±0.117) | (±0.055) | (±0.037) | (±0.027) | (±0.028) |
| wbc | 0.970 | 0.962 | 0.962 | 0.956 | 0.959 | 0.974 | 0.845 | 0.922 | 0.962 | 0.949 | 0.704 | 0.918 | 0.942 | 0.781 | 0.955 | 0.856 | 0.950 | 0.981 | 0.827 |
|  | (±0.007) | (±0.007) | (±0.008) | (±0.009) | (±0.010) | (±0.005) | (±0.022) | (±0.013) | (±0.011) | (±0.009) | (±0.043) | (±0.009) | (±0.011) | (±0.014) | (±0.010) | (±0.021) | (±0.025) | (±0.005) | (±0.019) |
| wine | 0.933 | 0.957 | 0.960 | 0.980 | 0.967 | 0.959 | 0.436 | 0.889 | 0.954 | 0.917 | 0.980 | 0.966 | 0.941 | 0.714 | 0.969 | 0.784 | 0.926 | 0.949 | 0.874 |
|  | (±0.022) | (±0.027) | (±0.021) | (±0.014) | (±0.015) | (±0.007) | (±0.180) | (±0.051) | (±0.021) | (±0.035) | (±0.010) | (±0.026) | (±0.042) | (±0.062) | (±0.019) | (±0.021) | (±0.021) | (±0.030) | (±0.023) |
| yeast | 0.427 | 0.311 | 0.369 | 0.435 | 0.384 | 0.359 | 0.370 | 0.299 | 0.420 | 0.390 | 0.351 | 0.384 | 0.242 | 0.361 | 0.455 | 0.288 | 0.381 | 0.648 | 0.369 |
|  | (±0.027) | (±0.018) | (±0.004) | (±0.055) | (±0.022) | (±0.030) | (±0.102) | (±0.019) | (±0.029) | (±0.013) | (±0.018) | (±0.045) | (±0.024) | (±0.008) | (±0.051) | (±0.031) | (±0.061) | (±0.082) | (±0.040) |

# G    PERFORMANCE DETAILS OF KEY FEATURE ESTIMATION

To support the results in Table 6, this section details the procedure used to measure the key feature alignment metric. Building on the insight from TabLLM (Hegselmann et al., 2023), which emphasized the importance of feature attribution analysis for interpreting model reasoning, we adopt a similar perspective for evaluating LLM explanations. Concretely, we assess whether the key features predicted by an LLM align with attribution scores derived from a supervised reference model. To this end, we train an XGBoost classifier using ground-truth labels and compute SHAP (Lundberg & Lee, 2017) values for each test instance. For a given anomalous instance $x_i$, the top-$K$ features ranked by absolute SHAP values define the reference set:

$$R_i^{(K)} = \text{TopK}\left(|\phi(x_i)|\right),$$

where $\phi(x_i)$ denotes the SHAP value vector for instance $x_i$. In parallel, the LLM outputs a set of key features for $x_i$, which we denote as $\hat{F}_i$. We then measure alignment between the two sets using the F1 score at $K$:

$$\text{F1@}K(x_i) = \frac{2 \cdot |R_i^{(K)} \cap \hat{F}_i|}{|R_i^{(K)}| + |\hat{F}_i|}.$$

The final metric is averaged over all anomalous instances:

$$\text{F1@}K = \frac{1}{|\mathcal{A}|} \sum_{x_i \in \mathcal{A}} \text{F1@}K(x_i),$$

where $\mathcal{A}$ denotes the set of anomalous samples. Overall, providing textual metadata (Type D) leads to clear improvements in F1 alignment compared to the no-description setting (Type A) across all LLMs (see Table 15). The effect is particularly pronounced in domain-specific datasets such as glioma, wine, equip, and glass, where feature semantics are well-defined and textual descriptions provide crucial contextual grounding that helps the LLM better align its reasoning with SHAP-derived ground truth. Moderate but stable improvements are also observed in datasets like campaign, churn, gallstone, wbc, and vertebral, indicating that even mid-level semantic cues can meaningfully guide feature-level interpretation. In contrast, limited or negative effects arise in datasets such as backdoor, covertype, quasar, and stroke. These cases are largely explained by factors such as ceiling effects from already high baseline alignment (e.g., quasar), structural complexity that weakens the usefulness of textual cues (e.g., covertype), or anomaly definitions that are too simple for additional metadata to provide substantial benefit (e.g., stroke). Despite such exceptions, textual metadata generally serves as a strong semantic signal that enhances the LLM's reasoning alignment, with the most substantial gains appearing in settings where domain-specific context plays a central role. However, this reasoning-alignment metric is not a full measure of explanation quality. Thus, the improvements from textual metadata should be seen as enhanced semantic grounding rather than definitive gains in explanation quality.

Table 15: **Impact of Textual Metadata on Reasoning Alignment across 20 Datasets.** We report F1-Score@K ($K = 1, 3$) on anomalous instances for Type A (No Description) and Type D (Full Description) across five LLMs.

| Dataset | GPT-4o-mini | | | | GPT-4.1 | | | | Claude-3.7-Sonnet | | | | Qwen3-235B | | | | Gemini-2.5-Pro | | | |
|---|---|---|---|---|---|---|---|---|---|---|---|---|---|---|---|---|---|---|---|---|
| | F1@1 | | F1@3 | | F1@1 | | F1@3 | | F1@1 | | F1@3 | | F1@1 | | F1@3 | | F1@1 | | F1@3 | |
| | Type A | Type D | Type A | Type D | Type A | Type D | Type A | Type D | Type A | Type D | Type A | Type D | Type A | Type D | Type A | Type D | Type A | Type D | Type A | Type D |
| automobile | 0.000 | 0.000 | 0.000 | 0.000 | 0.000 | 0.000 | 0.167 | 0.083 | 0.000 | 0.000 | 0.000 | 0.000 | 0.000 | 0.000 | 0.000 | 0.000 | 0.000 | 0.000 | 0.000 | 0.042 |
| backdoor | 0.000 | 0.167 | 0.080 | 0.220 | 0.000 | 0.108 | 0.069 | 0.273 | 0.093 | 0.164 | 0.185 | 0.174 | 0.278 | 0.207 | 0.261 | 0.279 | 0.452 | 0.356 | 0.397 | 0.413 |
| campaign | 0.179 | 0.122 | 0.199 | 0.166 | 0.145 | 0.080 | 0.188 | 0.125 | 0.146 | 0.132 | 0.144 | 0.203 | 0.024 | 0.117 | 0.086 | 0.190 | 0.129 | 0.344 | 0.224 | 0.466 |
| cardiotocography | 0.173 | 0.155 | 0.175 | 0.210 | 0.116 | 0.124 | 0.237 | 0.240 | 0.112 | 0.069 | 0.140 | 0.147 | 0.054 | 0.085 | 0.092 | 0.158 | 0.092 | 0.083 | 0.210 | 0.199 |
| census | 0.089 | 0.108 | 0.180 | 0.086 | 0.051 | 0.103 | 0.100 | 0.198 | 0.154 | 0.087 | 0.115 | 0.172 | 0.172 | 0.073 | 0.132 | 0.191 | 0.177 | 0.196 | 0.165 | 0.257 |
| churn | 0.179 | 0.231 | 0.235 | 0.266 | 0.179 | 0.203 | 0.290 | 0.321 | 0.224 | 0.236 | 0.211 | 0.310 | 0.230 | 0.216 | 0.267 | 0.345 | 0.133 | 0.288 | 0.160 | 0.400 |
| cirrhosis | 0.119 | 0.052 | 0.202 | 0.162 | 0.205 | 0.062 | 0.256 | 0.227 | 0.071 | 0.049 | 0.071 | 0.138 | 0.290 | 0.122 | 0.270 | 0.168 | 0.127 | 0.281 | 0.155 | 0.212 |
| covertype | 0.222 | 0.667 | 0.133 | 0.400 | 0.446 | 0.708 | 0.444 | 0.408 | 0.725 | 0.500 | 0.405 | 0.333 | 0.667 | 0.492 | 0.400 | 0.326 | 0.488 | 0.322 | 0.542 | 0.350 |
| credit | 0.106 | 0.077 | 0.173 | 0.319 | 0.010 | 0.077 | 0.185 | 0.258 | 0.025 | 0.177 | 0.072 | 0.268 | 0.026 | 0.206 | 0.160 | 0.268 | 0.111 | 0.140 | 0.253 | 0.349 |
| equip | 0.308 | 0.320 | 0.481 | 0.537 | 0.394 | 0.628 | 0.395 | 0.664 | 0.328 | 0.367 | 0.518 | 0.568 | 0.410 | 0.365 | 0.606 | 0.624 | 0.367 | 0.621 | 0.358 | 0.619 |
| gallstone | 0.004 | 0.077 | 0.107 | 0.051 | 0.089 | 0.000 | 0.200 | 0.112 | 0.045 | 0.042 | 0.106 | 0.025 | 0.250 | 0.150 | 0.125 | 0.192 | 0.071 | 0.228 | 0.050 | 0.174 |
| glass | 0.333 | 0.083 | 0.400 | 0.346 | 0.000 | 0.000 | 0.418 | 0.438 | 0.000 | 0.355 | 0.150 | 0.532 | 0.000 | 0.000 | 0.300 | 0.367 | 0.000 | 0.361 | 0.000 | 0.361 |
| glioma | 0.115 | 0.037 | 0.390 | 0.297 | 0.000 | 0.124 | 0.250 | 0.241 | 0.000 | 0.285 | 0.188 | 0.360 | 0.000 | 0.268 | 0.216 | 0.434 | 0.009 | 0.551 | 0.044 | 0.589 |
| quasar | 0.279 | 0.762 | 0.243 | 0.513 | 0.863 | 0.470 | 0.613 | 0.539 | 0.880 | 0.555 | 0.474 | 0.377 | 0.809 | 0.418 | 0.464 | 0.624 | 0.762 | 0.497 | 0.825 | 0.514 |
| seismic | 0.312 | 0.367 | 0.390 | 0.445 | 0.210 | 0.252 | 0.229 | 0.328 | 0.170 | 0.208 | 0.239 | 0.354 | 0.152 | 0.233 | 0.278 | 0.410 | 0.236 | 0.301 | 0.224 | 0.332 |
| stroke | 0.389 | 0.298 | 0.433 | 0.405 | 0.531 | 0.402 | 0.312 | 0.499 | 0.542 | 0.456 | 0.300 | 0.583 | 0.606 | 0.357 | 0.482 | 0.440 | 0.479 | 0.394 | 0.493 | 0.482 |
| vertebral | 0.265 | 0.095 | 0.255 | 0.171 | 0.167 | 0.148 | 0.343 | 0.415 | 0.233 | 0.200 | 0.320 | 0.265 | 0.212 | 0.259 | 0.209 | 0.326 | 0.233 | 0.350 | 0.425 | 0.422 |
| wbc | 0.045 | 0.097 | 0.077 | 0.129 | 0.114 | 0.121 | 0.285 | 0.323 | 0.014 | 0.131 | 0.050 | 0.281 | 0.023 | 0.138 | 0.134 | 0.268 | 0.110 | 0.259 | 0.170 | 0.340 |
| wine | 0.000 | 0.133 | 0.040 | 0.120 | 0.348 | 0.372 | 0.794 | 0.756 | 0.400 | 0.377 | 0.340 | 0.755 | 0.000 | 0.257 | 0.167 | 0.570 | 0.367 | 0.773 | 0.367 | 0.773 |
| yeast | 0.667 | 0.811 | 0.626 | 0.647 | 0.627 | 0.668 | 0.581 | 0.683 | 0.775 | 0.554 | 0.590 | 0.583 | 0.719 | 0.597 | 0.621 | 0.650 | 0.600 | 0.624 | 0.631 | 0.707 |
| **Average** | 0.180 | **0.233** | 0.229 | **0.261** | 0.225 | **0.233** | 0.318 | **0.357** | 0.235 | 0.235 | 0.220 | **0.306** | **0.234** | 0.217 | 0.261 | **0.343** | 0.247 | **0.349** | 0.285 | **0.400** |

# H    EXPERIMENTAL DETAILS FOR REASONING TEXT ANALYSIS

To assess the impact of reasoning quality on AD performance, we conduct a controlled experiment using test-time prompting. For each dataset, we split the test set in half: one half is used to extract reasoning texts from detected anomalies (based on high anomaly scores), and the other half is used to evaluate the effect of injecting these reasoning examples. Specifically, we vary the number of reasoning examples $\{1, 3, 5, 10\}$ included in the prompt in descending order of anomaly score.

As a metric, we compute the AUROC difference between the baseline (zero-shot prompt without examples) and the model performance after injecting reasoning examples. We further compare the impact of different reasoning types (e.g., Type A vs. Type D) and anomaly score levels (e.g., top-scoring vs. uncertain cases). We observe that high-quality reasoning (Type D) leads to more consistent and larger performance gains, particularly in semantically rich domains like `cirrhosis`.

# I    MORE GRANULAR METADATA ABLATION STUDIES

Beyond the primary metadata variants (Types A–D), we conduct a fine-grained analysis to disentangle the individual contribution of each semantic component. We introduce three additional variants:

- **Type E**: Feature description only.
- **Type F**: Domain knowledge only.
- **Type G**: Normal statistic + Domain knowledge.

These granular ablations reveal several notable patterns across the evaluated LLMs (see Table 16). (i) Domain knowledge provides a strong semantic foundation, and adding feature descriptions yields consistent further improvements (Type G vs. Type D). (ii) Feature descriptions alone are insufficient for anomaly reasoning, as models cannot infer appropriate normal ranges without domain cues—making statistical or domain information essential. For instance, in *cardiotocography*, fetal heart-rate ranges differ substantially from adult ranges (Type E vs. Types B/C). (iii) When only one semantic component is available, domain knowledge is considerably more informative than feature descriptions (Type E vs. Type F).

The metadata in our benchmark follows a natural hierarchy: domain-level descriptions are meaningful only when grounded in feature-level semantics. If feature descriptions are entirely removed, this hierarchy collapses—the model can no longer interpret what each column represents, and the domain information cannot be mapped to the observed values, making anomaly reasoning ill-defined. For this reason, even in our "domain w/o feature description" variants (Type F, G), we retain the feature *names* as a minimal anchor. Removing feature names would make the domain information unusable, as the model would have no interpretable linkage between domain concepts and table attributes. Although this is not a strictly pure domain-only setting, it is a necessary relaxation that preserves coherence and enables meaningful measurement of domain-level effects.

Building on this fine-grained quantitative analysis, we provide the complete results for Types E–G across all 20 datasets and all evaluated LLMs. The AUROC and AUPRC tables (shown in Tables 17 and 18) allow a detailed comparison at both dataset- and model-level granularity.

Table 16: Granular metadata ablations (Types A–G) and average AUROC across 20 datasets.

| Type | $C_{statistical}$ | $C_{feature}$ | $C_{domain}$ | GPT-4o-mini | GPT-4.1 | Claude-3.7 | Qwen3-235B | Gemini-2.5-pro |
|---|---|---|---|---|---|---|---|---|
| A (No Desc.) | ✓ | ✗ | ✗ | 0.692 | 0.696 | 0.725 | 0.665 | 0.691 |
| B | ✓ | ✓ | ✗ | 0.703 | 0.712 | 0.735 | 0.716 | 0.721 |
| C | ✗ | ✓ | ✓ | 0.705 | 0.632 | 0.741 | 0.631 | 0.673 |
| D (Full Desc.) | ✓ | ✓ | ✓ | 0.726 | 0.735 | 0.777 | 0.747 | 0.847 |
| E | ✗ | ✓ | ✗ | 0.584 | 0.581 | 0.648 | 0.562 | 0.571 |
| F | ✗ | ✗ | ✓ | 0.608 | 0.639 | 0.809 | 0.679 | 0.720 |
| G | ✓ | ✗ | ✓ | 0.671 | 0.721 | 0.792 | 0.726 | 0.823 |

## J  FULL GRANULAR RESULTS

Table 17: Comparison of the LLMs' AUROC performance across three granular context types (Type E, Type F, Type G). Each column represents a different LLM evaluated under the three metadata-ablation variants.

| Dataset | GPT-4o-mini | | | GPT-4.1 | | | Claude-3.7-sonnet | | | Qwen3-235B | | | Gemini-2.5-pro | | |
|---|---|---|---|---|---|---|---|---|---|---|---|---|---|---|---|
| | Type E | Type F | Type G | Type E | Type F | Type G | Type E | Type F | Type G | Type E | Type F | Type G | Type E | Type F | Type G |
| automobile | 0.564 | 0.531 | 0.551 | 0.531 | 0.536 | 0.665 | 0.645 | 0.628 | 0.807 | 0.518 | 0.574 | 0.604 | 0.522 | 0.701 | 0.726 |
| backdoor | 0.848 | 0.804 | 0.724 | 0.800 | 0.885 | 0.776 | 0.868 | 0.770 | 0.767 | 0.769 | 0.866 | 0.787 | 0.600 | 0.711 | 0.831 |
| campaign | 0.499 | 0.542 | 0.557 | 0.501 | 0.657 | 0.612 | 0.497 | 0.778 | 0.638 | 0.481 | 0.643 | 0.594 | 0.501 | 0.810 | 0.786 |
| cardiotocography | 0.681 | 0.565 | 0.667 | 0.665 | 0.535 | 0.740 | 0.776 | 0.644 | 0.751 | 0.594 | 0.545 | 0.649 | 0.745 | 0.789 | 0.798 |
| census | 0.325 | 0.579 | 0.591 | 0.711 | 0.607 | 0.666 | 0.795 | 0.896 | 0.819 | 0.627 | 0.741 | 0.719 | 0.692 | 0.790 | 0.854 |
| churn | 0.480 | 0.525 | 0.555 | 0.487 | 0.498 | 0.509 | 0.457 | 0.817 | 0.771 | 0.533 | 0.651 | 0.611 | 0.499 | 0.769 | 0.809 |
| cirrhosis | 0.748 | 0.712 | 0.772 | 0.562 | 0.571 | 0.730 | 0.710 | 0.827 | 0.836 | 0.641 | 0.743 | 0.803 | 0.578 | 0.821 | 0.816 |
| covertype | 0.605 | 0.704 | 0.865 | 0.490 | 0.825 | 0.985 | 0.673 | 0.922 | 0.993 | 0.504 | 0.810 | 0.878 | 0.489 | 0.925 | 0.979 |
| credit | 0.576 | 0.636 | 0.586 | 0.470 | 0.590 | 0.516 | 0.559 | 0.654 | 0.647 | 0.536 | 0.543 | 0.569 | 0.514 | 0.526 | 0.678 |
| equip | 0.822 | 0.844 | 0.915 | 0.845 | 0.883 | 0.959 | 0.901 | 0.926 | 0.971 | 0.856 | 0.866 | 0.959 | 0.908 | 0.884 | 0.974 |
| gallstone | 0.596 | 0.644 | 0.543 | 0.628 | 0.580 | 0.641 | 0.594 | 0.596 | 0.616 | 0.546 | 0.581 | 0.577 | 0.554 | 0.583 | 0.657 |
| glass | 0.769 | 0.769 | 0.811 | 0.830 | 0.830 | 0.908 | 0.826 | 0.857 | 0.874 | 0.667 | 0.862 | 0.888 | 0.748 | 0.885 | 0.910 |
| glioma | 0.459 | 0.459 | 0.626 | 0.497 | 0.656 | 0.631 | 0.454 | 0.878 | 0.877 | 0.457 | 0.788 | 0.709 | 0.492 | 0.705 | 0.887 |
| quasar | 0.600 | 0.689 | 0.753 | 0.468 | 0.325 | 0.860 | 0.544 | 0.956 | 0.938 | 0.522 | 0.869 | 0.909 | 0.456 | 0.217 | 0.929 |
| seismic | 0.595 | 0.681 | 0.706 | 0.503 | 0.665 | 0.606 | 0.594 | 0.691 | 0.654 | 0.513 | 0.666 | 0.667 | 0.551 | 0.640 | 0.672 |
| stroke | 0.517 | 0.538 | 0.544 | 0.516 | 0.509 | 0.587 | 0.605 | 0.770 | 0.734 | 0.519 | 0.624 | 0.616 | 0.492 | 0.698 | 0.768 |
| vertebral | 0.374 | 0.287 | 0.446 | 0.357 | 0.349 | 0.424 | 0.250 | 0.818 | 0.352 | 0.304 | 0.184 | 0.317 | 0.443 | 0.823 | 0.614 |
| wbc | 0.562 | 0.588 | 0.843 | 0.503 | 0.714 | 0.938 | 0.701 | 0.957 | 0.964 | 0.550 | 0.793 | 0.918 | 0.523 | 0.614 | 0.960 |
| wine | 0.417 | 0.525 | 0.572 | 0.752 | 0.959 | 0.856 | 0.840 | 0.994 | 0.990 | 0.537 | 0.700 | 0.915 | 0.558 | 0.995 | 0.982 |
| yeast | 0.639 | 0.528 | 0.787 | 0.512 | 0.535 | 0.808 | 0.673 | 0.809 | 0.847 | 0.557 | 0.527 | 0.832 | 0.551 | 0.517 | 0.832 |

Table 18: Comparison of the LLMs' AUPRC performance across three granular context types (Type E, Type F, Type G). Each column represents a different LLM evaluated under the three metadata-ablation variants.

| Dataset | GPT-4o-mini | | | GPT-4.1 | | | Claude-3.7-sonnet | | | Qwen3-235B | | | Gemini-2.5-pro | | |
|---|---|---|---|---|---|---|---|---|---|---|---|---|---|---|---|
| | Type E | Type F | Type G | Type E | Type F | Type G | Type E | Type F | Type G | Type E | Type F | Type G | Type E | Type F | Type G |
| automobile | 0.451 | 0.434 | 0.446 | 0.445 | 0.458 | 0.562 | 0.565 | 0.530 | 0.764 | 0.448 | 0.488 | 0.498 | 0.433 | 0.625 | 0.671 |
| backdoor | 0.237 | 0.158 | 0.084 | 0.310 | 0.343 | 0.108 | 0.308 | 0.163 | 0.140 | 0.235 | 0.271 | 0.144 | 0.060 | 0.083 | 0.167 |
| campaign | 0.374 | 0.393 | 0.406 | 0.372 | 0.521 | 0.455 | 0.370 | 0.620 | 0.482 | 0.364 | 0.514 | 0.446 | 0.371 | 0.691 | 0.661 |
| cardiotocography | 0.518 | 0.420 | 0.479 | 0.516 | 0.422 | 0.548 | 0.622 | 0.495 | 0.588 | 0.485 | 0.447 | 0.531 | 0.584 | 0.619 | 0.651 |
| census | 0.082 | 0.153 | 0.150 | 0.173 | 0.152 | 0.179 | 0.232 | 0.455 | 0.377 | 0.142 | 0.286 | 0.308 | 0.167 | 0.246 | 0.427 |
| churn | 0.413 | 0.432 | 0.449 | 0.415 | 0.420 | 0.411 | 0.400 | 0.720 | 0.669 | 0.453 | 0.561 | 0.496 | 0.418 | 0.674 | 0.704 |
| cirrhosis | 0.677 | 0.635 | 0.701 | 0.606 | 0.559 | 0.664 | 0.699 | 0.828 | 0.831 | 0.629 | 0.739 | 0.809 | 0.594 | 0.816 | 0.802 |
| covertype | 0.030 | 0.098 | 0.134 | 0.018 | 0.350 | 0.433 | 0.031 | 0.180 | 0.744 | 0.022 | 0.087 | 0.166 | 0.018 | 0.207 | 0.433 |
| credit | 0.409 | 0.462 | 0.410 | 0.353 | 0.436 | 0.358 | 0.397 | 0.504 | 0.497 | 0.388 | 0.381 | 0.408 | 0.368 | 0.388 | 0.523 |
| equip | 0.640 | 0.694 | 0.592 | 0.689 | 0.741 | 0.795 | 0.761 | 0.788 | 0.901 | 0.729 | 0.704 | 0.847 | 0.751 | 0.697 | 0.887 |
| gallstone | 0.573 | 0.597 | 0.523 | 0.593 | 0.557 | 0.610 | 0.581 | 0.565 | 0.576 | 0.522 | 0.549 | 0.545 | 0.551 | 0.567 | 0.597 |
| glass | 0.666 | 0.664 | 0.656 | 0.737 | 0.857 | 0.851 | 0.714 | 0.885 | 0.789 | 0.562 | 0.779 | 0.858 | 0.672 | 0.855 | 0.847 |
| glioma | 0.484 | 0.487 | 0.604 | 0.499 | 0.635 | 0.613 | 0.472 | 0.833 | 0.839 | 0.482 | 0.738 | 0.698 | 0.494 | 0.661 | 0.842 |
| quasar | 0.398 | 0.497 | 0.506 | 0.310 | 0.260 | 0.694 | 0.324 | 0.913 | 0.903 | 0.350 | 0.756 | 0.827 | 0.302 | 0.287 | 0.865 |
| seismic | 0.155 | 0.217 | 0.233 | 0.125 | 0.192 | 0.173 | 0.164 | 0.209 | 0.195 | 0.130 | 0.232 | 0.224 | 0.148 | 0.187 | 0.206 |
| stroke | 0.085 | 0.083 | 0.090 | 0.086 | 0.084 | 0.095 | 0.111 | 0.222 | 0.183 | 0.084 | 0.115 | 0.117 | 0.078 | 0.176 | 0.235 |
| vertebral | 0.452 | 0.461 | 0.462 | 0.431 | 0.432 | 0.441 | 0.376 | 0.761 | 0.400 | 0.411 | 0.363 | 0.414 | 0.484 | 0.772 | 0.572 |
| wbc | 0.556 | 0.587 | 0.811 | 0.505 | 0.707 | 0.916 | 0.650 | 0.947 | 0.956 | 0.544 | 0.769 | 0.893 | 0.527 | 0.618 | 0.936 |
| wine | 0.416 | 0.431 | 0.465 | 0.687 | 0.939 | 0.704 | 0.767 | 0.987 | 0.983 | 0.443 | 0.546 | 0.821 | 0.465 | 0.988 | 0.957 |
| yeast | 0.215 | 0.134 | 0.307 | 0.120 | 0.129 | 0.398 | 0.220 | 0.355 | 0.403 | 0.154 | 0.139 | 0.394 | 0.138 | 0.129 | 0.416 |

## K    SUPPLEMENTARY METRIC: F1-SCORE

In addition to the AUROC and AUPRC metrics, we also include a supplementary comparison using the threshold-based F1-score. While these metrics are widely used in anomaly detection for their threshold-free nature, recent work (McDermott et al., 2024) has pointed out that these metrics may not fully capture model differences under severe class imbalance.

Concretely, following the standard protocol adopted in tabular anomaly detection, including DAGMM (Zong et al., 2018) and GOAD (Bergman & Hoshen, 2020), we select the decision threshold such that the number of predicted anomalies matches the true anomaly count in the test set. This procedure aligns with common practice in the field and enables a consistent and fair comparison across baselines.

Table 19: **F1-score Performance on the ReTabAD.** This table compares training-based methods and our zero-shot LLM (Gemini-2.5-pro). *Full Desc* corresponds to the original prompt setting (Type D), and *No Desc* reports results without metadata descriptions (Type A).

| | Training-based Methods | | | | | | | | | | Zero-shot LLM | |
|---|---|---|---|---|---|---|---|---|---|---|---|---|
| Dataset | IForest | OCSVM | LOF | DeepSVDD | NeuTraL | SLAD | DIF | MCM | DRL | AnoLLM | *No Desc* | *Full Desc* |
| automobile | 0.476 | 0.548 | 0.571 | 0.714 | 0.571 | 0.595 | 0.524 | 0.643 | 0.619 | 0.524 | 0.762 | 0.667 |
| backdoor | 0.022 | 0.016 | 0.870 | 0.872 | 0.880 | 0.858 | 0.165 | 0.651 | 0.877 | 0.395 | 0.083 | 0.146 |
| campaign | 0.560 | 0.565 | 0.507 | 0.564 | 0.586 | 0.572 | 0.508 | 0.560 | 0.608 | 0.562 | 0.459 | 0.686 |
| cardiotocography | 0.660 | 0.711 | 0.684 | 0.660 | 0.628 | 0.590 | 0.590 | 0.677 | 0.633 | 0.537 | 0.527 | 0.706 |
| census | 0.144 | 0.376 | 0.176 | 0.267 | 0.280 | 0.292 | 0.354 | 0.322 | 0.391 | 0.236 | 0.348 | 0.504 |
| churn | 0.408 | 0.516 | 0.401 | 0.418 | 0.537 | 0.463 | 0.394 | 0.519 | 0.518 | 0.465 | 0.434 | 0.698 |
| cirrhosis | 0.780 | 0.768 | 0.793 | 0.732 | 0.756 | 0.744 | 0.780 | 0.744 | 0.780 | 0.756 | 0.622 | 0.768 |
| covertype | 0.323 | 0.833 | 0.771 | 0.779 | 0.850 | 0.848 | 0.871 | 0.838 | 0.796 | 0.698 | 0.158 | 0.368 |
| credit | 0.494 | 0.568 | 0.429 | 0.456 | 0.517 | 0.536 | 0.529 | 0.525 | 0.356 | 0.356 | 0.360 | 0.543 |
| equip | 0.879 | 0.918 | 0.920 | 0.914 | 0.915 | 0.896 | 0.919 | 0.915 | 0.909 | 0.920 | 0.850 | 0.865 |
| gallstone | 0.588 | 0.613 | 0.650 | 0.637 | 0.675 | 0.650 | 0.575 | 0.675 | 0.650 | 0.537 | 0.550 | 0.600 |
| glass | 0.804 | 0.824 | 0.882 | 0.804 | 0.863 | 0.804 | 0.784 | 0.824 | 0.725 | 0.783 | 0.647 | 0.784 |
| glioma | 0.683 | 0.634 | 0.671 | 0.708 | 0.745 | 0.819 | 0.683 | 0.679 | 0.374 | 0.666 | 0.539 | 0.827 |
| quasar | 0.805 | 0.830 | 0.874 | 0.886 | 0.854 | 0.840 | 0.634 | 0.850 | 0.766 | 0.888 | 0.639 | 0.884 |
| seismic | 0.306 | 0.300 | 0.188 | 0.294 | 0.212 | 0.288 | 0.318 | 0.229 | 0.341 | 0.306 | 0.253 | 0.253 |
| stroke | 0.282 | 0.297 | 0.134 | 0.206 | 0.201 | 0.215 | 0.273 | 0.268 | 0.263 | 0.244 | 0.116 | 0.302 |
| vertebral | 0.490 | 0.500 | 0.500 | 0.410 | 0.610 | 0.590 | 0.410 | 0.650 | 0.380 | 0.510 | 0.460 | 0.700 |
| wbc | 0.893 | 0.899 | 0.893 | 0.921 | 0.669 | 0.848 | 0.663 | 0.893 | 0.904 | 0.798 | 0.787 | 0.910 |
| wine | 0.917 | 0.917 | 0.917 | 0.917 | 0.938 | 0.917 | 0.667 | 0.917 | 0.917 | 0.854 | 0.521 | 0.938 |
| yeast | 0.463 | 0.516 | 0.411 | 0.411 | 0.453 | 0.453 | 0.389 | 0.463 | 0.368 | 0.400 | 0.347 | 0.684 |
| **Average** | 0.549 | 0.607 | 0.612 | 0.629 | 0.637 | 0.641 | 0.551 | 0.642 | 0.618 | 0.572 | 0.473 | 0.642 |

## L    LIMITATION OF ZERO-SHOT LLM FRAMEWORK

While semantic metadata generally improves zero-shot LLM scoring, its effectiveness can vary depending on the model's domain familiarity. For example, in cases where domain prior knowledge may be limited, we observe a tendency for the model to rely more on numerical extremeness rather than interpreting the metadata as intended. We also note that metadata may contain sensitive information, so practical deployments may benefit from on-premise or private model setups to ensure that neither raw data nor metadata leaves the organization. This brief discussion is intended to clarify typical considerations when applying metadata-driven LLM scoring in practice.

## M    FURTHER DISCUSSION OF FUTURE WORK

While our benchmark primarily investigates zero-shot LLMs as the simplest architecture capable of directly consuming feature-level textual metadata, we view this as only the starting point for context-aware tabular anomaly detection. An important future direction is to extend traditional tabular AD methods—such as tree models, representation learners, or density-based approaches—with mechanisms for integrating semantic metadata. Potential research avenues include (i) feature–text alignment modules, (ii) hybrid LLM–ML architectures that combine numerical pattern detectors with semantic reasoning, and (iii) multimodal tabular encoders that jointly embed values, schema information, and domain descriptions. We hope ReTabAD catalyzes these developments by providing a unified benchmark and standardized metadata resources.

