# OpenReview forum: "ReTabAD: A Benchmark for Restoring Semantic Context in Tabular Anomaly Detection"
_ICLR.cc/2026/Conference — ICLR 2026 Poster_

### Official Review · Reviewer_7nvM · 2025-10-20

**Soundness:** 3
**Presentation:** 3
**Contribution:** 4
**Rating:** 8
**Confidence:** 4

**Summary:**

This paper proposes a novel benchmark for tabular anomaly detection, comprising 20 datasets from diverse fields, including healthcare, finance, and biology. Those datasets are provided with:
- as little processing as possible of numerical features to preserve their semantic information,
- categorical text restoration and,
- useful metadata including (i) dataset-level descriptions, e.g., dataset name, purpose, description; (ii) human-readable column (feature) descriptions; (iii) label description.

To motivate these modifications and additions to the datasets, the authors test in a One-Class Classification setting, a zero-shot LLM model with and without the added metadata. Their experiments demonstrate that including this metadata significantly enhances the anomaly detection performance of different LLM models (e.g., GPT-4.1, Claude-3.7-sonnet). Additionally, the experiments conducted also emphasize how adding meta-information regarding features and the overall dataset significantly enhances the ability of the LLMs to provide interpretable results.

**Strengths:**

**S1**: The paper is easy to follow, well-written, and properly structured.

**S2**: The proposed work **addresses a relevant caveat** of existing benchmarks for tabular anomaly detection. In particular, authors emphasize the importance of favoring the quality of datasets over their quantity. As mentioned in section 3.1, the authors select datasets so that *too-easy* datasets that have previously been used in the literature are discarded.

**S3**: Their extensive experiments and ablation demonstrate the relevance of including metadata in the case of LLMs.

**Weaknesses:**

**W1**: Some relevant and recent approaches have not been included in the benchmarks, e.g. [1, 2, 3]. It might be worth including them in the benchmark. All three models are open source, and [1,2] have been applied to the ADBench benchmark in [4].

**W2**: While metadata and other additions proposed in the present work appear relevant for LLM-based AD approaches, they appear limited for pure ML-based methods.


[1] Anomaly Detection for Tabular Data with Internal Contrastive Learning. *Tom Shenkar, Lior Wolf*. ICLR 2022

[2] Beyond Individual Input for Deep Anomaly Detection on Tabular Data. *Hugo Thimonier, Fabrice Popineau, Arpad Rimmel, Bich-Liên Doan*. ICML 2024.

[3] Disentangling Tabular Data Towards Better One-Class Anomaly Detection. *Jianan Ye, Zhaorui Tan, Yijie Hu, Xi Yang, Guangliang Cheng, Kaizhu Huang*. AAAI 2025

[4]  DRL: Decomposed Representation Learning for Tabular Anomaly Detection. *Hangting Ye, He Zhao, Wei Fan, Mingyuan Zhou, Dan dan Guo, Yi Chang*. ICLR 2025.

**Questions:**

**Q1**: As mentioned in **W2**, these additions to existing datasets can be relevant but are mostly targeted for LLM-based approaches that could appear as overkill when it comes to anomaly detection on tabular data. Do you have other approaches in mind, apart from LLMs, that could benefit from such augmented datasets (apart from feature engineering)?

**Q2**: A core contribution of the present work also lies in **removing too easy** datasets as mentioned in **S2**. Those datasets often bias the benchmark's overall results.
Nevertheless, the proposed datasets in your benchmark still display a high average AUROC (>0.75 for all models listed in Table 3), which seems quite high. Real-life settings where OCC AD methods apply often include severely imbalanced datasets, where such metrics are rarely obtained (e.g., fraud detection). Have you considered including more severely imbalanced datasets? As shown in Table 7, very few datasets have an anomaly share of less than 10%.

**Q3**: An interesting addition to Table 3, in particular for overall rank comparison, would be your vanilla LLM-based method without the metadata (Type A in Table 5).

---

> ### Author Response · Authors · 2025-11-21
> **Rebuttal by Authors**
>
> Dear Reviewer 7nvM,
>
> We sincerely appreciate your valuable feedback and thoughtful comments. In what follows, we address your concerns one by one.
>
> ---
>
> **[W1] Missing recent baselines**
>
> **[A]** Thank you for pointing out the missing recent methods. Following the suggestion, we have already integrated two of the mentioned approaches—**ICL** [1] and **Disent-AD** [3]—into our benchmark, and updated the code accordingly.
>
> Their experimental results have been added to the revised manuscript, including:
>
> - **Model descriptions** (Appendix B),
> - **Hyperparameter search ranges** (Table 10),
> - **AUROC results** (Table 13), and
> - **AUPRC results** (Table 14).
>
> For **NPT-AD** [2], we attempted integration but encountered an issue in the official repository (see: [link](https://github.com/hugothimonier/NPT-AD/issues/2)). We plan to include this method, and we will update our benchmark accordingly during the rebuttal period.
>
> We appreciate you for highlighting these methods—adding them strengthens the completeness and relevance of our benchmark.
>
> ---
>
> **[W2/Q1] Metadata limitation on pure MLs**
>
> **[A]** While metadata-augmented inputs are beneficial for LLM-based AD, they are not restricted to LLMs. Prior works have shown that classical ML models can integrate domain knowledge or structural constraints in a principled way—for example, informed ML frameworks [4] and constraint-aware decision-tree methods [5]. These studies demonstrate that semantic information such as valid ranges, monotonic relations, type semantics, or consistency rules can be directly used as rule-violation signals or constraint terms in traditional models, independent of any LLM.
>
> Building on this, the metadata provided in ReTabAD (e.g., feature roles, allowed value sets, physical or logical dependencies) can serve exactly as such knowledge sources. For instance, range constraints can be used to compute rule-violation anomaly scores, monotonic relations can be incorporated into monotonic or constraint-regularized models, and dependency rules can support structured consistency checks. Thus, ReTabAD’s metadata enables a wide range of ML-based and hybrid semantic approaches.
>
> Importantly, although constraint- or knowledge-integrated ML approaches have been explored in general settings, they have rarely been applied to tabular anomaly detection—primarily because no benchmark has provided the required semantic information. ReTabAD fills this gap as the first unified testbed that supplies such metadata, enabling both classical ML and LLM-based approaches to exploit semantic information in a controlled and comparable manner.
>
> ---
>
> **[Q2] Too easy dataset issue & Imbalanced dataset**
>
> **[A]** As you mentioned in S2, removing overly easy datasets is our design choice to maintain the quality of the benchmark. Although Table 3 reports AUROC as the main metric, we also examine AUPRC, which is more informative under class imbalance [6]. For example, our zero-shot approach (Gemini-2.5-pro Type D) scored 0.847 in terms of average AUROC (Table 11), whereas 0.625 in terms of average AUPRC (Table 12), indicating that there is still substantial room for improvement.
>
> Regarding imbalance, ReTabAD already includes a meaningful number of severely imbalanced datasets: **6 out of 20 datasets contain fewer than 10% anomalies** (backdoor, census, covertype, seismic, stroke, yeast). Nevertheless, we agree that further expanding toward extremely imbalanced domains would be valuable. We plan to incorporate such datasets in future versions, as long as their anomaly definitions are semantically well-defined and aligned with the benchmark’s goals.
>
> ---
>
> **[Q3] Inclusion of Type A in Table 3**
>
> **[A]** Thank you for the insightful suggestion. This addition will allow readers to clearly compare the effect of metadata on zero-shot LLM performance at a glance. We have incorporated Type A into Table 3 in the revised version.
>
> ---
>
> *Reference*
>
> [1] Anomaly Detection for Tabular Data with Internal Contrastive Learning, *Shenkar and Wolf, ICLR 2022.*
>
> [2] Beyond Individual Input for Deep Anomaly Detection on Tabular Data, *Thimonier et al., ICML 2024.*
>
> [3] Disentangling Tabular Data Towards Better One-Class Anomaly Detection, *Ye et al., AAAI 2025.*
>
> [4] Informed machine learning–a taxonomy and survey of integrating prior knowledge into learning systems, *Von Rueden et al.*, *IEEE TKDE 2021.*
>
> [5] Constraint enforcement on decision trees: A survey, *Nanfack et al.*, *ACM Computing Surveys* 2022.
>
> [6] The Precision-Recall Plot Is More Informative than the ROC Plot When Evaluating Binary Classifiers on Imbalanced Datasets, Saito & Rehmsmeier, *PLoS ONE* 2015.

---

> > ### Comment · Reviewer_7nvM · 2025-11-25
> > **Answer to Rebuttal**
> >
> > Dear Authors,
> >
> > Thank you for your answers and discussion regarding the points we raised in our review.
> >
> > Allow us to mention two elements:
> > - Authors mention [1] as motivation for using AUPRC as the primary metric to compare models in an imbalanced setting. However, recent work [2] has shown how this statement is not necessarily true. We encourage the authors to include threshold dependent metrics like the F1-Score as previously done in the literature. Threshold is usually selected so that predicted anomalies' share match the true anomaly share found in the test set.
> > - Overall, authors should avoid making references to papers using the preprint references when the papers can be found in proceedings of conferences.
> >
> > A few examples:
> >
> > - On diffusion modeling for anomaly detection, -> accepted to ICLR 2024
> > - Classification-Based Anomaly Detection for General Data, -> accepted to ICLR 2020.
> > - Beyond individual input for deep anomaly detection on tabular data, -> accepted at ICML 2024
> > - Unsupervised representation learning by predicting random distances, -> IJCAI 2020
> >
> >
> > [1] The Precision-Recall Plot Is More Informative than the ROC Plot When Evaluating Binary Classifiers on Imbalanced Datasets, *Saito & Rehmsmeier*, PLoS ONE 2015.
> >
> > [2] A Closer Look at AUROC and AUPRC under Class Imbalance. *Matthew B. McDermott, Haoran Zhang, Lasse Hyldig Hansen, Giovanni Angelotti, Jack Gallifant*. NeurIPS 2024

---

> > > ### Author Response · Authors · 2025-11-28
> > > **Official Comment by Authors**
> > >
> > > Thank you for the thoughtful follow-up comments.
> > >
> > > 1. **Reporting threshold dependent metrics:**
> > >
> > >     We appreciate your suggestion. Following your guidance, we conducted additional experiments and reported **F1-Scores** (**Appendix K)**, using the standard setting where the predicted anomaly proportion matches the true anomaly proportion in the test set.
> > >
> > > 2. **Citation corrections:**
> > >
> > >     Thank you for pointing this out. We have updated all relevant references to cite the official conference versions rather than the preprints.
> > >
> > >
> > > Additionally, we have run the **NPT-AD** experiments and incorporated the results into the updated tables **(Table 13, Table 14)**. We appreciate your constructive feedback, which has helped us further strengthen the benchmark and the manuscript.

---

### Official Review · Reviewer_NrBy · 2025-10-26

**Soundness:** 2
**Presentation:** 3
**Contribution:** 2
**Rating:** 4
**Confidence:** 4

**Summary:**

The submission makes two main contributions:

 - It provides augmented versions of 20 tabular anomaly detection datasets by adding detailed information around the columns, in particular, their full names and unnormalized values, textual descriptions of the meanings of the dataset and the columns, and the definitions of normality in terms of the ranges/sets of the values. This has the advantage of making evaluation textually-interpretable, enabling the use of LLMs.

 - A subsequent method for using an LLM in a zero-shot manner for identifying anomalies, given the textual context around a dataset.

Experiments showcase the advantages of using this semantic information when prompting an LLM to identify anomalies in the proposed dataset.

**Strengths:**

The augmented datasets appear thoroughly upgraded with detailed semantic information around the tabular data. The overall motivation of aligning the problem of certain anomaly detection tasks with the underlying semantics is intuitive and sensible.

The LLM-based setup is a reasonable first attempt at such zero-shot anomaly detection, when prompting LLMs for the task. The ablations are a clear demonstration of LLMs being able to use additional semantic information to improve performance at the task.

**Weaknesses:**

1. Perhaps the precise advantages of using LLMs in such a setting could be made clearer.
 - For instance, if we take the time to explicitly "hardcode" the meaning of anomalies through specifying normal ranges and detailed textual descriptions, why not equivalently write out computational filters encoding this meaning? E.g. instead of saying "LB: Normal Range (5-95 percentile) = [118, 146]", we could equivalently encode this with a probabilistic function that could reflect the severity of lying outside the normal range, likely with greater calibration to the semantics of the task if we use further domain knowledge to specify the normality model.  It would be truly interesting if it turned out that LLMs are aware of the gold normality model given the task specification, and internally use such a computation, but this feels very unlikely given contemporary LLM capabilities.
 - Another advantage of explicit encoding might be the interpretability of the anomaly scores; to my knowledge, LLMs are not well-recognized to output meaningfully-calibrated probabilistic scores.
 - The comparisons between the different types of prompts seems hard to disentangle, since only Type D explicitly specifies how to use the normal ranges to identify anomalies in `### Numerical Analysis`, which I'd naively expect to be very helpful.

2. Anomaly detection problems aren't universally ones where the nature of the anomaly is pre-specifiable. In settings such as scientific discovery, for example, one can look for new patterns in data that weren't identified as an interesting axis of variation previously. It would be interesting if such things can be done zero-shot as well.

**Questions:**

* in the prompts, it says 5th - 9th percentile for normal ranges, confirming if this is accurate? (or should it have been 5th - 95th?)

---

> ### Author Response · Authors · 2025-11-21
> **Rebuttal by Authors (1/2)**
>
> Dear Reviewer NrBy,
>
> We sincerely appreciate your thoughtful feedback and valuable insights. Below, we provide point-by-point responses to your comments.
>
> ---
>
> **[W1] Advantages of using LLMs**
>
>
> **[W1-1] Normal-range probabilistic function vs. LLMs**
>
> **[A]** We agree that when anomaly criteria are fully explicit, hard-coded rules or probabilistic filters constitute a valid modeling strategy; TABARD [1] is one such example, of which LLMs extract explicit rules and convert them into Python code that functions as an anomaly detector.
>
> However, in real-world settings, such rules are often incomplete, implicit, or only partially documented. That is where the advantage of using LLMs comes in: **LLMs can infer appropriate rules from general knowledge and the provided metadata**. For example:
>
> - **Inferring missing rules**: Metadata may state only that a feature represents “water temperature,” without specifying that values should stay in the range [0°C, 100°C]; an LLM can fill in such implicit constraints with common knowledge.
> - **Handling context-dependent cases**: Even if a value lies outside a typical “normal range,” it is not necessarily anomalous. For instance, a height of 6'7" (≈200 cm) may be rare in the general population but entirely normal for a group of professional basketball players. Using LLMs, such contextual cases can be naturally reasoned when metadata provides the necessary context.
>
> We also note that pure probabilistic functions may have difficulty capturing context-dependent semantics. For example, distinguishing physiologically abnormal values from values that are unusual but contextually expected (e.g., athlete populations, clinical subgroups). In this scenario, LLMs can leverage the provided metadata to bridge this gap.
>
> Finally, we emphasize that our goal is **not to claim that LLM-based reasoning is the only way to use semantic metadata**, but to demonstrate one concrete instantiation of how metadata can enhance anomaly detection. Rule-based or hard-coded approaches—such as the one you mentioned—are also viable and can be built on top of our benchmark. We hope ReTabAD enables future work to explore these alternative modeling paradigms, including hybrid approaches that combine explicit rules with LLM-based reasoning. We appreciate your insightful perspective.
>
> ---
>
> **[W1-2] Calibrated probabilistic score vs. LLMs**
>
> **[A]** While LLM-generated scores are not probabilistically calibrated in the traditional sense, we note that LLMs can provide a *different* and complementary form of interpretability—namely, explicit reasoning that explains **why** a sample is anomalous rather than merely **how much** it is. In many real-world applications, understanding the underlying cause of an anomaly is often more actionable than interpreting a calibrated score alone. Our framework leverages this strength by coupling the anomaly score with natural-language explanations, which can offer richer, domain-relevant insight even when probability calibration is imperfect. We will clarify this distinction in the revision.
>
> In parallel, we acknowledge that calibration remains a relevant consideration, and future work may explore post-hoc techniques such as temperature scaling or quantile mapping to better balance interpretability and score reliability within the ReTabAD framework.
>
> ---
>
> **[W1-3] Prompt-type comparison clarity**
>
> **[A]** We would like to clarify that the prompt types themselves are cleanly disentangled, as each type is defined by a distinct and non-overlapping set of contextual information. The apparent ambiguity instead arises from the analysis guidelines, which are **inherently qualitative** and therefore less cleanly separable. In Type D—where the LLM receives the richest context, including dataset descriptions, feature semantics, and full numerical statistics—we include explicit numerical-analysis instructions solely to ensure consistent use of this expanded information. Importantly, the LLM already performs baseline numerical reasoning even without such instructions, so Types A–C are not disadvantaged by the absence of explicit numeric guidance. Empirically, we find that performance differences are driven by the information content rather than by variations in guideline phrasing, since the guidelines are lightweight.
>
> ---
>
> **[Concluding Remark]**  While our work highlights the importance of metadata primarily through its effectiveness in LLM-based approaches, we also acknowledge the strengths of other approaches (e.g., rule-based, probabilistic approaches). We believe that hybrid methods—combining these approaches with LLM—offer a promising direction for advancing tabular anomaly detection. We hope that ReTabAD serves as a foundation for such future research, enabling semantic-driven anomaly detection.
>
> ---
>
> *Reference*
>
> [1] TABARD: A Novel Benchmark for Tabular Anomaly Analysis, Reasoning and Detection, *Choudhury et al., EMNLP Findings 2025.*

---

> ### Author Response · Authors · 2025-11-21
> **Rebuttal by Authors (2/2)**
>
> **[W2] Nature of the anomaly is pre-specifiable**
>
> **[A]** Thank you for the interesting suggestion. We believe LLMs can deal with scenarios where no specific anomaly definition is provided. Although our benchmark does not include such settings to maintain fair and well-defined evaluation conditions, we here check the potential by replacing the specific anomaly definition with “NOT NORMAL” in our benchmark. Interestingly, we observe that LLMs can find anomalies using such ambiguous and nonspecific definitions thanks to the semantic metadata (table below).
>
> Through this experiment, we observe that LLMs are generally capable of detecting samples that do not align with normal patterns—suggesting that the open-ended anomaly discovery scenario raised in the reviewer’s question is indeed feasible. At the same time, we find that models with stronger semantic–reasoning capabilities tend to benefit more when explicit anomaly semantics are provided, showing clearer performance gains under the fully specified setting.
>
> |  | **w/ anomaly definition** | **w/o anomaly definition**  |
> | --- | --- | --- |
> | **GPT-4o-mini** | 0.726 | 0.712 |
> | **GPT-4.1** | 0.735 | 0.741 |
> | **Claude-3.7-Sonnet** | 0.777 | 0.766 |
> | **Qwen3-235B** | 0.747 | 0.733 |
> | **Gemini-2.5-Pro** | 0.847 | 0.777 |
> *Metric: AUROC averaged over 20 datasets*
>
> ---
>
> **[Q1] Percentile typo in prompts (5th–9th vs. 5th–95th)**
>
> **[A]** Thank you for catching this. The correct range is 5th–95th percentile, and the “5th–9th” wording appeared only as a typo in the narrative description. All experiments were conducted with the correct 5–95 percentile ranges, as reflected in the attached code. We have fixed the notation in the revised manuscript to avoid confusion.

---

> > ### Comment · Reviewer_NrBy · 2025-11-26
> > **Updated score**
> >
> > Thanks for the responses! I think these address the comments, I'm increasing my score.

---

> > > ### Author Response · Authors · 2025-11-27
> > > **Official Comment by Authors**
> > >
> > > Thank you for your thoughtful review and for taking the time to evaluate our work. We are pleased that the additional explanations and experiments have helped address the concerns you raised. Thank you again for your time!

---

### Official Review · Reviewer_jhCt · 2025-10-31

**Soundness:** 2
**Presentation:** 3
**Contribution:** 3
**Rating:** 6
**Confidence:** 3

**Summary:**

ReTabAD makes a significant contribution by pushing tabular anomaly detection towards a more semantic-aware paradigm, excelling particularly in benchmark construction and the design of a zero-shot LLM framework.

**Strengths:**

Provide a high-quality, resource-rich Benchmark. Innovative introduction and validation of a zero-shot LLM framework, establishing a baseline for semantic utilisation.

**Weaknesses:**

1. While the lack of semantic context in existing benchmarks is clearly highlighted, the paper does not sufficiently explain why traditional methods cannot be easily extended (e.g., by incorporating text embeddings) to address this gap.

2. The proposed concept of the conditional distribution p(x∣M) is somewhat vague and lacks a rigorous mathematical definition or a concrete modelling framework. It serves more as a high-level conceptual goal than a formally grounded theory.

3. Could the limited number of anomaly samples in certain datasets result in unstable evaluation outcomes and reduced statistical reliability?

4. The reliability of the LLM-generated reasoning texts is not assessed through human evaluation or detailed error analysis. It's unclear how often the explanations are factually correct or truly insightful.

5. Perform more granular ablation studies to disentangle the individual contribution of each metadata component (domain, feature description, statistics).

**Questions:**

The author is requested to respond to all the points mentioned under the "Weaknesses".

---

> ### Author Response · Authors · 2025-11-21
> **Rebuttal by Authors (1/2)**
>
> Dear Reviewer jhCt,
>
> We greatly appreciate your constructive feedback and insightful remarks. We address your concerns point by point below.
>
> ---
>
>
> **[W1] Addressing extensibility of traditional methods**
>
> **[A]** Traditional tabular AD methods are not inherently designed to consume or reason over textual semantic metadata, so simply attaching text embeddings is insufficient because these models lack mechanisms to align metadata with tabular inputs or interpret schema-level semantics. For this reason, we employ LLMs as a natural first baseline that can directly leverage rich semantic information without architectural modification.
>
> At the same time, this does not imply that traditional methods cannot be extended. On the contrary, we believe this opens an exciting direction for context-aware tabular anomaly detection. ReTabAD provides the foundation for such follow-up work, including feature-level text alignment, hybrid AD architectures, and multimodal tabular models. Our method is merely the simplest baseline that fully exploits the provided metadata. Indeed, in other domains (e.g., vision, time-series), incorporating textual semantics has already shown substantial benefits [1, 2, 3], suggesting that extending classical tabular AD methods with text embeddings is a promising avenue for future research.
>
> ---
>
> **[W2] Rigorous mathematical Definition**
>
> **[A]** We agree that the notation $p(x \mid \mathcal M)$ may misleadingly suggest a formally defined conditional density, whereas our intention was purely conceptual. We did not mean to imply a rigorous probabilistic modeling framework.
>
> To avoid this ambiguity, we have revised the **Problem Setup (Section 3.4)** to adopt a model-agnostic formulation based on a metadata-aware anomaly scoring function $f(x, \mathcal M)$. This formulation more accurately reflects our goal of characterizing how semantic metadata can influence anomaly detection, without assuming any specific probabilistic framework.
>
> We appreciate your suggestion.
>
> ---
>
> **[W3] Unstable evaluation for the limited anomaly sample dataset**
>
> **[A]** We agree that a small number of anomaly samples can lead to higher variance and reduced statistical reliability. This issue is inherent to many anomaly detection tasks rather than specific to our benchmark. To mitigate this, we report **mean and standard deviation over five independent runs with different random seeds** for all models, following previous anomaly detection studies [4, 5, 6]. Across datasets and methods, the observed variance remains consistently small, as can be seen in **Appendix E, F**, indicating that the evaluation is stable. We will highlight this point more clearly in the revised manuscript.
>
> ---
> *Reference*
>
> [1] On the powerfulness of textual outlier exposure for visual OOD detection, *Park et al., NeurIPS 2023.*
>
> [2] LogicAD: Explainable anomaly detection via VLM-based text feature extraction, *Jin et al., AAAI 2025.*
>
> [3] TimeCMA: Towards LLM-Empowered Multivariate Time Series Forecasting via Cross-Modality Alignment, *Liu et al., AAAI 2025.*
>
> [4] MCM: Masked Cell Modeling for Anomaly Detection in Tabular Data, *Yin et al., ICLR 2024.*
>
> [5] AnoLLM: Large Language Models for Tabular Anomaly Detection, *Tsai et al., ICLR 2025.*
>
> [6] Disentangling Tabular Data Towards Better One-Class Anomaly Detection, *Jianan et al., AAAI 2025.*

---

> ### Author Response · Authors · 2025-11-21
> **Rebuttal by Authors (2/2)**
>
> **[W4] Reliability of the LLM-generated reasoning texts**
>
> **[A]** We agree that assessing the reliability of LLM-generated explanations is an important direction. Our goal in this work, however, is not to evaluate explanation quality itself, but to analyze how semantic metadata influences model behavior, including the interpretability signals produced by LLMs.
>
> In practice, conducting human evaluation for tabular anomaly explanations is especially challenging because the datasets span diverse domains (e.g., healthcare, finance, biology), each requiring domain-specific expertise to reliably determine whether an explanation is factually correct. Moreover, the field currently lacks standardized ground-truth annotations or quantitative metrics for semantic validity in tabular AD.
>
> Instead, we provide **indirect evidences** of explanation reliability: (i) metadata improves downstream detection performance (Table 5), (ii) qualitative example of metadata enhancing explanation quality and coherence (Figure 4), (iii) reasoning texts further improve detection when used as prompts (Figure 3), and (iv) metadata increases key-feature alignment scores (Table 6).
>
> A comprehensive human evaluation or automated factual-consistency scoring would indeed strengthen explanation analysis, and we plan to incorporate such assessments in future work. ReTabAD is specifically designed to support these protocols by offering structured metadata and feature-level semantics that can serve as reference points for explanation evaluation.
>
> Furthermore, standard hallucination-mitigation techniques for LLM reasoning—such as chain-of-thought prompting, schema-constrained generation, and self-consistency—can be naturally incorporated. Because ReTabAD provides explicit feature semantics and domain descriptions, these mechanisms can directly operate on the structured metadata, offering a clear path toward enhancing reasoning reliability in future extensions.
>
> ---
>
> **[W5] Granular metadata ablations**
>
> **[A]** We appreciate the reviewer’s valuable suggestion. We here performed more granular metadata ablations beyond Types A–D to further disentangle the contribution of each component (domain description, feature description, normal statistics).
>
> Using multiple LLMs, we introduced three new variants:
>
> - **Type E (feature-only):** feature description only.
> - **Type F (domain-only):** domain description only.
> - **Type G (statistic + domain):**  normal statistics combined with domain description.
>
> Our additional studies reveal several notable patterns. (i) Domain knowledge already provides a strong semantic foundation, and adding feature descriptions yields consistent further improvements. (Type G vs Type D) (ii) Feature descriptions alone are insufficient for anomaly reasoning, because without domain cues, the model cannot infer appropriate normal ranges—making statistical or domain information essential. For example, in cardiotocography, fetal heart-rate ranges differ substantially from adults. (Type E vs Types B/C). (iii) When only one semantic component is available, domain knowledge is substantially more informative than feature descriptions (Type E vs Type F).
>
> We believe these insights meaningfully deepen the understanding of how metadata components interact. We are currently running the full set of experiments across all LLMs; although only partial results are available during the rebuttal period, the complete analysis will be included in the revised manuscript.
>
> | Type           | $C_{statistical}$ | $C_{feature}$ | $C_{domain}$ | Gemini 2.5-pro | GPT-4.1 | Qwen3-235B |
> | -------------- | ----------------- | ------------- | ------------ | -------------- | ------- | ---------- |
> | A              | ✔️                | ✖️            | ✖️           | 0.691          | 0.696   | 0.665      |
> | B              | ✔️                | ✔️            | ✖️           | 0.721          | 0.712   | 0.716      |
> | C              | ✖️                | ✔️            | ✔️           | 0.673          | 0.632   | 0.631      |
> | D (Full Desc.) | ✔️                | ✔️            | ✔️           | 0.847          | 0.735   | 0.747      |
> | E              | ✖️                | ✔️            | ✖️           | 0.571          | 0.581   | 0.562      |
> | F              | ✖️                | ✖️            | ✔️           | 0.720          | 0.639   | 0.679      |
> | G              | ✔️               | ✖️            | ✔️           | 0.823          | 0.721   | 0.726      |
> *Metric: AUROC averaged over 20 datasets*

---

> ### Comment · Area_Chair_tWvG · 2025-11-27
> **Please reply to the authors' rebuttal**
>
> Dear Reviewer,
>
> The authors have provided their rebuttal. Please reply to it before the rebuttal period ends. Thanks!
>
> Best regards,
>
> AC

---

### Official Review · Reviewer_jF5m · 2025-10-31

**Soundness:** 3
**Presentation:** 2
**Contribution:** 2
**Rating:** 4
**Confidence:** 4

**Summary:**

This paper focuses on providing a tabular anomaly detection benchmark, which contains more textual metadata compared to existing benchmarks. In addition, it provides a new zero-shot LLM baseline for this domain.

**Strengths:**

S1: The studied problem of constructing a textual tabular anomaly detection benchmark is important, and it is useful for the future research.

S2: This benchmark incorporates extensive advanced and classical tabular anomaly detection baseline methods.

S3: The new baseline is simple and effective.

S4: The paper is clear and easy to follow.

**Weaknesses:**

**Datasets perspective:**

W1: The benchmark comprises only 20 datasets, which is significantly fewer than established benchmarks such as ADBench (84 datasets). This limited scale may restrict the comprehensiveness and statistical power of the evaluation.

W2: The decision to exclude datasets where performance is already saturated (Line 181) may be overly restrictive. While such datasets may offer limited room for performance improvement, they remain valuable for evaluating other aspects such as model interpretability, robustness, and consistency. Their exclusion could narrow the scope of the benchmark and limit its utility for holistic model assessment.


W3: The authors will annotate anomalies for new datasets as illustrated in Line 183. However, the process for ensuring annotation accuracy and consistency is not clearly described. Is it trustable? Inaccurate anomaly labels can significantly mislead model evaluation and subsequent research.

W4: It is unclear how categorical features are processed for classical and deep learning baselines.

**Proposed new method perspective:**

W5: Why the Gemini-2.5-pro performs such significantly better than other LLMs according to Table 4? Is there any reasonable explanations? It remains unclear whether the observed gains are due to the proposed framework or inherent model superiority. Moreover, when the LLM is set to GPT, the performance of this method is the worst among all the baseline methods.


W6: Based on W5, it seems that the zero-shot LLM method heavily depends on the model’s pre-existing knowledge, which may limit its applicability in privacy-sensitive or domain-specific scenarios where such prior knowledge is unavailable or unreliable. In contrast, traditional methods rely solely on the data’s intrinsic distribution, offering better generalization in such settings. The authors should discuss the limitations of their approach in real-world deployments where LLM knowledge alignment cannot be assumed.


W7: The proposed method would query LLM to output the anomaly score for each sample. This is much different from other baseline methods, which rely on training to compute the anomaly score. How to ensure the anomaly score is reliable?

W8: Table 6 and Figure 4 present feature attribution results, but it is unclear whether these are based solely on Gemini-2.5-pro or averaged across multiple LLMs. To ensure the generalizability of the findings, the authors should report attribution performance across all evaluated LLMs and discuss whether the observed improvements hold consistently.


W9: Why the reported performance is not consistent with that in their original papers? For example, MCM achieves the AUC-ROC of 0.8902 on Campaign, but the reported performance in this paper is 0.761.

**Questions:**

Please see the weaknesses above.

---

> ### Author Response · Authors · 2025-11-21
> **Rebuttal by Authors (1/3)**
>
> Dear Reviewer jF5m,
>
> We truly appreciate your helpful and insightful feedback. Below, we respond to your comments one by one.
>
> ---
>
> **[W1] Dataset Scale and Benchmark Design Rationale**
>
> **[A]** We agree that larger benchmarks can offer broader coverage, but our benchmark adopts a quality-over-quantity philosophy, a point also emphasized positively by **reviewer 7nvM**. As noted in Section 3.1, we excluded datasets with unclear provenance or ambiguous task definitions to ensure a reliable and semantically coherent benchmark suitable for evaluating context-aware anomaly detection. Following this principle, we include only datasets with accurate feature descriptions, complete metadata, and clearly defined anomaly criteria.
>
> It is also important to emphasize that ADBench and our benchmark serve fundamentally different purposes. ADBench was designed to evaluate numerical AD models’ generalization across a broad range of datasets; thus, maximizing quantitative coverage is central to its design. Indeed, **ADBench [1] reports 57 tabular datasets, of which 10 originate from NLP or CV domains** and were later converted into tabular form for benchmarking. However, many of its datasets lack feature semantics or human-readable descriptions, which limits their suitability for studying context-aware anomaly detection.
>
> We agree that expanding the benchmark is valuable, especially given the variability of anomaly characteristics across domains. We are continually reviewing other datasets, such as **Dropout [2], Clinical Trial [3]**, to expand our benchmark’s coverage. We are actively adding new datasets while preserving semantic quality, which we view as fundamental for meaningful semantic evaluation.
>
> ---
>
> **[W2] Dataset exclusion criteria**
>
> **[A]** We excluded saturated datasets only in extremely limited cases and under a strict and conservative criterion: a dataset was removed only when three or more representative AD methods simultaneously **achieved AUROC and AUPRC above 0.99**. As you mentioned, they remain valuable for evaluating other aspects, but too easy datasets often bias the benchmarks’ overall results, as mentioned by the **Reviewer 7nvM.**
>
> For instance, *breastw* contains meaningful metadata, yet it met this criteria and was therefore excluded. This choice is consistent with the goal of ReTabAD, which is to evaluate models in settings where semantic metadata can contribute to performance. **Reviewer** **7nvM** also noted that discarding trivially solvable datasets is a sound methodological decision.
>
> We emphasize that this exclusion affected only a minimal number of datasets, and its purpose was to avoid evaluation distortion while preserving the benchmark’s focus on semantically grounded anomaly detection. We agree that the rationale for excluding saturated datasets was not sufficiently detailed in the original manuscript and have clarified this in Appendix A.1 of the revision.
>
> ---
>
> **[W3] Annotation accuracy**
>
> **[A]** The annotation process in ReTabAD does not involve subjective labeling or new manual annotation. All datasets are sourced from **widely used public repositories** (e.g., UCI and Kaggle), and anomaly labels are derived from the dataset’s original ground-truth classes and domain definitions. The criteria used to map original labels to anomaly/normal labels are explicitly provided in **Table 8.** This implies that our benchmark ensures annotation accuracy with public repositories, along with annotation consistency with coherent anomaly criteria.
>
> Moreover, we cross-validated the original data sources and corrected several ambiguous or erroneous cases that have been previously reported (e.g., in ADBench), thereby **improving the reliability of the labels** compared to existing benchmarks. These details are now clearly described in **Appendix A.2** of the revision.
>
> ---
>
> **[W4] Categorical value preprocessing in classical model**
>
> **[A]** We perform **hyperparameter optimization over categorical encodings**, searching over `cat_encoding ∈ {integer, one-hot}` . This is described in **Appendix C**.
>
> ---
>
> *Reference*
>
> [1]  ADBench: Anomaly Detection Benchmark, *Han et al., NeurIPS 2022.*
>
> [2] https://archive.ics.uci.edu/dataset/697/predict+students+dropout+and+academic+success
>
> [3] https://data.projectdatasphere.org/projectdatasphere/html/content/119

---

> > ### Comment · Reviewer_jF5m · 2025-11-28
> >
> > Thanks for the detailed response. Regarding the datases perspective, most of the concerns have been addressed. Regarding the new proposed method, my main concerns are unsolved:
> >
> > **For W5:** I agree with the authors that the method achieves improvement over the variant without Desc across different LLMs. However, as I already mentioned, when the LLM is set to GPT, the performance of this method is the worst among all the baseline methods (Table 3), indicating the non-robustness of LLM utilization.
> >
> > **For W6:** What I actually interested is if there is any failure case and how the authors handle them. It is easy to see that LLMs cannot cover all domains, and it is accept for there to be failure cases. In addition, even metadata can also exposure privacy information.
> >
> > **For W7:** My weakness was not to question the emperical performance, but to ask how you ensure stability or reliability from a methodological or theoretical perspective.

---

> > ### Comment · Reviewer_jF5m · 2025-11-28
> >
> > Regarding W2, I agree with the authors explain towards the bias. But I still think that while such datasets may offer limited room for performance improvement, they remain valuable for completeness. If you are concerned about bias, you can explain this part of the data in detail in the paper context.

---

> ### Author Response · Authors · 2025-11-21
> **Rebuttal by Authors (2/3)**
>
> **[W5]** **Performance gap between LLMs**
>
> **[A]** Gemini-2.5-Pro shows stronger performance primarily because it has inherently superior semantic reasoning ability, which allows it to extract and utilize metadata more effectively than other models. This difference reflects the intrinsic capability gap among LLMs, not any model-specific tuning in our framework. Importantly, our method itself remains “LLM-agnostic”: as shown in **Table 4**, all evaluated models—regardless of size or family—benefit from richer contextual metadata, consistently achieving non-trivial AUROC improvements over Type A. This demonstrates that the performance gains arise from our zero-shot framework and metadata design rather than from any particular LLM.  Still, even these weaker models improve with metadata—for example, GPT-4o-mini also shows clear gains, which underscores that *what metadata is provided* matters just as much as *which LLM is used*.
>
> ---
>
> **[W6] LLM prior knowledge dependency**
>
> **[A]** While zero-shot LLMs inevitably rely to some extent on their pre-existing knowledge, we would first like to clarify that our framework does not assume complete background knowledge about any specific domain (e.g., even if a factory’s welding or assembly process is proprietary, general engineering concepts such as current, voltage, temperature, or tool-operation semantics are still widely understood by LLMs). Rather, our experiments demonstrate that—even for domains unfamiliar to the model—explicitly provided metadata (e.g., feature descriptions, class semantics, dataset dictionaries) can yield consistent performance improvements across all evaluated LLMs (**Table 4**).
>
> As the reviewer noted, there are real-world scenarios where privacy or security constraints prevent LLMs from leveraging domain-specific external knowledge. In such settings, a metadata-driven approach may actually be *more* appropriate: users can provide domain information purely in textual form (e.g., data dictionaries) without exposing sensitive raw data to the model, while still allowing the LLM to perform meaningful reasoning based on that context.
>
> We agree that LLMs with weaker semantic reasoning may not fully exploit metadata, which we recognize as an important limitation. However, this also opens several promising research directions enabled by our benchmark. For example, one can develop **hybrid or cooperative models** that combine LLM-based semantic reasoning with traditional ML models’ strengths in numerical pattern recognition. We will explicitly discuss this limitation in the revised version of the paper.
>
> ---
>
> **[W7]** **Reliability of anomaly score**
>
> **[A]** To address the reviewer’s concern regarding the reliability of LLM-generated anomaly scores, we evaluated multi-run consistency across all LLMs studied in our benchmark. Overall, we observe a consistent and stable pattern across models: the anomaly scores produced by LLMs remain highly reproducible across repeated runs, with both score- and rank-level correlations remaining strong except for inherently borderline cases. As summarized in the table below, all models achieve **high average Pearson and Spearman correlations across the 20 datasets (typically around 0.83–0.95)**, indicating strong reliability of the anomaly scores. Lower correlations occur only on datasets containing many borderline samples—a behavior similarly observed in traditional AD models. Consistently, **Appendix E** further shows that AUROC/AUPRC fluctuations across five runs remain negligible, comparable to the variability typically seen in standard anomaly detection methods.
>
> |  | **Pearson (Average of 20 data)** | **Spearman (Average of 20 data)** |
> | --- | --- | --- |
> | **GPT-4.1** | 0.904 | 0.901 |
> | **GPT-4o-mini** | 0.894 | 0.896 |
> | **Claude-3.5-Sonnet** | 0.954 | 0.938 |
> | **Qwen3-235B** | 0.852 | 0.828 |
> | **Gemini-2.5-Pro** | 0.885 | 0.874 |

---

> ### Author Response · Authors · 2025-11-21
> **Rebuttal by Authors (3/3)**
>
> **[W8] Clarifying attribution results across LLMs**
>
> **[A]** As noted in the main text (**L315–316)**, our attribution results are reported solely using Gemini-2.5-Pro for methodological consistency. Because the metadata effect was already observed across all evaluated LLMs (**Table 4**), analyzing attribution with the strongest model provides a representative and stable assessment. Nevertheless, we agree that attribution behavior should be examined across multiple LLMs to ensure generalizability.
>
> Following the reviewer’s suggestion, we repeated the attribution evaluation with the other LLMs used in our benchmark. Across all models, we observed a similar qualitative trend—Type D metadata can improve attribution quality, aligning with the improvements reported in the main results. We have included these additional attribution results across LLMs in **Appendix G**, and to further avoid ambiguity, we have also added a clarifying footnote in the revised version explicitly stating that Table 6 and Figure 4 are based on Gemini-2.5-Pro.
>
> ---
>
> **[W9] Campaign performance difference from the original paper**
>
> **[A]** That is because the datasets used in our benchmark are not identical to those used in the MCM papers. We will clarify this rationale in the revision. To enable fair and consistent comparison across all 20 datasets, we deliberately reprocessed every dataset through a unified pipeline, including missing-value handling, categorical restoration, and normalization. Detailed preprecessing protocol is described in **Appendix 2**.
>
> |  | **MCM version** | **UCI** |
> | --- | --- | --- |
> | **Number of Samples** | 41,188 | 45,211 |
> | **Number of Features** | 62 | 16 |
>
> For the *Campaign* dataset as an example, the variant used in MCM differs substantially from the official UCI version, both in size and feature dimensionality. We presume that other unknown preprocessing (e.g., One-Hot Encoding) occurred. This convention makes fair comparisons across methods hard, and such processing makes it hard to utilize semantic information in modeling.
>
> In contrast, our principle is to apply a transparent and standardized preprocessing procedure. For the *Campaign* dataset as an example, UCI data is used as the source data with preprocessing code. Such preprocessing codes are provided for each data, as *.preprocess.py file. Our benchmark ensures that all methods are evaluated on data that is consistent and reproducible, along with restored semantics.

---

> ### Comment · Area_Chair_tWvG · 2025-11-27
> **Please reply to the authors' rebuttal**
>
> Dear Reviewer,
>
> The authors have provided their rebuttal. Please reply to it before the rebuttal period ends. Thanks!
>
> Best regards,
>
> AC

---

> ### Author Response · Authors · 2025-12-02
> **Official Comment by Authors**
>
> **[W5] Robustness of LLM utilization**
>
> **[A]** We appreciate your opinion and would like to clarify the objective of our work. Your concern addresses robustness of LLM utilization. However, our research objective focuses on **whether semantic metadata consistently improves anomaly detection** regardless of the underlying models’ performance, as stated in the introduction of the paper (L98–99).
>
> From this perspective, we would like to note that:
>
> (1) **Semantic metadata helps anomaly detection for all LLMs**: As shown in Table 4, the *Full Desc* consistently outperforms the *No Desc* variant across all LLMs, including GPT. This provides an affirmative answer to our core research question—"Does semantic metadata help?"—and demonstrates the robustness of metadata utilization.
>
> (2) **Differences in absolute performance come from the LLM itself, not from metadata utilization**: LLMs vary widely in their reasoning quality, calibration, and robustness, and such variability naturally influences their detection performance. These variations arise from LLMs themselves. Hence, these limitations are independent of whether metadata is used and do not indicate a failure of metadata utilization.
>
> Therefore, GPT underperforming baselines does not indicate non-robustness of metadata utilization. Instead, it highlights how metadata interacts with varying model capacities, one of the insights our benchmark is designed to provide.
>
> ---
>
> **[W6] LLM prior knowledge dependency**
>
> **[A]** We fully agree that LLMs cannot cover all domains and that failure cases are unavoidable. We also acknowledge that, in some scenarios, rich metadata alone may not prevent errors. The table below illustrates a representative failure mode in which the LLM may lacks the domain-level prior knowledge needed to correctly interpret the metadata. In this example, GPT-4o-mini assigns an erroneously high anomaly score and provides reasoning that misinterprets a clinically typical value as abnormal, whereas Gemini-2.5-pro produces a domain-consistent explanation. This suggests that, when domain prior knowledge is insufficient, a model’s reasoning can collapse into detecting superficial numerical extremeness rather than true semantic abnormality. We have added a brief discussion of these limitations to the paper (**Appendix L**).
>
> Regarding privacy, our approach can be applied within on-premise or private model environments, where neither raw data nor metadata leave the organization. We clarify this point in the revision to address concerns about potential privacy exposure.
>
> | **LLM** | **Sample index** | **True Label** | **Anomaly Score (0~1)** | **Outcome** | **Reasoning text** |
> | --- | --- | --- | --- | --- | --- |
> | **GPT-4o-mini** | 192 | Normal | **0.9 (almost abnormal)** | **❌ Failure** | Extreme value in degree_spondylolisthesis, indicative of MAJOR ANOMALY. |
> | **Gemini-2.5-pro** | 192 | Normal | **0.4 (correctly normal)** | **✓ Correct** | high degree_spondylolisthesis... This is a classic and severe sign of spondylolisthesis, making it a NON-ANOMALOUS (PATHOLOGICAL) CASE. |
>
> ---
>
> **[W7] Reliability of anomaly score**
>
> **[A]** The reliability of our LLM-based scoring is grounded in prior evidence showing that *LLM-as-a-Judge* performs consistently across diverse domains—including medical [1], time-series [2], and math [3]. Prior works [4, 5] report that LLM evaluators achieve high accuracy and strong agreement with human judgments with reliability driven by consistent, evidence-based reasoning rather than task-specific heuristics. Importantly, [5] highlights that LLM-as-a-Judge becomes more reliable when judgments are **contextualized with structured knowledge** and nuanced evaluation criteria—precisely the role played by our metadata-enriched prompts—indicating that our context-aware scoring aligns with the empirically validated conditions under which LLM-based evaluation is more stable.
>
> ---
>
> **[W2] Dataset exclusion criteria**
>
> **[A]** Thank you for the clarification. We agree that these datasets are valuable for completeness. To address your suggestion, we revisited these datasets (breastw, lymphography) and validated that their behavior is consistent with our benchmark’s overall patterns. Based on this verification, we will clarify this point in the paper and plan to include these datasets in a future revision for completeness.
>
> ---
>
> *Reference*
>
> [1] Evaluation of GPT-4 for 10-year Cardiovascular Risk Prediction, Han et al., iScience 2024
>
> [2] Large Language Models can Deliver Accurate and Interpretable Time Series Anomaly Detection, Liu et al., KDD 2025
>
> [3] Making Large Language Models Better Reasoners with Step-Aware Verifier, Li et al., ACL 2023
>
> [4] Judging LLM-as-a-Judge with MT-Bench and Chatbot Arena, Zheng et al., NeurIPS 2023 dataset and benchmark track
>
> [5] A Survey on LLM-as-a-Judge, Gu et al., arxiv preprint

---

### Author Response · Authors · 2025-11-21
**General Response**

Dear reviewers and AC,

We thank reviewers **jF5m, jhCt, NrBy, and 7nvM** for their constructive and thoughtful feedback. All reviewers agreed on the importance of advancing tabular anomaly detection toward a semantic-aware benchmark, recognizing that ReTabAD addresses a timely gap in the community.

Reviewers highlighted the importance of **ReTabAD** as **a semantic-aware tabular anomaly detection benchmark** (jF5m), emphasizing the **high quality and richness** of the benchmark (jhCt, NrBy) and the importance of prioritizing **dataset quality over quantity (7nvM).** These assessments collectively affirm the validity and necessity of our approach to benchmark construction.

The proposed **zero-shot LLM framework** was recognized as a simple and effective baseline (jF5m, jhCt). Reviewers further noted that the ablation studies provide clear empirical evidence that **semantic metadata substantially improves model behavior** (NrBy, 7nvM). By evaluating this framework alongside **comprehensive classical and modern baselines** (jF5m), our results demonstrate how semantic information enables modeling capabilities that extend beyond conventional feature-based methods—highlighting the unique role of metadata and the broader potential of semantic-aware anomaly detection. Reviewers also praised the **clarity and organization** of the paper (jF5m, 7nvM).

We appreciate these positive assessments and address all remaining concerns in the detailed responses.

To reflect the reviewers’ suggestions, we have carefully revised and enhanced the manuscript with the following updates:

- Addition of zero-shot LLM results for the Type A setting (Table 3)
- Revision of the notation from $p(x \mid M)$ to the metadata-aware scoring function $f(x,M)$ (Section 3.4)
- Clarification of dataset collection and curation criteria (Appendix A.1)
- Clarification of the anomaly annotation procedure and verification of label mappings (Appendix A.2)
- Incorporation of recent baselines: implementation of **ICL** and **Disent-AD**, with **NPT-AD** to be added during the rebuttal period (Appendix B; Tables 10, 13, and 14)
- Expansion of reasoning-alignment experiments (F1-scores@K) across all LLMs (Table 15)
- Improvement of clarity, organization, and wording throughout the manuscript

These updates are temporarily highlighted in **blue** for your convenience to check.

We hope these revisions meaningfully address the reviewers’ concerns and further strengthen the contribution of ReTabAD. Thank you again for your thoughtful feedback.

---

### Author Response · Authors · 2025-12-03
**Rebuttal Summary for AC**

Dear Area Chair,

We sincerely appreciate your time and service to the community.

Across all four reviewers, our submission was consistently recognized for:

- proposing a **meaningful and timely benchmark** **for context-aware tabular anomaly detection (jF5m, jhCt, NrBy, 7nvM)**,
- providing **high-quality datasets with the principle of quality-over-quantity (jhCt, 7nvM)**,
- **empirical analyses showing the importance of metadata** **(NrBy, 7nvM)**,
- and delivering a clear and well-organized manuscript **(jF5m, 7nvM)**.

During the rebuttal period, we believe we have successfully addressed the reviewers’ concerns through additional analyses, clarifications, and revisions. These include:


- Adding **zero-shot LLM results for the Type A setting** (Table 3), enabling clearer comparison of metadata effects.
- **Revising the notation** $p(x∣M)$  **to the metadata-aware scoring function** $f(x, M)$ (Section 3.4).
- Clarifying **dataset collection and curation criteria** (Appendix A.1).
- Clarifying **anomaly annotation procedures and verifying all label mappings** (Appendix A.2).
- **Incorporating recent baselines:** *ICL,* *Disent-AD*, *NPT-AD* (Appendix B; Tables 10, 13, 14).
- Expanding **reasoning-alignment experiments across all LLMs** (F1@K; Table 15).
- Adding more granular **ablation studies of metadata component** (Appendix I).
- Adding **threshold-based F1** metrics as suggested (Appendix K).
- Including a dedicated discussion of **limitations** of our zero-shot LLM framework (Appendix L).
- Including **future directions**, highlighting that our benchmark supports both LLMs and traditional AD methods (Appendix M).
- **Improving clarity** throughout the manuscript and correcting references to official conference versions.


After discussion:

- **Reviewer NrBy** posted follow-up comments that our rebuttal addresses the concerns, and increased the score **(4→6)**.
- **Reviewer 7nvM**, who had initially recommended acceptance **(8)**, did not raise further objections regarding the initial concerns after our clarification (e.g., dataset difficulty and imbalance). Only follow-up points were raised (adding F1 evaluation and correcting citation formats), which we addressed fully in the revision.
- **Reviewer jhCt**, who had initially given a positive score **(6)**, has not yet provided a follow-up response, but our rebuttal provides direct answers to all points, and the added analyses and clarifications are intended to fully resolve the concerns raised.
- **Reviewer jF5m** has mentioned that dataset concerns **(W1, W3, W4)** all have been addressed, and did not raise further objections on **W8 and W9**, indicating these points were resolved. For the remaining concerns, we fully addressed in the revision.


We are grateful for the reviewers’ constructive engagement.

Before the OpenReview issue, our paper had a score of 4-6-6-8, indicating an **overall positive assessment**. We were in the midst of a productive discussion with the reviewers and had already incorporated their feedback in the revised manuscript, we believe that the **remaining concerns had been fully addressed.**

As reviewers can no longer update their scores, we kindly ask the Area Chair to consider the full context of our rebuttal and the revisions made in response to all feedback.

A brief summary of our rebuttal is provided below.

Thank you for your consideration.

Sincerely,

The Authors

---

> ### Author Response · Authors · 2025-12-03
> **Summary of responses to each reviewer**
>
> | **Reviewer** | **Issue Raised** | **Rebuttal Summary** |
> | --- | --- | --- |
> | **jF5m** | **W1**: Only 20 datasets; concerns about limited coverage vs. ADBench | Explained “quality-over-quantity” principle; clarified dataset curation criteria. Currently we are reviewing dataset candidates **(Section 3.1)**. |
> |  | **W2**: Excluding saturated datasets may reduce completeness | Clarified strict exclusion criterion (AUROC & AUPRC ≥ 0.99 across ≥3 baselines). Explained risk of bias from trivially easy datasets **(Appendix A.1)**. |
> |  | **W3**: Trustworthiness of anomaly annotation | Clarified dataset source, criteria for labeling **(Table 8)** deterministic, added detailed annotation verification **(Appendix A.2)**. |
> |  | **W4**: Unclear categorical processing | Clarified categorical encoding HPO over {int, one-hot} **(Appendix C)**. |
> |  | **W5**: non-robust LLM utilization | Clarified that the goal is to show the importance of metadata; **Metadata improves all models** → confirms the value of semantic metadata. |
> |  | **W6**: Failure cases + privacy concerns | Shown the failure case of GPT-4o;  Clarified the limitation **(Appendix L)**. |
> |  | **W7**: Methodological reliability of LLM scores | Shown 5-run stability study; reported high Pearson/Spearman consistency across LLMs. |
> |  | **W8**: Attribution only with Gemini | Added attribution experiments for all LLMs; shown consistent results **(Appendix G)**. |
> |  | **W9**: Mismatch with original MCM numbers | Clarified dataset differences; provided detailed preprocessing explanation. |
>
> ---
>
> | **Reviewer** | **Issue Raised** | **Rebuttal Summary** |
> | --- | --- | --- |
> | **jhCt** | **W1.** Traditional methods could use text embeddings; why not directly extend them? | Explained that text embeddings alone do not provide **contextual semantic reasoning**, which ReTabAD is designed to evaluate; clarified the benchmark enables future hybrid methods. Clarified Benchmark position **(Appendix M)**. |
> |  | **W2.** $p(x \mid M)$ is vague; lacks rigorous probabilistic grounding. | Revised notation to metadata-aware scoring function $f(x,M)$ **(Section 3.4)**. |
> |  | **W3.** Small anomaly counts may lead to unstable evaluation. | Clarified multi-run variance analysis showing variances are small across all datasets **(Appendix E, F)**. |
> |  | **W4.** Reliability of LLM explanations is not assessed. | Provided empirical indicators of reasoning reliability and outlined future work for more direct evaluation. |
> |  | **W5.** Need more granular metadata ablation. | Added expanded metadata-type ablations **(Appendix I)**. |
> ---
>
> | **Reviewer** | **Issue Raised** | **Rebuttal Summary** |
> | --- | --- | --- |
> | **NrBy** | **W1-1:** Hard-coded rules may outperform LLMs | Explained complimentary relation; LLMs infer implicit/contextual rules; provided examples; clarified benchmark goal. |
> |  | **W1-2:** LLM scores not calibrated | Clarified distinction: LLM explanations give semantic insight beyond calibrated probabilities; addressed in revision. |
> |  | **W1-3:** Prompt types hard to disentangle | Clarified prompt-type separation and showed performance is driven by information content, not guideline phrasing. |
> |  | **W2:** AD not always pre-specifiable | Verified feasibility of open-ended anomaly detection (“NOT NORMAL” setting). |
>
> ---
>
> | **Reviewer** | **Issue Raised** | **Rebuttal Summary** |
> | --- | --- | --- |
> | **7nvM** | **W1:** Missing recent baselines | Added recent baselines (ICL, Disent-AD, NPT-AD); results updated **(Tables 10,13,14)**. |
> |  | **W2:** Still high AUROC → not difficult enough | Provided AUPRC analysis showing non-trivial difficulty; highlighted that 6/20 datasets have <10% anomalies; future expansion planned. |
> |  | **Q3:** Include Type A in Table 3 | Added Type A to **Table 3** for clarity. |
> |  | Additional comment during discussion: use F1 | Added threshold-based F1 results **(Appendix K)**. |
> |  | Additional comment during discussion: Incorrect citation | Fixed preprint citations to official versions. |

---

### Meta-Review · Area_Chair_Zxg4 · 2026-01-02

**Summary:**

this paper introduces RetabAD a new benchmark for tabular AD where each dataset is augmented with semantic context.

It also contains a zero-shot LLM baseline with the use of meta data at inference/

reviewers are generally agree the benchmark is timly and useful.

**Reviewer Concerns:**

Some concerns

- Dataset scale (20 is small vs ADBench w/ 57) and the choice to exclude “saturated” datasets. This might be fine as many data in ADBench are quite small

- LLM baseline looks non-robust across LLMs (Gemini strong, GPT weak). Indeed this part is half baked in my view

- Some baseline numbers differ from original paper

The rebuttal clarifies aspects of data curation and may adjust the inclusion criteria. They also added some reliability analysis. Indeed I am not a big fan for "we deliberately reprocessed every dataset through a unified pipeline, including missing-value handling, categorical restoration, and normalization". I would suggest just developing on top of ADBench so things are more comparable.

Remaining concerns:
 - The absolute weakness of some LLM choices (GPT underperforming classic baselines) remains true. During the rebuttal, the authors reframes it as “benchmark reveals model limits”, which is fair, but it still weakens the method story.

- On excluding saturated datasets: rebuttal moved a bit toward “we will clarify / maybe include later”. This is still open

**Reviewer Scores:**

7nvM (was 8): would stay 8.

NrBy (was 4, then explicitly increased): would raised to 6

jhCt (was 6): likely would stay 6. Explanation reliability is still open, so I do not see it jumping to 8.

jF5m (was 4): they still do not buy the robustness/reliability story for the LLM baseline. The added failure case + stability experiments help, but I do not think it fully flips their view. So I guess stay with 4

Final score is 8664 -- an acceptable score for acceptance while not the strong one, though.

---

### Decision · Program_Chairs · 2026-01-26

Accept (Poster)